# BEYOND CALIBRATION: ESTIMATING THE GROUPING LOSS OF MODERN NEURAL NETWORKS

**Alexandre Perez-Lebel, Marine Le Morvan, Gaël Varoquaux**
Soda project team, Inria Saclay, Palaiseau, France
{alexandre.perez,marine.le-morvan,gael.varoquaux}@inria.fr

## ABSTRACT

The ability to ensure that a classifier gives reliable confidence scores is essential to ensure informed decision-making. To this end, recent work has focused on miscalibration, *i.e.*, the over or under confidence of model scores. Yet calibration is not enough: even a perfectly calibrated classifier with the best possible accuracy can have confidence scores that are far from the true posterior probabilities. This is due to the grouping loss, created by samples with the same confidence scores but different true posterior probabilities. *Proper scoring rule* theory shows that given the calibration loss, the missing piece to characterize individual errors is the *grouping loss*. While there are many estimators of the calibration loss, none exists for the grouping loss in standard settings. Here, we propose an estimator to approximate the grouping loss. We show that modern neural network architectures in vision and NLP exhibit grouping loss, notably in distribution shifts settings, which highlights the importance of pre-production validation.

## 1 INTRODUCTION

Validating the compliance of a model to a predefined set of specifications is important to control operational risks related to performance but also trustworthiness, fairness or robustness to varying operating conditions. It often requires that probability estimates capture the actual uncertainty of the prediction, *i.e.* are close to the true posterior probabilities. Indeed, many situations call for probability estimates rather than just a discriminant classifier. Probability estimates are needed when the decision is left to a human decision maker, when the model needs to avoid making decisions if they are too uncertain, when the context of model deployment is unknown at training time, etc.

To evaluate probabilistic predictions, statistics and decision theory have put forward *proper scoring rules* (Dawid, 1986; Gneiting et al., 2007), such as the Brier or the log-loss. Strictly proper scoring rules are minimized when a model produces the true posterior probabilities, which make them a valuable tool for comparing models and selecting those with the best estimated probabilities (*e.g.* Dawid & Musio, 2014). What they do not provide though is a means of validating whether the best estimated probabilities are good enough to be put into production, or whether further effort is needed to improve the model. Indeed, proper scores compound the *irreducible loss* –due to the inherent randomness of a problem, *i.e.* the aleatoric uncertainty– and the *epistemic loss* –which measures how far a model is from the best possible one. For example, a classifier with a Brier score of 0.15 could have optimal estimated probabilities (irreducible loss close to 0.15) or poor ones (irreducible loss close to 0).

Calibration errors are another tool to evaluate probabilistic predictions, and measuring them is an active research topic in the machine learning community. (Kumar et al., 2019; Minderer et al., 2021; Roelofs et al., 2022). The calibration error is in fact a component of proper scoring rules (Bröcker, 2009; Kull & Flach, 2015): it measures whether among all samples to which a calibrated classifier gave the same confidence score, on average, a fraction equal to the confidence score is positive. Importantly, the calibration error can be evaluated efficiently as it does not require access to the ground truth probabilities, but solely to their calibrated version. Calibration is however an incomplete characterization of predictive uncertainty. It measures an aggregated error that is blind to potential individual errors compensating each other. For example, among a group of individuals to which a calibrated cancer-risk classifier assigns a probability of 0.6, a fraction of 60% actually has cancer. But a subgroup of them could be composed of 100% cancer patients while another would only contain 20% of cancer patients.

In general, estimating the true posterior probabilities or obtaining individual guarantees is impossible (Vovk et al., 2005; Barber, 2020). Recent works have thus attempted to refine guarantees on uncertainty estimates at an intermediary subgroup level. In particular, Hébert-Johnson et al. (2018) has introduced the notion of multicalibration, generalizing the notion of calibration within groups studied in fairness (Kleinberg et al., 2016) to every efficiently-identifiable subgroup. Barber et al. (2019); Barber (2020) defines subgroups-based coverage guarantees which lie in between the coarse marginal coverage and the impossible conditional coverage guarantees. In a similar vein, we study the remaining term measuring the discrepancy between the calibrated probabilities and the unknown true posterior probabilities (Kull & Flach, 2015), *i.e.* the grouping loss, for which no estimation procedure exists to date. In particular:

- We provide a new decomposition of the grouping loss into explained and residual components, together with a debiased estimator of the explained component as a lower bound (Section 4).
- We demonstrate on simulations that the proposed estimator can provide tight lower-bounds on the grouping loss (Section 5.1).
- We evidence for the first time the presence of grouping loss on *pre-trained* vision and language architectures, notably in distribution shifts settings (Section 5.2).

## 2 CALIBRATION IS NOT ENOUGH

Calibration can be understood with a broad conceptual meaning of alignment of measures and statistical estimates (Osborne, 1991). However, in the context of decision theory or classifiers, the following definitions are used (Foster & Vohra, 1998; Gneiting et al., 2007; Kull & Flach, 2015):

**True posterior probabilities:** $Q := P(Y = 1|X)$,
**Confidence scores:** $S := f(X)$ score output by a classifier,
**Calibrated scores:** $C := P(Y = 1|S) = \mathbb{E}[Q \,|\, S]$, average true posterior probabilities for a score $S$.

**Confusion about calibration** A common confusion is to mistake confidence scores of a calibrated classifier with true posterior probabilities and think that a calibrated classifier outputs true posterior probabilities, which is false. We identified three main sources of confusion in the literature –see Appendix A for specific quotes. First, the vocabulary used sometimes leaves room for ambiguity, *e.g.*, *posterior probabilities* may refer to confidence scores or to the true posterior probabilities without further specifications. Second, plain-English definitions of calibration are sometimes incorrect, defining calibrated scores as the true posterior probabilities. Lastly, even when everything is correctly defined, it is sometimes implicitly supposed that true posterior probabilities are close to the calibrated scores. While it may be true in some cases, equating the two induces misconceptions.

**Calibration with good accuracy does not imply good individual confidences** It is tempting to think that a calibrated classifier with optimal accuracy should provide confidence scores close to the true posterior probabilities. However, caution is necessary: Figure 1 shows a simple counterexample. The classifier presented gives an optimal accuracy as its confidence scores are always on the same side of the decision threshold as the true posterior probabilities. It is moreover calibrated, as for a given score $s$ (either 0.2 or 0.7 here), the expectation of $Q$ over the region where the confidence score is $s$ is actually equal to $s$. Yet, the confidence scores are not equal to $Q$ as $Q$ displays variance over regions of constant scores. This variance can be made as large as desired as long as both $Q$ and $S$ stay on the same side of the decision threshold to preserve accuracy. The flaws of a perfectly calibrated classifier that always predicts the same score are typically due to variations of the true posterior probabilities over constant confidence scores. As we formalize below, such variations are captured by the grouping loss (Kull & Flach, 2015). Appendix B provides a more realistic variant of this example based on the output of a neural network.

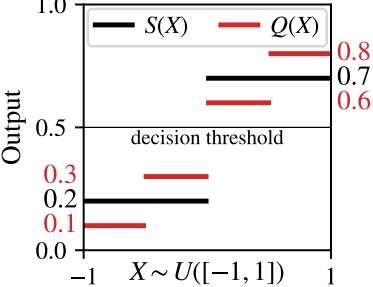

Figure 1: A calibrated binary classifier with optimal accuracy and confidence scores $S(X)$ everywhere different from the true posterior probabilities $Q(X)$.

## 3  THEORETICAL BACKGROUND

**Notations**  Let $(X, Y) \in \mathcal{X} \times \mathcal{Y}$ be jointly distributed random variables describing the features and labels of a K-class classification task. Let $e_k$ be the one-hot vector of size $K$ with its $k^{th}$ entry equal to one. The label space $\mathcal{Y} = \{e_1, \ldots, e_K\}$ is the set of all one-hot vectors of size $K$. We assume that labels are drawn according to the true posterior distribution $Q = (Q_1, \ldots, Q_K) \in \Delta_K$ where $Q_k := P(Y = e_k | X)$ and $\Delta_K$ is the probability simplex $\Delta_K = \left\{ (p_1, \ldots, p_K) \in [0,1]^K : \sum_k p_k = 1 \right\}$. We consider a probabilistic classifier $f$ giving scores $S = f(X)$ with $S = (S_1, \ldots, S_K) \in \Delta_K$. Note that $S$ and $Q$ are random vectors since they depend on $X$. This section introduces the formal definition of the grouping loss, which uses the concepts of calibrated scores as well as scoring rules.

### 3.1  CALIBRATION IN A MULTI-CLASS SETTING

In multi-class settings various definitions of calibration give different trade offs between control stringency and practical utility (Vaicenavicius et al., 2019; Kull et al., 2019). The strongest definition controls the proportion of positives for groups of samples with the same *vector* of scores $S$.

**Definition 3.1.** *A probabilistic classifier giving scores $s = (s_1, \ldots, s_k)$ is **jointly calibrated** if among all instances getting score s, the class probabilities are actually equal to s:*

$$\text{Calibration} \qquad P(Y = e_k | S = s) = s_k \quad \text{for } k = 1, \ldots, K. \qquad (1)$$

The score $S$ being a vector of size $K$ the number of classes, estimating the probability of $Y$ conditioned on $S$ is a difficult task that requires many samples. A weaker notion of multi-class calibration, introduced in Zadrozny & Elkan (2002), requires calibration for each class marginally:

**Definition 3.2.** *A probabilistic classifier giving scores $s = (s_1, \ldots, s_k)$ is **classwise-calibrated** if among all instances getting score $s_k$, the probability of class $k$ is actually equal to $s_k$:*

$$\text{Classwise calibration} \qquad P(Y = e_k | S_k = s_k) = s_k \quad \text{for } k = 1, \ldots, K. \qquad (2)$$

As classwise calibration can still be challenging to estimate when the number of samples per class is too small, an even weaker definition is used in the machine learning community (Guo et al., 2017).

**Definition 3.3.** *A probabilistic classifier giving scores $s = (s_1, \ldots, s_k)$ is **top-label-calibrated** if among all instances for which the confidence score of the predicted class is s, the probability that the predicted class is the correct one is s:*

$$\text{Top-label calibration} \qquad P\big(Y = e_{\text{argmax}(s)} | \max(S) = s\big) = s. \qquad (3)$$

Top-label calibration simplifies the problem by reducing it to a binary problem. However, it has an important limitation (Vaicenavicius et al., 2019): as it only accounts for the confidence of the predicted class, it does not tell whether smaller probabilities are also calibrated.

### 3.2  PROPER SCORING RULES AND THEIR DECOMPOSITION

**Scoring rules**  Scoring rules measure how well an estimated probability vector $S$ explains the observed labels $Y$. The two most widely used scoring rules are the log-loss and Brier score:

$$\text{Log-loss}: \quad \phi^{\text{LL}}(S, Y) := -\sum_{k=1}^{K} Y_k \log S_k \quad \text{Brier score}: \quad \phi^{\text{BS}}(S, Y) := \sum_{k=1}^{K} (S_k - Y_k)^2 \qquad (4)$$

Scoring rules are defined per sample, and the score over a dataset is obtained by averaging over samples. More generally, the *expected score* for rule $\phi$ of the estimated probability vector $S$ with regards to the class label $Y$ drawn according to $Q$ is given by $s_\phi(S, Q) := \mathbb{E}_{Y \sim Q}[\phi(S, Y)]$. Proper scoring rules decompositions have been introduced in terms of their divergences rather than their scores. The *divergence* between probability vectors $S$ and $Q$ is then defined as $d_\phi(S, Q) := s_\phi(S, Q) - s_\phi(Q, Q)$. The divergences for the Brier score and the log-loss read:

$$\text{Log-loss}: \quad d^{\text{LL}}(S, Q) := \sum_{k=1}^{K} Q_k \log \frac{Q_k}{S_k} \quad \text{Brier score}: \quad d^{\text{BS}}(S, Q) := \sum_{k=1}^{K} (S_k - Q_k)^2 \qquad (5)$$

Minimizing the Brier score *in expectation* thus amounts to minimizing the mean squared error between $S$ and the unknown $Q$. A scoring rule is said *strictly proper* if its divergence is non-negative and $d_\phi(S, Q) = 0$ implies $S = Q$. Both the log-loss and Brier score are strictly proper.

**Scoring rules decomposition** Let $C$ be the calibrated scores in the sense of Definition 3.1, the strongest one *i.e.*, $C_k = P(Y = e_k | S = s)$ for $k = 1, \ldots, K$. The divergence of strictly proper scoring rules can be decomposed as (Kull & Flach, 2015):

$$\mathbb{E}\left[d_\phi(S, Y)\right] = \underbrace{\mathbb{E}\left[d_\phi(S, C)\right]}_{\text{Calibration: CL}} + \underbrace{\mathbb{E}\left[d_\phi(C, Q)\right]}_{\text{Grouping: GL}} + \underbrace{\mathbb{E}\left[d_\phi(Q, Y)\right]}_{\text{Irreducible: IL}} \tag{6}$$

where the expectation is taken over $Y \sim Q$ and $X$. CL is the *calibration loss*. IL is the *irreducible loss* which stems from the fact that one point may not have a deterministic label, making perfect predictions impossible. GL is the *grouping loss*. Intuitively, while the calibration loss captures the deviation of the expected score in a bin vs the expected posterior probabilities, the grouping loss captures variations of the true posterior probabilities around their expectation. Together calibration and grouping form the epistemic loss, capturing intrinsic the randomness of the best possible predictor. The scoring rule decomposition (6) holds for top-label calibration (Definition 3.3) as it can be reduced to a binary problem. In the case of classwise calibration, the extension is not straightforward in the general case but we prove in Proposition C.3 that it holds for the Brier score and the log-loss.

## 4 CHARACTERIZATION OF THE GROUPING LOSS

In this section, we focus for simplicity on all settings where the calibrated scores can be expressed as $C_k = \mathbb{E}\left[Y_k | S\right]$, which includes binary classification as well as the multi-class setting with joint or top-label calibration. For classwise calibration, Appendix C.9 shows that all the results presented in this section also hold for the Brier score and log-loss.

### 4.1 REWRITING THE GROUPING LOSS AS A FORM OF VARIANCE

To shed light on the grouping loss, we rewrite it using $f$-variances:

**Definition 4.1** ($f$-variance). *Let $U, V : \Omega \to \mathbb{R}^d$ be two random variables defined on the same probability space, and function $f : \mathbb{R}^d \to \mathbb{R}$. Assuming the required expectations exist, the $f$-variance of $U$ given $V$ is:*

$$\mathbb{V}_f[U \,|\, V] := \mathbb{E}[f(U) \,|\, V] - f(\mathbb{E}[U \,|\, V]).$$

The $f$-variance corresponds to the Jensen gap. It is positive by Jensen's inequality when $f$ is convex. Beyond positivity, it can be seen as an extension of the variance as using the square function for $f$ recovers the traditional notion of variance.

**Lemma 4.1** (The grouping loss as an $h$-variance). *Let $h$ be the negative entropy of the scoring rule $\phi$, i.e. $h : p \mapsto -s_\phi(p, p)$. The grouping loss GL of the classifier $S$ with calibrated scores $C = \mathbb{E}[Q \,|\, S]$ and scoring rule $\phi$ writes:*

$$\underbrace{\mathbb{E}[d_\phi(C, Q)]}_{\text{GL}(S)} = \mathbb{E}[\mathbb{V}_h[Q \,|\, S]] \tag{7}$$

The proof is given in Appendix C.1. In other words, the grouping loss associated to a scoring rule $\phi$ is an $h$-variance of the true posterior probability $Q$ around the average scores $C$ on groups of same level confidence $S$ (Equation 7). In particular for the Brier score, the $h$-variance is a classical variance. It measures discrepancy between $Q$ and $C$ with a squared norm: $\mathbb{V}_h[Q \,|\, S] = \mathbb{E}\left[\|Q - C\|^2 \,|\, S\right]$. For the log-loss, it is a Kullback-Leibler divergence: $\mathbb{V}_h[Q \,|\, S] = \mathbb{E}[D_{\text{KL}}(Q \,\|\, C) \,|\, S]$. These expressions highlight two challenges in estimating the grouping loss. First, it relies on the true posterior probabilities $Q$, which we do not have access to. Second, it involves a conditioning on the confidence scores $S$, which are difficult to estimate for continuous scores.

### 4.2 GROUPING LOSS DECOMPOSITION AND LOWER-BOUND

As an $h$-variance of $Q$ given $S$, evaluating the grouping loss requires access to $Q(X)$ for any point $X$. Unfortunately $Q(X)$ is difficult to estimate, except in special settings – *e.g.* multiple labels per sample as in Mimori et al. (2021). In fact, the scores $S$ of a classifier are generally one's best estimate of $Q$, and the whole point of the grouping loss is to quantify how far this best estimate is from the unknown oracle $Q$. We show that it is nevertheless possible to estimate a lower bound on the grouping loss. On the level set where a classifier score is $S$, it is indeed possible to estimate the average of $Q$

Figure 2: **Intuition.** In the feature space $\mathcal{X}$, the level set of confidence $S = 0.7$ displays $\mathbb{E}[Q \mid S] = 0.7$, which we expect from a calibrated classifier. However, a partition of the level set into 2 regions $\mathcal{R}_1$ and $\mathcal{R}_2$ reveals that $\mathbb{E}[Q \mid S, \mathcal{R}_1] = 0.6$ while $\mathbb{E}[Q \mid S, \mathcal{R}_2] = 0.8$, suggesting a high grouping loss. Intra-region variances $\mathbb{V}_h[Q \mid S, \mathcal{R}_1]$ and $\mathbb{V}_h[Q \mid S, \mathcal{R}_2]$ remain uncaptured.

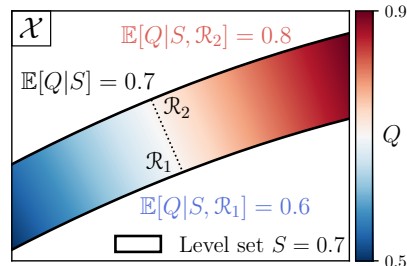

on regions of the feature space. Since by definition $Q$ is non-constant on the level set of a classifier with non-zero grouping loss, it allows to capture part of the grouping loss (Figure 2). Intra-region variance remains uncaptured but can be reduced by choosing smarter and more numerous regions in the partition of the feature space. Theorem 4.1 formalizes this intuition:

**Theorem 4.1** (Grouping loss decomposition). *Let $\mathcal{R} : \mathcal{X} \to \mathbb{N}$ be a partition of the feature space. It holds that:*

$$\text{GL}(S) = \underbrace{\mathbb{E}[\mathbb{V}_h[\mathbb{E}[Q \mid S, \mathcal{R}] \mid S]]}_{\text{GL}_{explained}(S)} + \underbrace{\mathbb{E}[\mathbb{V}_h[Q \mid S, \mathcal{R}]]}_{\text{GL}_{residual}(S)} \tag{8}$$

*Moreover if the scoring rule is proper, then:*
$$\text{GL}(S) \geq \text{GL}_{explained}(S) \geq 0. \tag{9}$$

Appendix C.2 gives the proof by showing that the law of total variance is also valid for the $h$-variance, which allows to decompose the grouping loss into explained and residual terms. $\text{GL}_{explained}$ quantifies the $h$-variance captured through the partition $\mathcal{R}$, *i.e.* coarse-grained $h$-variance reflecting between-region variations of $Q$, while $\text{GL}_{residual}$ captures the remaining intra-region $h$-variance. Due to the positivity of $\text{GL}_{residual}$, $\text{GL}_{explained}$ is a lower-bound of the grouping loss that ranges between 0 and GL depending on how much $h$-variance the partition captures. Importantly, while $\mathbb{V}_h[Q \mid S, \mathcal{R}]$ cannot be estimated because the oracle $Q$ is unknown, it is possible to estimate $\mathbb{E}[Q \mid S, \mathcal{R}]$ and thus $\text{GL}_{explained}$.

### 4.3 CONTROLLING THE GROUPING LOSS INDUCED BY BINNING CLASSIFIER SCORES $S$

The grouping loss as well as $\text{GL}_{explained}$ involve a conditioning on the confidence scores $S$, which cannot be estimated by mere counting when the scores are continuous. To overcome this difficulty, standard practice in calibration approximates the conditional expectation using a binning strategy: the classifier scores are binned into a finite number of values (Definition 4.2).

**Definition 4.2** (Binned classifier). *Let $S : \mathcal{X} \to \Delta_K$ be a classifier. Let $\mathcal{B} := \{\mathcal{B}_j\}_{1 \leq j \leq J}$ be a partition of $\Delta_K$. The binned version of $S$ outputs the average of $S$ on each bin:*

$$S_B : \left| \begin{array}{l} \mathcal{X} \to \mathcal{S} \\ x \mapsto \mathbb{E}[S | S \in \mathcal{B}_j] \end{array} \right. \quad \text{where } \mathcal{B}_j \text{ is the bin } S(x) \text{ falls into.} \tag{10}$$

*The **binned calibrated scores** are defined by:*
$$C_B := P(Y = 1 | S_B) = \mathbb{E}[Q \mid S_B] = \mathbb{E}[C \mid S_B].$$

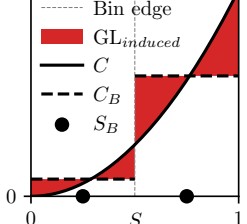

Figure 3: **Binning inflates the grouping loss.**

This is the approach taken by the popular Expected Calibration Error (ECE) (Naeini et al., 2015). However, the loss estimated for a binned classifier deviates from that of the original one. In particular, binning biases the calibration loss downwards (Kumar et al., 2019). Here we show that on the contrary it creates an upwards bias for the grouping loss. Binning a classifier $S$ into $S_B$ boils down to merging the level sets $S$ into a finite number of larger level sets of confidence score $S_B$. For example in Figure 3, all level sets with $S \in [0.5, 1]$ are merged into one level set of confidence $S_B = 0.75$, which artificially inflates the variance of $Q$ in each bin. This intuition is formalized in Proposition 4.1

**Proposition 4.1** (Binning-induced grouping loss). *The grouping loss of the binned classifier $\text{GL}(S_B)$ deviates from that of the original classifier $\text{GL}(S)$ by an induced grouping loss $\text{GL}_{induced}(S, S_B)$:*

$$\underbrace{\mathbb{E}[\mathbb{V}_h[Q \mid S_B]]}_{\text{GL}(S_B)} = \underbrace{\mathbb{E}[\mathbb{V}_h[Q \mid S]]}_{\text{GL}(S)} + \underbrace{\mathbb{E}[\mathbb{V}_h[C \mid S_B]]}_{\text{GL}_{induced}(S, S_B)} \tag{11}$$

*Moreover, if the scoring rule is proper:*
$$\text{GL}_{induced}(S, S_B) \geq 0.$$

Appendix C.3 gives the proof. Proposition 4.1 shows that the difference between the grouping loss of the binned and original classifier is given by the $h$-variance of the original calibrated scores in a bin. This result provides an expression for $\mathrm{GL}_{induced}$ which can then be estimated as shown in Section 4.4.

**Remark 1.** *Interestingly, the binning-induced grouping and calibration losses partly compensate each other (Corollary C.1 in Appendix C.8).*

Applying the decomposition of Theorem 4.1 to the binned classifier $S_B$ and accounting for binning using Proposition 4.1, we obtain a new decomposition of the grouping loss:

**Proposition 4.2** (Explained grouping loss accounting for binning).

$$\mathrm{GL}(S) = \mathrm{GL}_{explained}(S_B) - \mathrm{GL}_{induced}(S, S_B) + \mathrm{GL}_{residual}(S_B) \qquad (12)$$

*If the scoring rule is proper, then:* $\qquad \mathrm{GL}(S) \geq \underbrace{\mathrm{GL}_{explained}(S_B) - \mathrm{GL}_{induced}(S, S_B)}_{\mathrm{GL}_{\mathrm{LB}}(S,S_B)}.$ $\qquad (13)$

The proof is given in Appendix C.4. Importantly, contrary to the grouping loss, both terms in the lower-bound (Equation 13) can be estimated. In the remainder of this paper, we will be interested in the estimation and optimization of the lower bound $\mathrm{GL}_{\mathrm{LB}}(S, S_B)$.

## 4.4 GROUPING LOSS ESTIMATION

We now derive a grouping-loss estimation procedure by focusing on each of its components in turn: $\mathrm{GL}_{explained}(S_B)$ and $\mathrm{GL}_{induced}(S, S_B)$.

**A debiased estimator for the explained grouping loss** $\mathrm{GL}_{explained}(S_B)$ The most natural estimator for the explained grouping loss is a plugin estimator, replacing $\mathbb{E}[Q \mid S, \mathcal{R}]$ by the empirical means of $Y$ over each region. It is nonetheless generally biased. We show below that in the case of the Brier scoring rule, a direct empirical estimation of $\mathrm{GL}_{explained}$ on the partition is biased upwards (cf Appendix C.6), and propose a debiased estimator.

**Proposition 4.3** (Debiased estimator for the Brier score). *For all class $k \in \{1, \dots, K\}$ and bin $s \in \mathcal{S}$, let $n^{(s,k)}$ (resp. $n_j^{(s,k)}$) be the number of samples belonging to level set $\mathcal{R}^{(s)}$ (resp. region $\mathcal{R}_j^{(s)}$). We define the empirical average of $Y$ over these regions as:*

$$\hat{\mu}_j^{(s,k)} := \frac{1}{n_j^{(s,k)}} \sum_{i:X^{(i)} \in \mathcal{R}_j^{(s)}} Y_k^{(i)} \quad and \quad \hat{c}^{(s,k)} = \frac{1}{n^{(s,k)}} \sum_{i:X^{(i)} \in \mathcal{R}^{(s)}} Y_k^{(i)}$$

*The debiased estimator of* $\mathrm{GL}_{explained}$ *is:* $\quad \widehat{\mathrm{GL}}_{explained}(S_B) = \sum_{k=1}^{K} \sum_{s \in \mathcal{S}} \frac{n^{(s,k)}}{n} \widehat{\mathrm{GL}}_{explained}^{(s,k)}(S_B)$

*with:*

$$\widehat{\mathrm{GL}}_{explained}^{(s,k)}(S_B) = \underbrace{\sum_{j=1}^{J} \frac{n_j^{(s,k)}}{n^{(s,k)}} \left(\hat{\mu}_j^{(s,k)} - \hat{c}^{(s,k)}\right)^2}_{\text{plugin estimator } \widehat{\mathrm{GL}}_{plugin}} - \underbrace{\left(\sum_{j=1}^{J} \frac{n_j^{(s,k)}}{n^{(s,k)}} \frac{\hat{\mu}_j^{(s,k)}(1 - \hat{\mu}_j^{(s,k)})}{n_j^{(s,k)} - 1} - \frac{\hat{c}^{(s,k)}(1 - \hat{c}^{(s,k)})}{n^{(s,k)} - 1}\right)}_{\text{bias estimation } \widehat{\mathrm{GL}}_{bias}}$$

Appendix C.5 gives the proof, with a debiasing logic similar to Bröcker (2012). The leftmost term corresponds to the plugin estimate: the estimator of the explained grouping loss (Theorem 4.1) with sample estimators for the quantities of interest. The two rightmost terms represent the finite-sample variance in estimating expectations over regions. They correct the upwards bias of the plugin estimate.

**Estimation of the grouping loss induced by binning classifier scores.** $\mathrm{GL}_{induced}(S, S_B)$ involves the $h$-variance of the calibrated scores $C$ inside each bin, thus its estimation requires $C$. A solution is to estimate a continous calibration curve $\hat{C}$, which amounts to a one-dimensional problem for which various methods are available. In our experiments, we use a kernel-based method (*e.g.* LOWESS). It is then easy to compute the $h$-variance of $\hat{C}$ inside each bin by evaluating $\hat{C}$ for all available samples. The resulting expression of the estimator $\widehat{\mathrm{GL}}_{induced}$ is given in Appendix C.7.

**A partition to minimize** $\text{GL}_{residual}$ In order to achieve the best possible lower-bound, we choose partitions in Theorem 4.1 to minimize $\text{GL}_{residual}$. We use a decision tree with a loss corresponding to the scoring rule –squared loss for Brier score– on the labels $Y$ to define regions that minimize the loss on a given level set of $S$. As this approach relies on $Y$, a train-test split is used to control for overfitting: a partitioning of the feature space is defined using the leaves of the tree fitted on one part, then the empirical means used in $\widehat{\text{GL}}_{explained}$ are estimated on the other part given this partitioning. In the experiments of Section 5.2, we work in the output space of the penultimate layer of the networks.

## 5 EXPERIMENTAL STUDY

### 5.1 SIMULATIONS: FINER PARTITIONS GIVE A TIGHT GROUPING-LOSS LOWER BOUND

Here we investigate the behavior of our estimation procedure with respect to the number of bins and number of regions on simulated data with known grouping loss. The importance of both corrections - the binning-induced grouping loss (Proposition C.1) and the debiasing (Proposition 4.3) - is also evaluated. For this, data $Y \in \{0, 1\}$ is drawn according to a known true posterior probability $Q$ and we consider a calibrated logistic regression classifier for the scores $S$ (details in Appendix B.2 and Figure 11). The estimation procedures are then applied according to two different scenarios. First we vary the number of samples per region (*e.g.* region ratio) while the number of bins is fixed (Figure 4a.). Then we vary the number of bins while the region ratio is fixed (Figure 4b.).

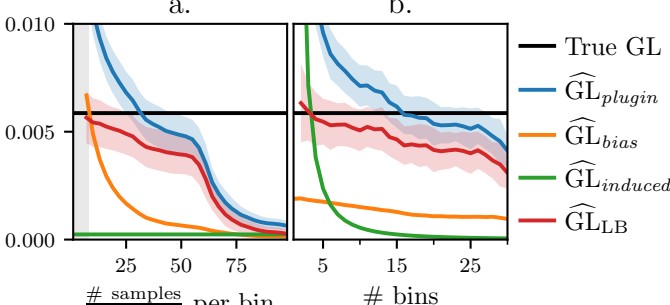

Figure 4: **Simulation:** estimating the grouping loss lower bound $\widehat{\text{GL}}_{\text{LB}}$ on a simulated problem (Appendix B.2, Figure 11). Right has a fixed ratio $\frac{\text{\# samples}}{\text{\# regions}} = 30$ per bin. Bins are equal-width. Averaged curves are plotted with a $\pm 1$ standard deviation envelop.

For a fine-enough partition (a large number of regions, and hence a small region ratio), $\widehat{\text{GL}}_{\text{LB}}$ provides a tight lower bound to the true grouping loss GL. If the average number of samples per region becomes too small, some regions have less than two samples, which breaks the estimate (grayed out area in Figure 4a.). Conversely, the naive plugin estimate $\widehat{\text{GL}}_{plugin}$ substantially overestimates the true grouping loss as it does not include the corrections $\widehat{\text{GL}}_{induced}$ and $\widehat{\text{GL}}_{bias}$. Figure 4b. shows that to control the $\text{GL}_{induced}$ due to binning, a reasonably large number of bins is needed, *e.g.* 15 as typical to compute ECE. Given these bins, we suggest to use a tree to divide them in as many regions as possible while controlling the probability of regions ending up with less than two samples, typically targeting a region ratio of a dozen, to obtain the best possible lower bound $\widehat{\text{GL}}_{\text{LB}}$.

### 5.2 MODERN NEURAL NETWORKS DISPLAY GROUPING LOSS

**The grouping diagram: visualization of the grouping loss** In a binary setting, calibration curves display the calibrated scores $C$ versus the confidence scores $S$ of the positive class. To visualize the heterogeneity among region scores in a level set, we add to this representation the estimated region scores $\hat{\mu}_j$, *i.e.* the fraction of positives in each region obtained from the partitioning of level sets (Figure 5). The further apart the region scores are, the greater the grouping loss.

**Vision** We evaluate 15 vision models (listed on Figure 7) from PyTorch *pre-trained* on ImageNet-1K (Deng et al., 2009). Here we report evaluation on ImageNet-R (Hendrycks et al., 2021), an ImageNet variant with 15 different renditions: paintings, toys, tattoos, origami... The dataset contains 30 000 images and 200 ImageNet classes. Appendix D reports evaluation on the validation set of ImageNet-1K and ImageNet-C (Hendrycks & Dietterich, 2019), an ImageNet variant with corrupted versions of ImageNet images. As often with many classes, the small number of samples per class (50) does not allow to study the classwise calibration and grouping loss. Hence, following common practice, we consider top-label versions (Definition 3.3). Appendix D gives experimental details.

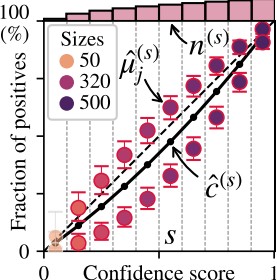

Figure 5: **Grouping diagram.** Calibration curve of a binned binary classifier augmented with the estimated region scores $\hat{\mu}_j$ for a partitioning of each level set into 2 regions. Region sample sizes are plotted as a gradient color. The classifier is binned into 10 equal-width bins whose sample sizes $n^{(s)}$ are given as an histogram. A Clopper-Pearson 95% confidence interval is plotted on the region scores. Regions for which the calibrated score $\hat{c}^{(s)}$ lie within this interval are grayed out.

We find substantial grouping loss inside level sets for most networks on ImageNet-R (ConvNeXt Tiny and ViT L-16 in Figure 6, others in Appendix D.1), even after post-hoc recalibration (Figure 6 right). For instance, while ConvNext + Isotonic is calibrated (third graph), it is strongly over-confident in one part of the feature space and under-confident in the other, creating a high grouping loss.

Figure 7 shows that the grouping loss varies across architectures, even with comparable accuracy. For example, ViT has a slightly better accuracy than ConvNeXt, but a lower estimated grouping loss. Post-hoc recalibration does not affect the grouping loss (Figure 6 right and Figure 7 right), leading to the same conclusions (see Appendix C.10 for the analytical impact of recalibration on the grouping loss). We observe the same effects on ImageNet-C (Appendix D.2), but little or none on ImageNet-1K (Appendix D.3). This suggests that stronger grouping loss arises in out-of-distribution settings. Visual inspection of the images suggests that the partitions capture heterogeneity coming from how realistic an image is, or the different rendition types (Appendix D.1).

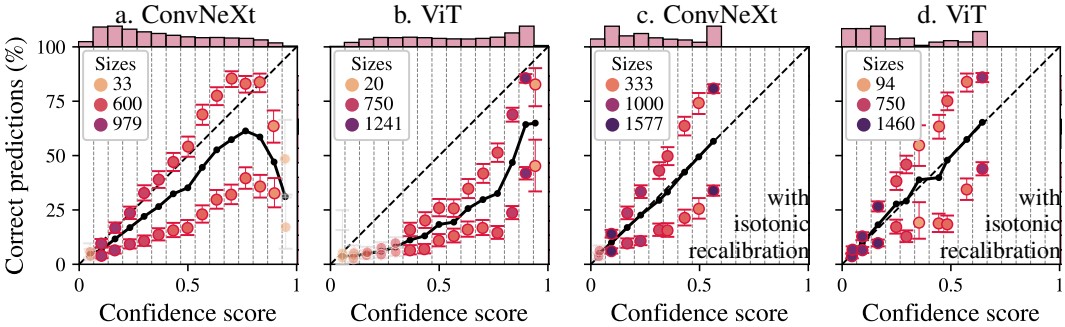

Figure 6: **Vision**: Fraction of correct predictions versus confidence score of predicted class ($\max_k S_k$) for ConvNeXt Tiny and ViT L-16 on ImageNet-R, without post-hoc recalibration (a. and b.) and with isotonic recalibration (c. and d.). In each bin on confidence scores, the level set is partitioned into 2 regions with a decision stump constrained to one balanced split, with a 50-50 train-test split strategy.

Figure 7: **Evaluating vision models**: a de-biased estimate of the grouping loss lower bound $\widehat{\mathrm{GL}}_{\mathrm{LB}}$ (Equation 13) and an estimate of the calibration loss $\widehat{\mathrm{CL}}$, both accounting for binning, evaluated on ImageNet-R and sorted by model accuracy. Partitions $\mathcal{R}$ are obtained from a decision tree partitioning constrained to create at most $^{\#\,\text{samples in bin}}/_{30}$ regions in each bin. Isotonic regression is used for post-hoc recalibration of the models (right). Table 1 gives the raw values.

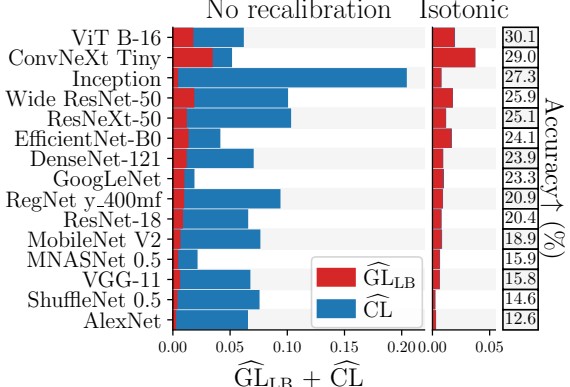

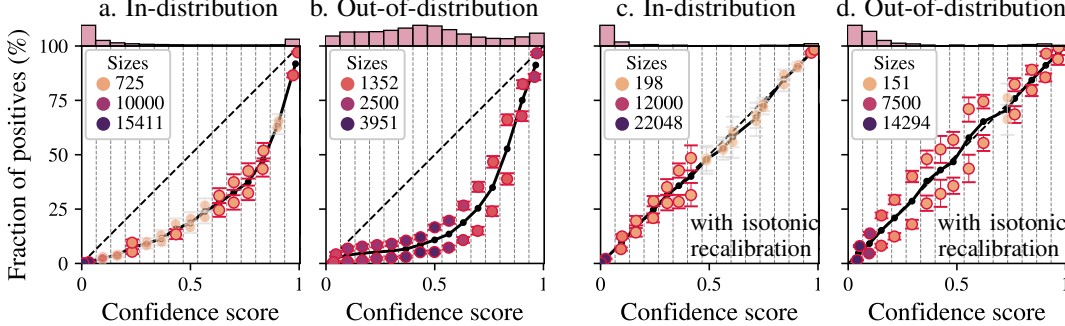

Figure 8: **NLP**: Fraction of positives versus confidence score of the positive class of fine-tuned BART for zero-shot classification on the test set of Yahoo Answers Topics without post-hoc recalibration (a. and b.) and with isotonic recalibration (c. and d.). The test set is either restricted to the 5 topics on which the network was trained (in-distribution) or to 5 unseen topics (out-of-distribution). In each level set clusters are built with a balanced decision stump and a 50-50 train-test split strategy.

**NLP**  We evaluate the grouping loss on BART Large (Lewis et al., 2019) from HuggingFace *pre-trained* on the Multi-Genre Natural Language Inference dataset (Williams et al., 2018). We consider zero-shot topic classification on the Yahoo Answers Topics dataset, composed of questions and topic labels. There are 60 000 test samples and 10 topics. The model is fine-tuned on 5 out of the 10 topics of the training set, totaling 700 000 samples. Given a question title and a hypothesis (*e.g.* "This text is about Science & Mathematics"), the model outputs its confidence in the hypothesis to be true. The classification being zero-shot, the hypothesis can be about an unseen topic. We evaluate the model separately on the 5 unseen topics and the 5 seen topics of the test set. Both results in a binary classification task on whether the hypothesis is correct or not. Appendix E gives experimental details.

The partitioning reveals grouping loss in the out-of-distribution setting both before and after recalibration (Figure 8b. and d.). However, we found no evidence of grouping loss in the in-distribution setting. As in vision, this suggests that out-of-distribution settings lead to stronger grouping loss.

## 6  DISCUSSION AND CONCLUSION

**A working estimator of grouping loss**  While calibrated scores can be estimated by solving a one-dimensional problem, the grouping loss is much harder to estimate: it measures the discrepancy to the true posterior probabilities, which are unknown. We show that combining debiased partition-based estimators with an optimized partition captures the grouping loss well. This procedure allows us to characterize the grouping loss of popular neural networks for the first time. We find that in vision and NLP, models can be calibrated –if needed via post-hoc recalibration– but significant heterogeneity of errors remains, *e.g.* ConvNeXt has larger grouping loss than calibration loss.

Several avenues could be explored to better capture the grouping loss. Complex level sets may not be approximated well with the partitioning defined by a tree, leaving a large residual in th. 4.1. In this case, the estimated grouping loss may only be a rather loose lower bound. Such a lower bound is nevertheless useful to reject models with high grouping loss. In addition, we apply the tree on the penultimate layer of neural networks, where class boundaries are simplified. Finally, complementing the proposed lower bound with an upper bound would also allow to identify models without grouping loss.

**We need to talk about grouping loss**  Model should be evaluated not only on aggregate measures, but also on their individual predictions, using grouping loss. The presence of grouping loss means that the model is systematically under-confident for certain groups of individuals and over-confident for others, questioning the use of such models for individual decision making. The presence of grouping loss also means that downstream tasks relying on confidence scores can be hindered, such as causal inference with propensity scores or simulation-based inference. Finally, this heterogeneity raises fairness concerns. In fact, the grouping loss and our lower bound are fundamentally related to fairness –see sufficiency and group calibration (Barocas et al., 2017, chap 3), and multicalibration (Kleinberg et al., 2016). We hope that our measure of grouping loss will spur new research in this area.

## Reproducibility statement

All datasets are publicly available (ImageNet-R, ImageNet-C, ImageNet-1K, Yahoo Answers Topics) and all models involved are pre-trained and publicly available on PyTorch and HuggingFace. Simulated examples are described in Appendix B. Detailed experimental methods are given in Appendix D and E. Proofs of all theoretical results are listed in Appendix C. The source code for the implementation of the algorithm, experiments, simulations and figures is available on GitHub: https://github.com/aperezlebel/beyond_calibration.

## Acknowledgments

We acknowledge support in part by the French Agence Nationale de la Recherche under Grant ANR-20-CHIA-0026 (LearnI).

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

# SUPPLEMENTARY MATERIALS

## Table of Contents

## A    EXAMPLES OF CONFUSING STATEMENTS ON CALIBRATION

Here we detail specific examples of confusing statements on calibration in the literature. We choose most of these examples in well-cited and well regarded works.

- Kuhn & Johnson (2013): "*We desire that the estimated class probabilities are reflective of the true underlying probability of the sample. That is, the predicted class probability (or probability-like value) needs to be well-calibrated. To be well-calibrated, the probabilities must effectively reflect the true likelihood of the event of interest.*"

  The authors write that it is desirable to have confidence scores $S$ reflective of the true posterior probabilities $Q$, which is indeed desirable as discussed in Section 1. However, they write this is obtained through calibration. Although post-hoc recalibration makes the confidence scores closer to $Q$ in some sense, there is an implicit shortcut. As pointed out in Section 2 and Appendix B, calibration, even with optimal accuracy, does not guarantee confidence scores $S$ to be close to the true posterior probabilities $Q$.

- (Gupta et al., 2020): "*A classifier is said to be calibrated if the probability values it associates with the class labels match the true probabilities of correct class assignments.*"

  The authors write that calibration is matching the confidence scores $S$ of a classifier to the true posterior probabilities $Q$. In fact, calibration is matching the confidence scores $S$ to the calibrated scores $C$, which can be far from the true posterior probabilities $Q$ as pointed out in Section 2 and Appendix B.

- Garcin & Stéphan (2021): "*Ideally, we would like machine learning models to output accurate probabilities in the sense that they reflect the real unobserved probabilities. This is exactly the purpose of calibration techniques, which aim to map the predicted probabilities to the true ones in order to reduce the probability distribution error of the model.*"

  The authors write that calibration is outputting confidence scores $S$ that are true posterior probabilities $Q$. As in the previous citations, calibration is outputting calibrated scores $C$, which can be far from $Q$ (Section 2 and Appendix B).

- Flach (2016): "*A probabilistic classifier is well calibrated if, among the instances receiving a predicted probability vector p, the class distribution is approximately distributed as p. Hence, the classifier approximates, in some sense, the class posterior.*" "*The main point is that knowing the true class posterior allows the classifier to make optimal decisions. It therefore makes sense for a classifier to (approximately) learn the true class posterior.*"

  Here, calibration is rightly defined as outputting confidence scores $S$ that are equal to the calibrated scores $C$. However, by writing that confidence scores $S$ of a calibrated classifier approximate the true class posterior $Q$, the author makes an implicit assumption that the calibrated scores $C$ are close to the true posterior probabilities $Q$, which is not guaranteed in theory as pointed out in Section 2 and Appendix B.

## B    EXAMPLES OF ACCURATE AND CALIBRATED CLASSIFIERS WITH HIGH GROUPING LOSS

Here we build simple binary classification examples of calibrated classifiers with optimal accuracy having their confidence scores far from the true posterior probabilities. In Appendix B.1 we build examples with an arbitrary link between true posterior probabilities $Q$ and confidence scores $S$ (up to a limit to keep the classifier's accuracy optimal). In Appendix B.2 we build a more realistic example based on the output of a neural network.

### B.1    ARBITRARY LINK BETWEEN TRUE POSTERIOR PROBABILITIES Q AND CONFIDENCE SCORES S

To show that calibration, even combined with optimal accuracy, does not impose strong constraints on how close the true posterior probabilities $Q$ should be from the classifiers' confidence scores $S$, we build examples in which $Q$ and $S$ have an arbitrary link. For simplicity we consider binary examples with a one-dimensional feature space $\mathcal{X}$. These can be extended to multiple dimensions by projecting onto a vector $\omega$ (via $x \mapsto \omega^T x$).

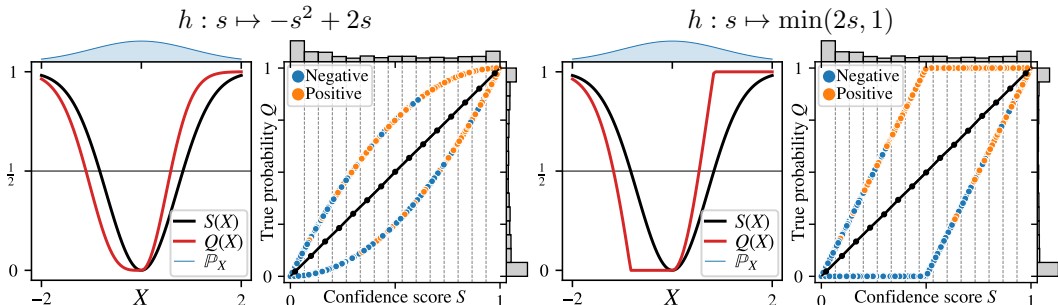

Figure 9: **Calibrated but not accurate.** Example of calibrated classifiers $S$ constructed from links $h$ following the procedure described in Appendix B.1. The accuracy of these two classifiers is not optimal as $Q$ and $S$ are not on the same side of the decision threshold ($\frac{1}{2}$) wherever $Q \neq \frac{1}{2}$. Refer to Figure 10 for an example with optimal accuracy. Calibration curves (in black on 2$^{\text{nd}}$ and 4$^{\text{th}}$ plot) are obtained from 1 million samples.

The idea is to build a classifier that outputs confidence scores having at most two antecedents each. One antecedent should have its true posterior probability $Q$ at an arbitrary distance $+\Delta$ from the associated confidence score $S$, while the other has a distance $-\Delta$. Scores with only one antecedent should have $Q = S$. This combined with an equal density weight of $\mathcal{X}$ onto the two antecedents guarantees calibration: $\mathbb{E}[Q \,|\, S] = S$. To maintain the classifier's accuracy optimal, the offset $\Delta$ is constrained to keep $Q$ and $S$ on the same side of the decision threshold.

To achieve this, we cut the one-dimensional feature space $\mathcal{X}$ into three parts: $\mathbb{R}^\star_+$, $\mathbb{R}^\star_-$ and $\{0\}$. As a classifier, we take an even function $S(X)$ with $S^{-1}(\{0\})$ reduced to a singleton so that each confidence score has either two antecedents (one in $\mathbb{R}^\star_+$ and one in $\mathbb{R}^\star_-$) or one antecedent in $\{0\}$. To assign an equal weight to each antecedent, we choose a symmetric distribution for $\mathcal{X}$, *e.g.* a standard normal distribution centered on $0$. We build the true posterior probabilities $Q$ from deviations $h : [0, 1] \to [0, 1]$ and $g : [0, 1] \to [0, 1]$ of the confidence scores $S$ in $\mathbb{R}^\star_+$ and $\mathbb{R}^\star_-$:

$$Q : x \mapsto \mathbb{1}_{x>0} h(S(x)) + \mathbb{1}_{x<0} g(S(x)) + \mathbb{1}_{x=0} S(0) \tag{14}$$

For $S$ to be calibrated, deviations must average to identity, *i.e.* $\forall s \in S(\mathbb{R}), \frac{1}{2}(h(s) + g(s)) = s$. A proof of this statement is given below:

*Proof.*

$$\mathbb{E}[Q(X) \,|\, S(X)] = \mathbb{E}[\mathbb{1}_{X>0} \,|\, S(X)] \, h(S(X)) + \mathbb{E}[\mathbb{1}_{X<0} \,|\, S(X)] \, g(S(X)) + \mathbb{E}[\mathbb{1}_{X=0} \,|\, S(X)] \, S(0)$$
$$= P(X>0|S(X)) \, h(S(X)) + P(X<0|S(X)) \, g(S(X)) + P(X=0|S(X)) \, S(0)$$
$$= \tfrac{1}{2} \mathbb{1}_{S(X) \neq S(0)} (h(S(X)) + g(S(X))) + \mathbb{1}_{S(X) = S(0)} S(0)$$

since $P(X > 0|S(X)) = P(X < 0|S(X)) = \frac{1}{2} \mathbb{1}_{S(X) \neq S(0)}$.

Hence, $S(X) \text{ calibrated} \Leftrightarrow \mathbb{E}[Q(X) \,|\, S(X)] = S(X) \Leftrightarrow \frac{1}{2}(h(S(X)) + g(S(X))) = S(X)$. □

From here, we choose $h : [0, 1] \to [0, 1]$ and define $g : s \mapsto 2s - h(s)$. Note that to keep $g(s) \in [0, 1]$, $h$ is constrained by: $\forall s \in S(\mathbb{R}), 2s - 1 \leq h(s) \leq 2s$. At this point of the procedure, classifiers $S$ may not have an optimal accuracy. Figure 9 shows two examples of links $h$, one of which saturates the constraint $h(s) <= \min(2s, 1)$.

To make the classifiers accurate, the deviations $h(s) - s$ should be small enough to keep $S$ and $Q$ on the same side of the decision threshold. This adds two constraints on $h$: $\forall s \in S(\mathbb{R}) \cap [0, \nicefrac{1}{2}[, h(s) < \frac{1}{2}$ and $\forall s \in S(\mathbb{R}) \cap [\nicefrac{1}{2}, 1], h(s) \geq \frac{1}{2}$ (with the convention that a score of exactly $\frac{1}{2}$ predicts the positive class). Figure 10 (left) shows a classifier built following the above procedure. Figure 10 (right) shows that we can release the constraint $\nicefrac{1}{2}(h(s) + g(s)) = s$ if we tweak the distribution of $\mathcal{X}$ to adapt the weights between the two antecedents accordingly (and take *e.g.* $g(s) = \mathbb{1}_{h(s)<s}$).

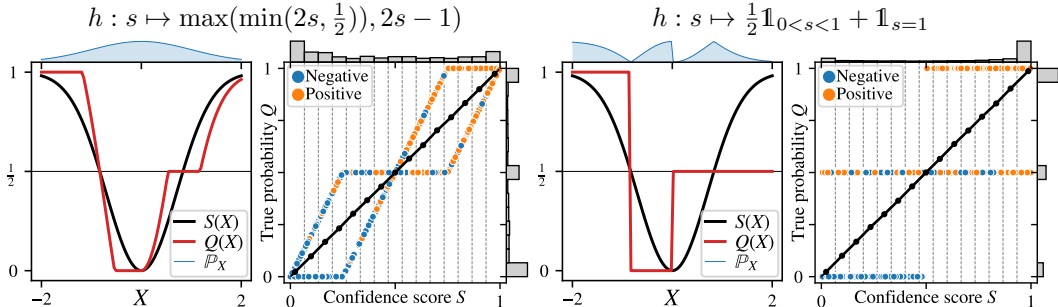

Figure 10: **Calibrated and optimal accuracy.** Example of calibrated classifiers $S$ constructed from links $h$ following the procedure described in Appendix B.1. The accuracy of these two classifiers is optimal as $Q$ and $S$ are on the same side of the decision threshold ($\frac{1}{2}$) wherever $Q \neq \frac{1}{2}$. However, confidence scores $S$ are almost everywhere different from the true posterior probabilities $Q$. Calibration curves (in black on 2nd and 4th plot) are obtained from 1 million samples.

## B.2 REALISTIC EXAMPLE BASED ON NEURAL NETWORK'S OUTPUT

The examples of Appendix B.1, while proving our point, are quite unusual in practice especially in the choice of classifier $S$. In this section we build a more realistic example based on the output of a neural network. We focus on a binary classification setting with a feature space $\mathcal{X}$ being at least two-dimensional. The classifier is taken as a sigmoid of $\omega^T X$ (akin to the last layer of a neural network predicting the confidence score of the positive class). Based on this choice of model, we build a class of calibrated and accurate classifiers with confidence scores $S$ far from the true posterior probabilities $Q$.

The idea is to create heterogeneity in the blind spot of calibration, *i.e.* orthogonally to $\omega$. The perturbations creating heterogeneity must balance each other out to keep the classifier calibrated.

To achieve this, we define:

- $d \geq 2$ the dimension of the feature space $\mathcal{X}$.
- $\omega \in \mathbb{R}^d$, the last layer's weights.
- $\varphi : \mathbb{R} \to [0, 1]$ the link function mapping $\omega^T x$ to confidence scores, *e.g.* a sigmoid.
- $S : x \in \mathbb{R}^d \mapsto \varphi(\omega^T x) \in [0, 1]$ the classifier's confidence scores of the positive class.
- $\omega_\perp \in \mathbb{R}^d$ such that $\omega^T \omega_\perp = 0$, the direction in which heterogeneity will be introduced.
- $\psi : \mathbb{R} \to [-1, 1]$ an odd perturbation introducing balanced heterogeneity along $\omega_\perp$.
- $\Delta_{max} : x \mapsto \min(1 - S(x), S(x))$ modulating the range of the perturbation to keep $Q \in [0, 1]$.
- $Q : x \in \mathbb{R}^d \mapsto S(x) + \psi(\omega_\perp^T x)\Delta_{max}(x) \in [0, 1]$ the constructed true posterior probabilities.
- $X \sim \mathcal{N}(0, \Sigma)$ the data distribution, with $\Sigma \in \mathbb{R}^{d \times d}$ having $\omega$ and $\omega_\perp$ among its eigenvectors.

With the above construction, the classifier $S$ is calibrated. Indeed,

$$\mathbb{E}[Q(X) \,|\, S(X)] = S(X) + \mathbb{E}\big[\psi(\omega_\perp^T X)\Delta_{max}(X) \,\big|\, S(X)\big] \tag{15}$$

$$= S(X) + \mathbb{E}\big[\psi(\omega_\perp^T X) \,\big|\, S(X)\big] \Delta_{max}(X) \tag{16}$$

since $\Delta_{max}(X)$ is a function of $S(X)$. We have $\mathbb{E}\big[\psi(\omega_\perp^T X) \,\big|\, S(X)\big] = 0$ by construction: $\psi$ is odd and the distribution of $X$ has a symmetric weight along $\omega_\perp$ since $\Sigma$ is aligned on $\omega$ and $\omega_\perp$. Hence $\mathbb{E}[Q(X) \,|\, S(X)] = S(X)$. Figure 11 shows two examples generated with this procedure. However, it is not yet accurate. As in Appendix B.1, the perturbation should be constrained to keep $Q$ and $S$ on the same side of the decision threshold to keep the accuracy optimal. This is simply achieved by defining $\Delta_{max} : x \mapsto \min(1 - S(x), S(x), |\frac{1}{2} - S(x)|)$.

**a. Classifier** $S(X) = (1 + \exp(-\omega^T X))^{-1}$

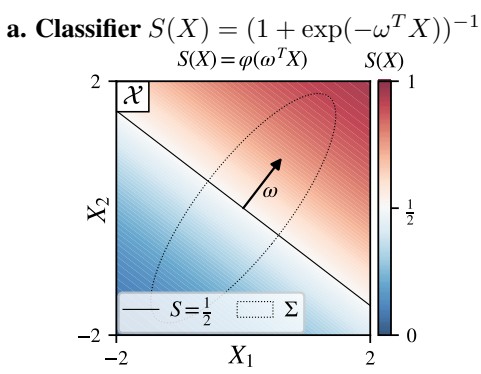

**b. Perturbation** $\psi(z) = 2(1 + \exp(-z))^{-1} - 1$

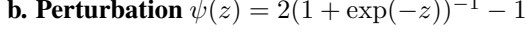

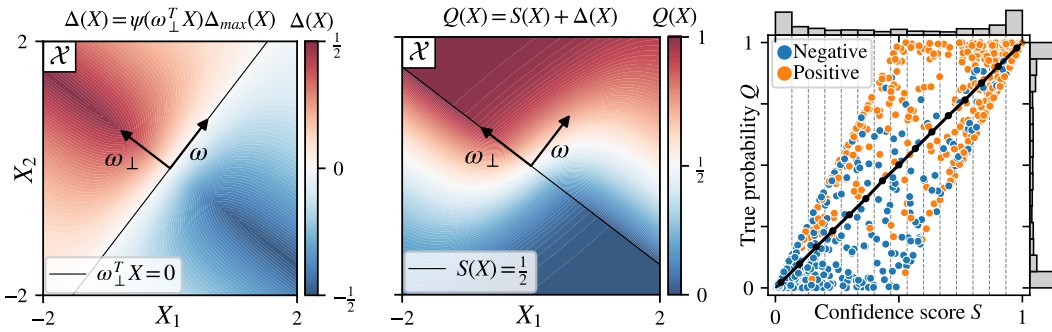

**c. Perturbation** $\psi(z) = \mathbb{1}_{z>0} - \mathbb{1}_{z<0}$

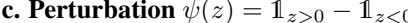

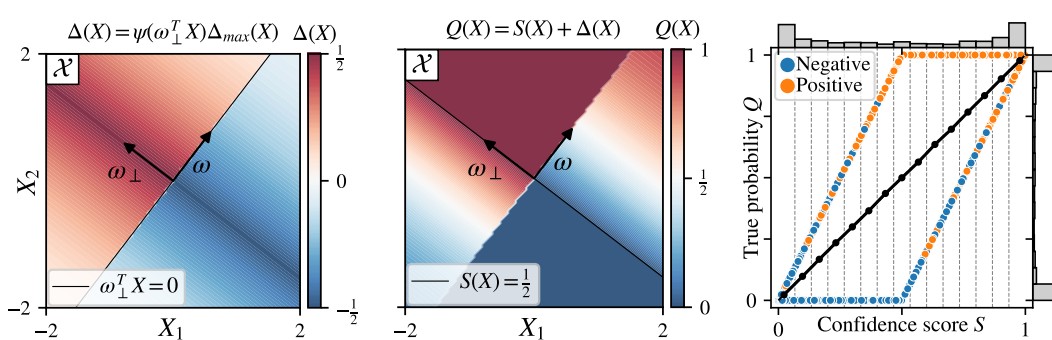

Figure 11: **Calibrated but not accurate.** Example of a calibrated classifier $S$ constructed following the procedure described in Appendix B.2. Its accuracy is not optimal as $Q$ and $S$ are not on the same side of the decision threshold ($\frac{1}{2}$) wherever $Q \neq \frac{1}{2}$. Refer to Figure 12 for an example with optimal accuracy. Calibration curves (in black on last column) are obtained from 1 million samples.

**a. Classifier** $S(X) = (1 + \exp(-\omega^T X))^{-1}$

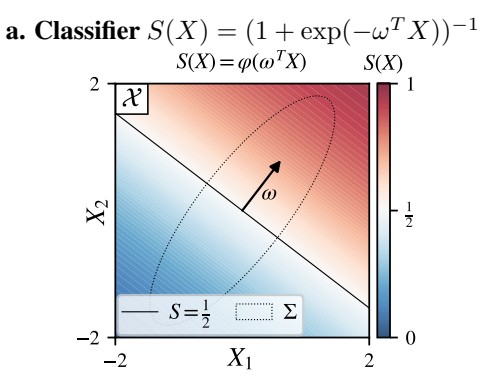

**b. Perturbation** $\psi(z) = 2(1 + \exp(-z))^{-1} - 1$

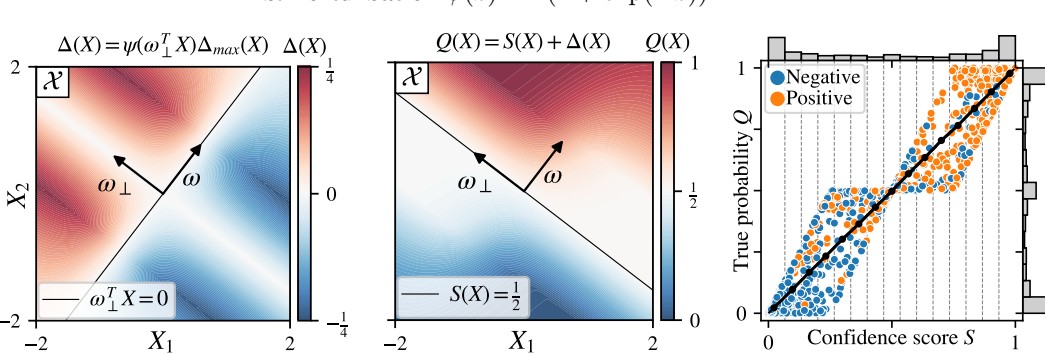

**c. Perturbation** $\psi(z) = \mathbb{1}_{z>0} - \mathbb{1}_{z<0}$

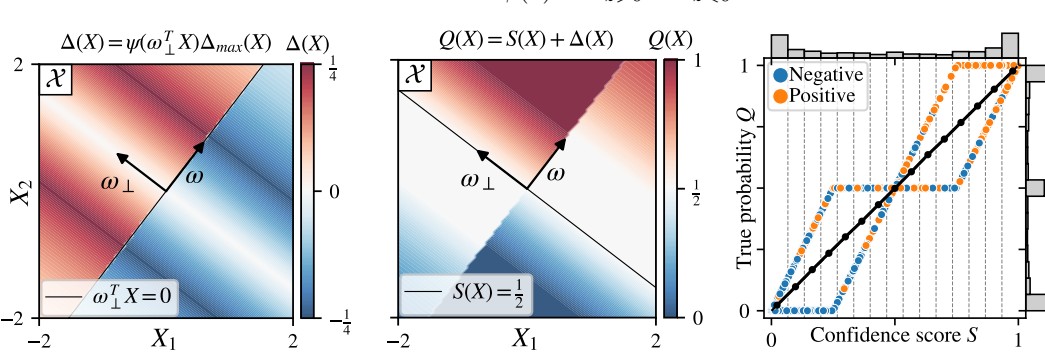

Figure 12: **Calibrated and optimal accuracy.** Example of a calibrated classifier $S$ constructed following the procedure described in Appendix B.2. Its accuracy is optimal as $Q$ and $S$ are on the same side of the decision threshold wherever $Q \neq \frac{1}{2}$. However, confidence scores $S$ are almost everywhere different from the true posterior probabilities $Q$. Calibration curves (in black on last column) is obtained from 1 million samples.

## C  PROOFS

### C.1  LEMMA 4.1: THE GROUPING LOSS AS AN H-VARIANCE

**Lemma 4.1** (The grouping loss as an $h$-variance). *Let $h$ be the negative entropy of the scoring rule $\phi$, i.e. $h : p \mapsto -s_\phi(p, p)$. The grouping loss GL of the classifier $S$ with calibrated scores $C = \mathbb{E}[Q \mid S]$ and scoring rule $\phi$ writes:*

$$\underbrace{\mathbb{E}[d_\phi(C, Q)]}_{\mathrm{GL}(S)} = \mathbb{E}[\mathbb{V}_h[Q \mid S]] \tag{7}$$

*Proof of Lemma 4.1.* Let $\phi$ be a scoring rule, $h : p \mapsto -s_\phi(p, p)$ and $C = \mathbb{E}[Q \mid S]$.

$$
\begin{align}
\mathbb{E}[d_\phi(C, Q)] &= \mathbb{E}[s_\phi(C, Q) - s_\phi(Q, Q)] & \text{Definition of divergence} \tag{17} \\
&= \mathbb{E}[s_\phi(C, Q) + h(Q)] & \text{Definition of } h \tag{18} \\
&= \mathbb{E}\left[\sum_{k=1}^{K} \phi(C, e_k)Q_k + h(Q)\right] & \text{Definition of expected score} \tag{19} \\
&= \mathbb{E}\left[\mathbb{E}\left[\sum_{k=1}^{K} \phi(C, e_k)Q_k + h(Q) \,\middle|\, S\right]\right] & \text{Law of total expectation} \tag{20} \\
&= \mathbb{E}\left[\sum_{k=1}^{K} \mathbb{E}[\phi(C, e_k)Q_k \mid S] + \mathbb{E}[h(Q) \mid S]\right] & \text{Linearity of expectation} \tag{21} \\
&= \mathbb{E}\left[\sum_{k=1}^{K} \phi(C, e_k)\mathbb{E}[Q_k \mid S] + \mathbb{E}[h(Q) \mid S]\right] & \phi(C, e_k) \text{ is a function of } S \tag{22} \\
&= \mathbb{E}\left[\sum_{k=1}^{K} \phi(C, e_k)C_k + \mathbb{E}[h(Q) \mid S]\right] & C_k = \mathbb{E}[Q_k \mid S] \tag{23} \\
&= \mathbb{E}[-h(C) + \mathbb{E}[h(Q) \mid S]] & \text{Definition of } h \tag{24} \\
&= \mathbb{E}[\mathbb{E}[h(Q) \mid S] - h(\mathbb{E}[Q \mid S])] & C = \mathbb{E}[Q \mid S] \tag{25} \\
&= \mathbb{E}[\mathbb{V}_h[Q \mid S]] & \text{Definition of } \mathbb{V}_h[Q \mid S] \tag{26}
\end{align}
$$

$\square$

### C.2  THEOREM 4.1: GROUPING LOSS DECOMPOSITION

**Lemma C.1** (Law of total $h$-variance). *Let $X, Y, Z : \Omega \to \mathbb{R}^d$ be random variables defined on the same probability space and a function $f : \mathbb{R}^d \to \mathbb{R}$. The law of total variance holds for the $f$-variance:*

$$\mathbb{V}_f[Y \mid Z] = \mathbb{E}[\mathbb{V}_f[Y \mid X, Z] \mid Z] + \mathbb{V}_f[\mathbb{E}[Y \mid X, Z] \mid Z] \tag{27}$$

*Proof.*

$$
\begin{align}
\mathbb{E}[f(Y)] &= \mathbb{E}[\mathbb{E}[f(Y) \mid X]] & \text{Law of total expectation} \\
&= \mathbb{E}[\mathbb{V}_f[Y \mid X]] + \mathbb{E}[f(\mathbb{E}[Y \mid X])] & \text{Definition of } \mathbb{V}_f[Y \mid X]
\end{align}
$$

$$
\begin{align}
\mathbb{E}[f(Y)] - f(\mathbb{E}[Y]) &= \mathbb{E}[\mathbb{V}_f[Y \mid X]] + \mathbb{E}[f(\mathbb{E}[Y \mid X])] - f(\mathbb{E}[\mathbb{E}[Y \mid X]]) & \text{Law of total expectation} \\
&= \mathbb{E}[\mathbb{V}_f[Y \mid X]] + \mathbb{V}_f[\mathbb{E}[Y \mid X]] & \text{Definition of } \mathbb{V}_f[\mathbb{E}[Y \mid X]]
\end{align}
$$

The same proof holds when the expectations and $h$-variances are conditioned on $Z$. $\square$

**Theorem 4.1** (Grouping loss decomposition). *Let $\mathcal{R} : \mathcal{X} \to \mathbb{N}$ be a partition of the feature space. It holds that:*

$$\mathrm{GL}(S) = \underbrace{\mathbb{E}[\mathbb{V}_h[\mathbb{E}[Q \mid S, \mathcal{R}] \mid S]]}_{\mathrm{GL}_{explained}(S)} + \underbrace{\mathbb{E}[\mathbb{V}_h[Q \mid S, \mathcal{R}]]}_{\mathrm{GL}_{residual}(S)} \tag{8}$$

*Moreover if the scoring rule is proper, then:* $\qquad \mathrm{GL}(S) \geq \mathrm{GL}_{explained}(S) \geq 0.$ \hfill (9)

*Proof of Theorem 4.1.* Applying Lemma C.1 with $(\mathcal{R}, Q, S)$ as $(X, Y, Z)$ gives the decomposition. Proper scoring rules have a convex negative entropy $h$ (see Gneiting & Raftery, 2007, th. 1). Note that depending on the convention (maximization or minimization of scoring rules), one may find in the litterature that the entropy is either convex or concave. In the convention taken by this article (minimization of scoring rules), the entropy is concave and the negative entropy is convex. Using Jensen's inequality, we thus have $\mathbb{V}_h[Q \,|\, S, \mathcal{R}] \geq 0$. Hence both $\mathrm{GL}_{explained}$ and $\mathrm{GL}_{residual}$ are positive, which gives $\mathrm{GL} \geq \mathrm{GL}_{explained}$. $\qquad\square$

## C.3    PROPOSITION 4.1: BINNING-INDUCED GROUPING LOSS

**Proposition 4.1** (Binning-induced grouping loss). *The grouping loss of the binned classifier* $\mathrm{GL}(S_B)$ *deviates from that of the original classifier* $\mathrm{GL}(S)$ *by an induced grouping loss* $\mathrm{GL}_{induced}(S, S_B)$:

$$\underbrace{\mathbb{E}[\mathbb{V}_h[Q \,|\, S_B]]}_{\mathrm{GL}(S_B)} = \underbrace{\mathbb{E}[\mathbb{V}_h[Q \,|\, S]]}_{\mathrm{GL}(S)} + \underbrace{\mathbb{E}[\mathbb{V}_h[C \,|\, S_B]]}_{\mathrm{GL}_{induced}(S, S_B)} \qquad (11)$$

*Moreover, if the scoring rule is proper:* $\qquad \mathrm{GL}_{induced}(S, S_B) \geq 0.$

*Proof of Proposition 4.1.*

$$
\begin{aligned}
\mathbb{V}_h[Q \,|\, S_B] &= \mathbb{E}[\mathbb{V}_h[Q \,|\, S, S_B] \,|\, S_B] + \mathbb{V}_h[\mathbb{E}[Q \,|\, S, S_B] \,|\, S_B] && \text{Law of total } h\text{-variance (Lemma C.1)}\\
&= \mathbb{E}[\mathbb{V}_h[Q \,|\, S] \,|\, S_B] + \mathbb{V}_h[\mathbb{E}[Q \,|\, S] \,|\, S_B] && S_B \text{ is a function of } S\\
&= \mathbb{E}[\mathbb{V}_h[Q \,|\, S] \,|\, S_B] + \mathbb{V}_h[C \,|\, S_B] && C = \mathbb{E}[Q \,|\, S]
\end{aligned}
$$

$$
\begin{aligned}
\mathbb{E}[\mathbb{V}_h[Q \,|\, S_B]] &= \mathbb{E}[\mathbb{V}_h[Q \,|\, S]] + \mathbb{E}[\mathbb{V}_h[C \,|\, S_B]] && \text{Law of total expectation}\\
\mathrm{GL}(S_B) &= \mathrm{GL}(S) + \mathrm{GL}_{induced}(S, S_B) && \text{Lemma 4.1 and definition of } \mathrm{GL}_{induced}
\end{aligned}
$$

Remark: this proposition does not require $S_B$ to be the average scores on the bins $\mathbb{E}[S \,|\, S \in \mathcal{B}_j]$. $\quad\square$

## C.4    PROPOSITION 4.2: EXPLAINED GROUPING LOSS ACCOUNTING FOR BINNING

**Proposition 4.2** (Explained grouping loss accounting for binning).

$$\mathrm{GL}(S) = \mathrm{GL}_{explained}(S_B) - \mathrm{GL}_{induced}(S, S_B) + \mathrm{GL}_{residual}(S_B) \qquad (12)$$

*If the scoring rule is proper, then:* $\qquad \mathrm{GL}(S) \geq \underbrace{\mathrm{GL}_{explained}(S_B) - \mathrm{GL}_{induced}(S, S_B)}_{\mathrm{GL}_{\mathrm{LB}}(S, S_B)}. \qquad (13)$

*Proof of Proposition 4.2.*

$$
\begin{aligned}
\mathrm{GL}(S) &= \mathrm{GL}(S_B) - \mathrm{GL}_{induced}(S, S_B) && \text{Propostion 4.1}\\
&= \mathrm{GL}_{explained}(S_B) + \mathrm{GL}_{residual}(S_B) - \mathrm{GL}_{induced}(S, S_B) && \text{Theorem 4.1 on } \mathrm{GL}(S_B)
\end{aligned}
$$

For proper scoring rules, Theorem 4.1 gives $\mathrm{GL}_{residual}(S_B) \geq 0$ which completes the proof. $\quad\square$

## C.5    PROPOSITION 4.3: DEBIASED ESTIMATOR FOR THE BRIER SCORE

**Proposition 4.3** (Debiased estimator for the Brier score). *For all class* $k \in \{1, \dots, K\}$ *and bin* $s \in \mathcal{S}$, *let* $n^{(s,k)}$ *(resp.* $n_j^{(s,k)}$*) be the number of samples belonging to level set* $\mathcal{R}^{(s)}$ *(resp. region* $\mathcal{R}_j^{(s)}$*). We define the empirical average of* $Y$ *over these regions as:*

$$\hat{\mu}_j^{(s,k)} := \frac{1}{n_j^{(s,k)}} \sum_{i: X^{(i)} \in \mathcal{R}_j^{(s)}} Y_k^{(i)} \quad and \quad \hat{c}^{(s,k)} = \frac{1}{n^{(s,k)}} \sum_{i: X^{(i)} \in \mathcal{R}^{(s)}} Y_k^{(i)}$$

*The debiased estimator of* $\mathrm{GL}_{explained}$ *is:* $\quad \widehat{\mathrm{GL}}_{explained}(S_B) = \sum_{k=1}^{K} \sum_{s \in \mathcal{S}} \frac{n^{(s,k)}}{n} \widehat{\mathrm{GL}}_{explained}^{(s,k)}(S_B)$

*with:*

$$\widehat{\mathrm{GL}}_{explained}^{(s,k)}(S_B) = \underbrace{\sum_{j=1}^{J} \frac{n_j^{(s,k)}}{n^{(s,k)}} \left( \hat{\mu}_j^{(s,k)} - \hat{c}^{(s,k)} \right)^2}_{\text{plugin estimator } \widehat{\mathrm{GL}}_{plugin}} - \underbrace{\left( \sum_{j=1}^{J} \frac{n_j^{(s,k)}}{n^{(s,k)}} \frac{\hat{\mu}_j^{(s,k)}(1-\hat{\mu}_j^{(s,k)})}{n_j^{(s,k)}-1} - \frac{\hat{c}^{(s,k)}(1-\hat{c}^{(s,k)})}{n^{(s,k)}-1} \right)}_{\text{bias estimation } \widehat{\mathrm{GL}}_{bias}}$$

*Proof.* Let $s \in \mathcal{S}^{(k)}$ and $k \in \{1, K\}$, and define $\hat{p}_j^{(s,k)} := \frac{n_j^{(s,k)}}{n^{(s,k)}}$. We now compute the bias of the plugin estimator for $\widehat{\mathrm{GL}}_{explained}^{(s,k)}$. To ease calculations, we start by rewriting the plugin estimate:

$$\widehat{\mathrm{GL}}_{plugin}^{(s,k)} = \sum_{j=1}^{J} \hat{p}_j^{(s,k)} \left( \hat{\mu}_j^{(s,k)} - \hat{c}^{(s,k)} \right)^2 \tag{28}$$

$$= \sum_{j=1}^{J} \hat{p}_j^{(s,k)} \left( \hat{\mu}_j^{(s,k)} \right)^2 - 2\hat{c}^{(s,k)} \left( \sum_{j=1}^{J} \hat{p}_j^{(s,k)} \hat{\mu}_j^{(s,k)} \right) + \left( \hat{c}^{(s,k)} \right)^2 \tag{29}$$

$$= \sum_{j=1}^{J} \hat{p}_j^{(s,k)} \left( \hat{\mu}_j^{(s,k)} \right)^2 - \left( \hat{c}^{(s,k)} \right)^2 \tag{30}$$

From now on, we omit the exponent $(s, k)$ to lighten notations. We now take the expectation of both terms in the lower-bound.

$$\mathbb{E}\left[ \hat{c}^2 \right] = \mathbb{E}\left[ \hat{c} \right]^2 + \mathrm{Var}(\hat{c}) \tag{31}$$

$$= c^2 + \frac{c(1-c)}{n} \tag{32}$$

where we made use of Lemma C.2 for equation 32. Similarly,

$$\mathbb{E}\left[ \hat{\mu}_j^2 \mid \hat{p}_j \right] = \mathbb{E}\left[ \hat{\mu}_j \right]^2 + \mathrm{Var}(\hat{\mu}_j) \tag{33}$$

$$= \mu_j^2 + \frac{\mu_j(1-\mu_j)}{n_j} \tag{34}$$

When $n_j = 0$ (or equivalently $\hat{p}_j = 0$), which happens with probability $\nu_j = (1-p_j)^n$, $\hat{\mu}_j$ as well as the right term in equation 34 are undefined. The problem disappears when multiplying by $\hat{p}_j$, and agreeing that $\hat{\mu}_j = 0$ whenever $n_j = 0$.

$$\mathbb{E}\left[ \sum_{j=1}^{J} \hat{p}_j \hat{\mu}_j^2 \right] = \sum_{j=1}^{J} \mathbb{E}\left[ \mathbb{E}\left[ \hat{p}_j \hat{\mu}_j^2 \mathbb{1}_{\hat{p}_j \geq 0} \mid \hat{p}_j \right] \right] \tag{35}$$

$$= \sum_{j=1}^{J} \mathbb{E}\left[ \hat{p}_j \mathbb{1}_{\hat{p}_j \geq 0} \left( \mu_j^2 + \frac{\mu_j(1-\mu_j)}{n_j} \right) \right] \tag{36}$$

$$= \sum_{j=1}^{J} \left( p_j \mu_j^2 + \mathbb{E}\left[ \mathbb{1}_{\hat{p}_j \geq 0} \frac{\mu_j(1-\mu_j)}{n} \right] \right) \tag{37}$$

$$= \sum_{j=1}^{J} \left( p_j \mu_j^2 + (1-\nu_j) \frac{\mu_j(1-\mu_j)}{n} \right) \tag{38}$$

Putting together equations 32 and 38, we get:

$$\mathbb{E}\left[\widehat{\mathrm{GL}}_{plugin}\right] = \sum_{j=1}^{J} p_j \mu_j^2 - c^2 + \sum_{j=1}^{J}(1-\nu_j)\frac{\mu_j(1-\mu_j)}{n} - \frac{c(1-c)}{n} \tag{39}$$

$$= \underbrace{\sum_{j=1}^{J} p_j(\mu_j - c)^2}_{\mathrm{GL}_{explained}} + \underbrace{\sum_{j=1}^{J}(1-\nu_j)\frac{\mu_j(1-\mu_j)}{n} - \frac{c(1-c)}{n}}_{\mathrm{GL}_{bias}} \tag{40}$$

In practice $\nu_j$, which gives the probability that no sample falls in component $j$, is very close to 0 unless $p_j$ and $n$ are very small. Hence, we will approximate $\nu_j \approx 0$. More importantly, the expression of the bias given in 40 depends on oracle quantities $\mu_j$ and $c$, which are unavailable. Therefore, we resort to debiasing the plugin estimate of the lower-bound using sample estimates of the bias, which gives:

$$\widehat{\mathrm{GL}}_{explained}^{(s,k)} = \underbrace{\sum_{j=1}^{J}\frac{n_j^{(s,k)}}{n^{(s,k)}}\left(\hat{\mu}_j^{(s,k)} - \hat{c}^{(s,k)}\right)^2 - \sum_{j=1}^{J}\frac{n_j^{(s,k)}}{n^{(s,k)}}\frac{\hat{\mu}_j^{(s,k)}(1-\hat{\mu}_j^{(s,k)})}{n_j^{(s,k)}-1}}_{\text{plugin estimator } \widehat{\mathrm{GL}}_{plugin}} + \frac{\hat{c}^{(s,k)}(1-\hat{c}^{(s,k)})}{n^{(s,k)}-1}$$

$$\tag{41}$$

where we used a Bessel correction for the estimation of population variances. Finally, a debiased estimator of $\mathrm{GL}_{explained}$ is obtained by summing over the debiased estimators for all $k \in \{1, K\}$ and all $s \in \mathcal{S}_k$. □

**Lemma C.2.** *Define $\hat{\mu}_j^{(s,k)}$ and $\hat{c}^{(s,k)}$ as in Proposition 4.3. Then:*

$$\mathbb{E}\left[\hat{\mu}_j^{(s,k)}\right] = \mu_j^{(s,k)} \quad and \quad Var\left(\hat{\mu}_j^{(s,k)}\right) = \frac{\mu_j^{(s,k)}\left(1-\mu_j^{(s,k)}\right)}{n_j^{(s,k)}}. \tag{42}$$

*Similarly,*

$$\mathbb{E}\left[\hat{c}^{(s,k)}\right] = c^{(s,k)} \quad and \quad Var\left(\hat{c}^{(s,k)}\right) = \frac{c^{(s,k)}\left(1-c^{(s,k)}\right)}{n^{(s,k)}}. \tag{43}$$

The labels $Y_k^{(i)}$ are by definition drawn from a Bernoulli distribution with probability $P(Y_k^{(i)}|X^{(i)}) = Q_k^{(i)}$, *i.e.*, for each sample $i$, the probability of the Bernoulli changes.

This lemma shows that despite these varying Bernoulli probabilities, the empirical average of labels $Y_k$ over a given subspace has the same expectation and variance as a binomial variable that would be drawn with a probability equal to the expectation of $Q_k$ over this subspace.

*Proof of Lemma C.2.* Below we write the proof for the case of $\hat{\mu}_j^{(s,k)}$ (equation 42) as the one for $\hat{c}^{(s,k)}$ (equation 43) follows exactly the same lines. Let $\mathcal{I}_j^{(s)} = \left\{i : X^{(i)} \in \mathcal{R}_j^{(s)}\right\}$, be the subset of samples such that $X^{(i)}$ belongs to bin $\mathcal{R}_j^{(s)}$.

$$\mathbb{E}\left[\hat{\mu}_j^{(s,k)}\right] = \frac{1}{n_j^{(s,k)}} \sum_{i \in \mathcal{I}_j^{(s)}} \mathbb{E}\left[Y_k^{(i)} \mid S_k = s, \mathcal{R}(X^{(i)}) = j\right] \tag{44}$$

$$= \frac{1}{n_j^{(s,k)}} \sum_{i \in \mathcal{I}_j^{(s)}} \mathbb{E}\left[\mathbb{E}\left[Y_k^{(i)} \mid X^{(i)}\right] \mid S_k = s, \mathcal{R}(X^{(i)}) = j\right] \tag{45}$$

$$= \frac{1}{n_j^{(s,k)}} \sum_{i \in \mathcal{I}_j^{(s)}} \mathbb{E}\left[Q_k^{(i)} \mid S_k = s, \mathcal{R}(X^{(i)}) = j\right] \tag{46}$$

$$= \frac{1}{n_j^{(s,k)}} \sum_{i \in \mathcal{I}_j^{(s)}} \mu_j^{(s,k)} \tag{47}$$

$$= \mu_j^{(s,k)} \tag{48}$$

where we used the law of total expectation in eq 45, the definition of $Q_k$ in eq 46, and the definition of $\mu_j^{(s,k)}$ in eq 47.

$$\mathrm{Var}\left(\hat{\mu}_j^{(s,k)}\right) = \mathbb{E}\left[(\hat{\mu}_j^{(s,k)} - \mu_j^{(s,k)})^2 \mid S_k = s, \mathcal{R}(X^{(i)}) = j\right] \tag{49}$$

$$= \mathbb{E}\left[\left(\hat{\mu}_j^{(s,k)}\right)^2 \mid S_k = s, \mathcal{R}(X^{(i)}) = j\right] - \left(\mu_j^{(s,k)}\right)^2 \tag{50}$$

$$= \frac{1}{\left(n_j^{(s,k)}\right)^2} \mathbb{E}\left[\sum_{i \in \mathcal{I}_j^{(s)}} Y^{(i)} \sum_{l \in \mathcal{I}_j^{(s)}} Y_k^{(l)} \mid S_k = s, \mathcal{R}(X^{(i)}) = j\right] - \left(\mu_j^{(s,k)}\right)^2 \tag{51}$$

$$= \frac{1}{\left(n_j^{(s,k)}\right)^2} \mathbb{E}\left[\sum_{i \in \mathcal{I}_j^{(s)}} Y_k^{(i)} + \sum_{\substack{i \neq l \\ i,l \in \mathcal{I}_j^{(s)}}} Y_k^{(i)} Y_k^{(l)} \mid S_k = s, \mathcal{R}(X^{(i)}) = j\right] - \left(\mu_j^{(s,k)}\right)^2 \tag{52}$$

$$= \frac{1}{\left(n_j^{(s,k)}\right)^2} \left(\sum_{i \in \mathcal{I}_j^{(s)}} \mu_j^{(s,k)} + \sum_{\substack{i \neq l \\ i,l \in \mathcal{I}_j^{(s)}}} \left(\mu_j^{(s,k)}\right)^2\right) - \left(\mu_j^{(s,k)}\right)^2 \tag{53}$$

$$= \frac{1}{\left(n_j^{(s,k)}\right)^2} \left(n_j^{(s,k)} \mu_j^{(s,k)} + n_j^{(s,k)}(n_j^{(s,k)} - 1)\left(\mu_j^{(s,k)}\right)^2\right) - \left(\mu_j^{(s,k)}\right)^2 \tag{54}$$

$$= \frac{\mu_j^{(s,k)}(1 - \mu_j^{(s,k)})}{n_j^{(s,k)}} \tag{55}$$

where we used the fact that $Y_k^{(i)}$ and $Y_k^{(l)}$ are independent when $i \neq l$ in eq 52. □

## C.6 THE PLUGIN ESTIMATOR FOR THE GROUPING LOSS LOWER BOUND IS BIASED UPWARDS.

**Analytical evaluation of the sign of the bias** Let $k \in \{1, \dots, K\}$ and $s \in \mathcal{S}$. The bias of the plugin estimate $\widehat{\mathrm{GL}}_{explained}^{(s,k)}(S_B)$ is given by (40):

$$\mathrm{bias}\left(\widehat{\mathrm{GL}}_{explained}^{(s,k)}(S_B)\right) = \sum_{j=1}^{J} \left(1 - \nu_j^{(s,k)}\right) \frac{\mu_j^{(s,k)}(1 - \mu_j^{(s,k)})}{n^{(s,k)}} - \frac{c^{(s,k)}(1 - c^{(s,k)})}{n^{(s,k)}} \tag{56}$$

By convexity of the function $x \mapsto (x - \mathbb{E}[x])^2$, we have:

$$\left( \sum_{j=1}^{J} \frac{n_j^{(s,k)}}{n^{(s,k)}} \hat{\mu}_j^{(s,k)} - \mathbb{E}\left[ \sum_{j=1}^{J} \frac{n_j^{(s,k)}}{n^{(s,k)}} \hat{\mu}_j^{(s,k)} \right] \right)^2 \leq \sum_{j=1}^{J} \frac{n_j^{(s,k)}}{n^{(s,k)}} \left( \hat{\mu}_j^{(s,k)} - \mathbb{E}\left[ \hat{\mu}_j^{(s,k)} \right] \right)^2 \qquad (57)$$

Using the fact that $\hat{c}^{(s,k)} = \sum_{j=1}^{J} \frac{n_j^{(s,k)}}{n^{(s,k)}} \hat{\mu}_j^{(s,k)}$, and taking the expectation of both sides, we get:

$$\mathrm{Var}(\hat{c}^{(s,k)}) \leq \sum_{j=1}^{J} \frac{n_j^{(s,k)}}{n^{(s,k)}} \mathrm{Var}(\hat{\mu}_j^{(s,k)}) \qquad (58)$$

Finally, using Lemma C.2, we get:

$$\frac{c^{(s,k)} \left( 1 - c^{(s,k)} \right)}{n^{(s,k)}} \leq \sum_{j=1}^{J} \frac{\mu_j^{(s,k)} \left( 1 - \mu_j^{(s,k)} \right)}{n^{(s,k)}} \qquad (59)$$

Hence, we have:

$$\mathrm{bias}\left( \widehat{\mathcal{L}}_{GL}^{(s,k)} \right) = \underbrace{\sum_{j=1}^{J} \frac{\mu_j^{(s,k)}(1 - \mu_j^{(s,k)})}{n^{(s,k)}} - \frac{c^{(s,k)}(1 - c^{(s,k)})}{n^{(s,k)}}}_{\geq 0} - \sum_{j=1}^{J} \nu_j^{(s,k)} \frac{\mu_j^{(s,k)}(1 - \mu_j^{(s,k)})}{n^{(s,k)}} \qquad (60)$$

Because of the term involving $\nu_j^{(s,k)}$, this inequality does not prove that the bias is always positive. However in practice $\nu_j^{(s,k)} = \left( 1 - p_j^{(s,k)} \right)^{n^{(s,k)}}$, which represents the probability that no point belongs to region $j$, is very close to 0 unless $p_j^{(s,k)}$ is very small or the total number of points $n^{(s,k)}$ is small. Hence, equality 60 shows that the bias can only be 'slightly' negative. In the simulations below, the upwards bias of the plugin estimate appears clearly.

## C.7 ESTIMATOR FOR THE INDUCED GROUPING LOSS

**Proposition C.1** (Estimator for the induced grouping loss). *Let $\hat{C}$ be an estimator of $C$. An estimator of $C_B$ is $\hat{C}_B(s) = \frac{1}{n^{(s)}} \sum_{i:S_B(X^{(i)})=s} \hat{C}(S(X^{(i)}))$ with $n^{(s)}$ the number of sample in the level set $s$. An estimator of the grouping loss induced by the binning of $S$ into $S_B$ is:*

$$\widehat{\mathrm{GL}}_{induced}(S, S_B) = \sum_{s \in \mathcal{S}} \frac{n^{(s)}}{n} \left[ \frac{1}{n^{(s)}} \sum_{i:S_B(X^{(i)})=s} e(\hat{C}(S(X^{(i)}))) - e(\hat{C}_B(s)) \right] \qquad (61)$$

## C.8 ANALYSIS OF BINNING-INDUCED ERRORS FOR THE BRIER SCORE

It is well known that binning can induce error in estimating calibration loss, leading to underestimating it (Bröcker, 2012; Kumar et al., 2019; Roelofs et al., 2022). Proposition 4.1 shows that it also leads to errors on the grouping loss, overestimating it. Here we characterize the errors on the calibration and grouping loss for the Brier score and show that they partly compensate each other and the error on the sum of both can be bounded.

Proposition C.2 gives the deviation term induced by the binning for the calibration loss with the Brier scoring rule.

**Proposition C.2** (Calibration loss decomposition). *Let $h$ be the negative entropy of the Brier scoring rule and $C = \mathbb{E}[Q \mid S]$. The binned calibration loss $\mathrm{CL}(S_B)$ deviates from the calibration loss $\mathrm{CL}(S)$ by a negative induced calibration loss $\mathrm{CL}_{induced}(S, S_B)$:*

$$\underbrace{\mathbb{E}\left[\|S_B - C_B\|^2\right]}_{\mathrm{CL}(S_B)} = \underbrace{\mathbb{E}\left[\|S - C\|^2\right]}_{\mathrm{CL}(S)} - \underbrace{\mathbb{E}\left[\mathbb{V}_h\left[S - C \mid S_B\right]\right]}_{\mathrm{CL}_{induced}(S, S_B)} \qquad (62)$$

The calibration loss induced by the binning, $\mathrm{CL}_{induced}(S, S_B)$, is always negative. $\mathrm{CL}(S_B)$ is thus biased downward, which is already known from Kumar et al. (2019); Roelofs et al. (2022). Conversely, the grouping loss induced by the binning, $\mathrm{GL}_{induced}(S, S_B)$, is always positive. $\mathrm{GL}(S_B)$ is thus biased upward. The mere effect of binning artificially creates grouping loss and artificially reduces calibration error. For calibrated continuous classifiers, $\mathrm{CL}_{induced} = 0$ and induced grouping loss is small: with $N$ equal-width bins, $\mathrm{GL}_{induced} \leq \frac{1}{4N^2}$. If in addition the scores are uniform on the bins: $\mathrm{GL}_{induced} = \frac{1}{12N^2}$ (Lemma C.3). Both induced calibration and grouping losses can be large since $\mathbb{V}[C \mid S_B]$ can be large. High $\mathrm{GL}_{induced}$ expresses strong miscalibrations within the bin. However interestingly, both induced losses compensate. In a binary setting, the sum of induced calibration and grouping losses is contained as showed by Theorem C.1, and can be bounded by estimable quantities (Corollary C.1). While measuring $\mathrm{CL}(S_B)$ and $\mathrm{GL}(S_B)$ separately can lead to high binning-induced bias, measuring $\mathrm{CL}(S_B) + \mathrm{GL}(S_B)$ through $\widehat{\mathrm{CL}}(S_B) + \widehat{\mathrm{GL}}_{explained}(S_B)$ enables reducing binning-induced errors and minorizing $\mathrm{MSE(S, Q)}$ (Corollary C.2).

**Theorem C.1** (Bounds on induced calibration and grouping losses). *In a binary setting, the calibration and grouping losses induced by the binning of classifier $S$ into $S_B$ sums to:*

$$\mathrm{CL}_{induced} + \mathrm{GL}_{induced} = \mathbb{E}[2\mathrm{Cov}[S, C \mid S_B] - \mathbb{V}[S \mid S_B]]$$

*which is bounded by:*

$$-\mathbb{E}\left[\sqrt{\mathbb{V}[S \mid S_B]}\left(2\sqrt{\mathbb{V}[C \mid S_B]} + \sqrt{\mathbb{V}[S \mid S_B]}\right)\right] \leq \mathrm{CL}_{induced} + \mathrm{GL}_{induced}$$
$$\leq \mathbb{E}\left[\sqrt{\mathbb{V}[S \mid S_B]}\left(2\sqrt{\mathbb{V}[C \mid S_B]} - \sqrt{\mathbb{V}[S \mid S_B]}\right)\right]$$

*Suppose that $[0, 1]$ is divided in $N$ equal-width bins. Then:*

$$-\frac{1}{N}\mathbb{E}\left[\sqrt{C_B(1 - C_B)}\right] - \frac{1}{4N^2} \leq \mathrm{CL}_{induced} + \mathrm{GL}_{induced} \leq \frac{1}{N}\mathbb{E}\left[\sqrt{C_B(1 - C_B)}\right]$$

**Corollary C.1.**

$$-\mathbb{E}\left[\sqrt{\mathbb{V}[S \mid S_B]}\left(2\sqrt{C_B(1 - C_B)} + \sqrt{\mathbb{V}[S \mid S_B]}\right)\right] \leq \mathrm{CL}_{induced} + \mathrm{GL}_{induced}$$
$$\leq \mathbb{E}\left[\sqrt{\mathbb{V}[S \mid S_B]}\left(2\sqrt{C_B(1 - C_B)} - \sqrt{\mathbb{V}[S \mid S_B]}\right)\right]$$

*With $N$ equal-width bins:*

$$-\frac{1}{N}\mathbb{E}\left[\sqrt{C_B(1 - C_B)}\right] - \frac{1}{4N^2} \leq \mathrm{CL}_{induced} + \mathrm{GL}_{induced} \leq \frac{1}{N}\mathbb{E}\left[\sqrt{C_B(1 - C_B)}\right]$$

**Corollary C.2.** *The mean square error (MSE) between continuous $S$ and $Q$ is lower bounded by:*

$$\mathrm{MSE}(S, Q) = \mathrm{CL} + \mathrm{GL}$$
$$\geq \ell^2\text{-}\mathrm{ECE}_B + \mathcal{L}_{\mathrm{GL}_B} - \mathbb{E}\left[\sqrt{\mathbb{V}[S \mid S_B]}\left(2\sqrt{\mathbb{V}[C \mid S_B]} - \sqrt{\mathbb{V}[S \mid S_B]}\right)\right]$$
$$\geq \ell^2\text{-}\mathrm{ECE}_B + \mathcal{L}_{\mathrm{GL}_B} - \mathbb{E}\left[\sqrt{\mathbb{V}[S \mid S_B]}\left(2\sqrt{C_B(1 - C_B)} - \sqrt{\mathbb{V}[S \mid S_B]}\right)\right]$$

*With $N$ equal bins:* $\geq \ell^2\text{-}\mathrm{ECE}_B + \mathcal{L}_{\mathrm{GL}_B} - \frac{1}{N}\mathbb{E}\left[\sqrt{C_B(1 - C_B)}\right]$

*where $\ell^2\text{-}\mathrm{ECE}_B$ is the $\ell^2$ Expected Calibration Error of the binned classifier $S_B$ and $\mathcal{L}_{\mathrm{GL}_B}$ is the grouping loss lower bound of $S_B$.*

PROOFS

*Proof of Proposition C.2.* Let $h$ the negative entropy of the Brier scoring rule.

$$\|S_B - C_B\|^2 = \|\mathbb{E}[S \mid S_B] - \mathbb{E}[C \mid S_B]\|^2 \qquad S_B = \mathbb{E}[S \mid S_B], C_B = \mathbb{E}[C \mid S_B]$$
$$= \|\mathbb{E}[S - C \mid S_B]\|^2 \qquad \text{Linearity of expectation}$$
$$= \mathbb{E}\left[\|S - C\|^2 \mid S_B\right] - \mathbb{V}_h[S - C \mid S_B] \qquad \text{Definition of } \mathbb{V}_h[S - C \mid S_B]$$

$$\mathbb{E}\left[(S_B - C_B)^2\right] = \mathbb{E}\left[\|S - C\|^2\right] - \mathbb{E}[\mathbb{V}_h[S - C \mid S_B]] \qquad \text{Law of total expectation}$$

$\square$

**Lemma C.3.** *In a binary setting, suppose that $[0, 1]$ is divided in $N$ equal-width bins. Then:*

$$\mathbb{V}[S \,|\, S_B] \leq \tfrac{1}{4N^2} \tag{63}$$

*If in addition, scores $S$ are uniform:*

$$\mathbb{V}[S \,|\, S_B] = \tfrac{1}{12N^2} \tag{64}$$

*Proof of Lemma C.3.* Without loss of generality, consider the first bin $[0, \tfrac{1}{N}]$ with binned score $s_1$.

$$
\begin{aligned}
\mathbb{V}[S \,|\, S_B = s_1] &= \mathbb{E}\big[S^2 \,\big|\, S_B = s_1\big] - \mathbb{E}[S \,|\, S_B = s_1]^2 && \text{Definition of the variance} \\
&\leq \tfrac{1}{N}\mathbb{E}[S \,|\, S_B = s_1] - \mathbb{E}[S \,|\, S_B = s_1]^2 && 0 \leq S \leq \tfrac{1}{N} \Rightarrow S^2 \leq \tfrac{1}{N}S \\
&= \tfrac{1}{N^2}(1 - N\mathbb{E}[S \,|\, S_B = s_1])N\mathbb{E}[S \,|\, S_B = s_1] \\
&\leq \tfrac{1}{4N^2} && \text{Max when } N\mathbb{E}[S \,|\, S_B = s_1] = \tfrac{1}{2}
\end{aligned}
$$

For uniform scores: $S|S_B = s_1 \sim \mathcal{U}([0, \tfrac{1}{N}])$. Hence $\mathbb{V}[S \,|\, S_B = s_1] = \tfrac{1}{12}(\tfrac{1}{N} - 0)^2 = \tfrac{1}{12N^2}$.

Other bins have same variance as $\mathbb{V}[S \,|\, S_B = s_1]$ (variance is translation-invariant).
Remark: this proves that $\mathrm{GL}_{induced} \leq \tfrac{1}{4N^2}$ for $S$ calibrated ($S = C \Rightarrow \mathbb{V}[C \,|\, S_B] = \mathbb{V}[S \,|\, S_B]$).  $\square$

*Proof of Theorem C.1.* In a binary setting for the Brier scoring rule, we have $\mathbb{V}_h = \mathbb{V}$. Hence:

$$
\begin{aligned}
\mathrm{CL}_{induced} + \mathrm{GL}_{induced} &= -\mathbb{E}[\mathbb{V}_h[S - C \,|\, S_B]] + \mathbb{E}[\mathbb{V}_h[C \,|\, S_B]] && \text{Propositons 4.1 and C.2} \\
&= -\mathbb{E}[\mathbb{V}[S - C \,|\, S_B]] + \mathbb{E}[\mathbb{V}[C \,|\, S_B]] && \mathbb{V}_h = \mathbb{V} \\
&= \mathbb{E}[2\mathrm{Cov}[S, C \,|\, S_B] - \mathbb{V}[S \,|\, S_B]] && \text{Expansion of } \mathbb{V}[S - C \,|\, S_B]
\end{aligned}
$$

$$2\mathrm{Cov}[S, C \,|\, S_B] - \mathbb{V}[S \,|\, S_B] \leq 2\sqrt{\mathbb{V}[S \,|\, S_B]}\sqrt{\mathbb{V}[C \,|\, S_B]} - \mathbb{V}[S \,|\, S_B] \qquad \text{Cauchy-Schwarz}$$

$$
\begin{aligned}
2\mathrm{Cov}[S, C \,|\, S_B] - \mathbb{V}[S \,|\, S_B] &\geq -2\,|\,\mathrm{Cov}[S, C \,|\, S_B]\,| - \mathbb{V}[S \,|\, S_B] \\
&\geq -2\sqrt{\mathbb{V}[S \,|\, S_B]}\sqrt{\mathbb{V}[C \,|\, S_B]} - \mathbb{V}[S \,|\, S_B] && \text{Cauchy-Schwarz}
\end{aligned}
$$

With $N$ equal-width bins:

$$
\begin{aligned}
2\mathrm{Cov}[S, C \,|\, S_B] - \mathbb{V}[S \,|\, S_B] &\leq 2\sqrt{\mathbb{V}[S \,|\, S_B]}\sqrt{\mathbb{V}[C \,|\, S_B]} && \text{Positivity of the variance} \\
&\leq \tfrac{1}{N}\sqrt{\mathbb{V}[C \,|\, S_B]} && \mathbb{V}[S \,|\, S_B] \leq \tfrac{1}{4N^2} \\
&\leq \tfrac{1}{N}\sqrt{C_B(1 - C_B)} && \mathbb{V}[C \,|\, S_B] \leq C_B(1 - C_B)
\end{aligned}
$$

$$
\begin{aligned}
2\mathrm{Cov}[S, C \,|\, S_B] - \mathbb{V}[S \,|\, S_B] &\geq -\tfrac{1}{N}\sqrt{\mathbb{V}[C \,|\, S_B]} - \tfrac{1}{4N^2} && \mathbb{V}[S \,|\, S_B] \leq \tfrac{1}{4N^2} \\
&\geq -\tfrac{1}{N}\sqrt{C_B(1 - C_B)} - \tfrac{1}{4N^2} && \mathbb{V}[C \,|\, S_B] \leq C_B(1 - C_B)
\end{aligned}
$$

$\square$

## C.9 EXTENSION TO CLASSWISE CALIBRATION

### C.9.1 PROPER SCORING RULES DECOMPOSITION

We show below that the proper scoring rules decomposition of Kull & Flach (2015) holds for classwise-calibration (Definition 3.2) for the Brier score and the log-loss.

**Proposition C.3** (Brier and log-loss classwise decomposition). *For the Brier score as well as the log-loss, the decomposition into calibration, grouping, and irreducible losses (Equation 6) holds when replacing the calibrated scores by the classwise-calibrated scores (Definition 3.2).*

*Proof of Proposition C.3.* For all $k \in \{1, \ldots, K\}$, let $C_k = \mathbb{E}[Y_k | S_k]$ be the classwise-calibrated scores (Definition 3.2).

**Brier Score**   Given any two probability vectors $P$ and $Q$, the divergence associated to the Brier score reads:

$$d(P, Q) = \sum_{k=1}^{K} (P_k - Q_k)^2 \tag{65}$$

For all $k \in \{1, \ldots, K\}$, let $d_k : P_k, Q_k \mapsto (P_k - Q_k)^2$.

$$d_k(S_k, Y_k) = (S_k - Y_k)^2 \tag{66}$$

$$= (S_k - C_k + C_k - Q_k + Q_k - Y_k)^2 \tag{67}$$

$$\begin{aligned} &= (S_k - C_k)^2 + (C_k - Q_k)^2 + (Q_k - Y_k)^2 + 2(S_k - C_k)(C_k - Q_k) \\ &\quad + 2(S_k - C_k)(Q_k - Y_k) + 2(C_k - Q_k)(Q_k - Y_k) \end{aligned} \tag{68}$$

Taking the expectation on both sides conditional on $X$:

$$\begin{aligned} \mathbb{E}\left[d_k(S_k, Y_k) \mid X\right] &= (S_k - C_k)^2 + (C_k - Q_k)^2 + \mathbb{E}\left[(Q_k - Y_k)^2 \mid X\right] \\ &\quad + 2(S_k - C_k)(C_k - Q_k) \end{aligned} \tag{69}$$

since $S_k$ and $Q_k$ are function of $X$, $C_k$ is a function of $S_k$ and thus of $X$, and $\mathbb{E}\left[Y_k \mid X\right] = Q_k$. Then taking the expectation conditional on $S_k$:

$$\mathbb{E}\left[d_k(S_k, Y_k) \mid S_k\right] = (S_k - C_k)^2 + \mathbb{E}\left[(C_k - Q_k)^2 \mid S_k\right] + \mathbb{E}\left[(Q_k - Y_k)^2 \mid S_k\right] \tag{70}$$

where we use the fact that $C_k$ is a function of $S_k$, that $\mathbb{E}\left[Q_k \mid S_k\right] = C_k$, and the property according to which for two random variables $U$ and $V$ and a function $h$, $\mathbb{E}\left[\mathbb{E}\left[V \mid U\right] \mid h(U)\right] = \mathbb{E}\left[V \mid h(U)\right]$. Finally, taking the expectation over $S_k$ we get:

$$\mathbb{E}\left[d_k(S_k, Y_k)\right] = \mathbb{E}\left[(S_k - C_k)^2\right] + \mathbb{E}\left[(C_k - Q_k)^2\right] + \mathbb{E}\left[(Q_k - Y_k)^2\right] \tag{71}$$

The desired decomposition is then obtained by summing over the K classes on both sides.

**log-loss**   Given any two probability vectors $P$ and $Q$, the divergence associated to the log loss reads:

$$d(P, Q) = \sum_{k=1}^{K} Q_k \log\left(\frac{Q_k}{P_k}\right) \tag{72}$$

For all $k \in \{1, \ldots, K\}$, let $d_k : P_k, Q_k \mapsto Q_k \log\left(\frac{Q_k}{P_k}\right)$.

$$d_k(S_k, Y_k) = Y_k \log\left(\frac{Y_k}{S_k}\right) \tag{73}$$

$$= Y_k \log\left(\frac{Y_k}{Q_k}\right) + Y_k \log\left(\frac{Q_k}{C_k}\right) + Y_k \log\left(\frac{C_k}{S_k}\right) \tag{74}$$

$$\mathbb{E}\left[d_k(S_k, Y_k) \mid X\right] = \mathbb{E}\left[Y_k \log\left(\frac{Y_k}{Q_k}\right) \bigg| X\right] + Q_k \log\left(\frac{Q_k}{C_k}\right) + Q_k \log\left(\frac{C_k}{S_k}\right) \tag{75}$$

$$\mathbb{E}\left[d_k(S_k, Y_k) \mid S_k\right] = \mathbb{E}\left[Y_k \log\left(\frac{Y_k}{Q_k}\right) \bigg| S_k\right] + \mathbb{E}\left[Q_k \log\left(\frac{Q_k}{C_k}\right) \bigg| S_k\right] + C_k \log\left(\frac{C_k}{S_k}\right) \tag{76}$$

$$= \mathbb{E}\left[d_k(Q_k, Y_k) \mid S_k\right] + \mathbb{E}\left[d_k(C_k, Q_k) \mid S_k\right] + d_k(S_k, C_k) \tag{77}$$

where we have used the same properties as those described for the proof of the Brier score classwise decomposition above. The desired decomposition is then obtained by taking the expectation over $S_k$ and summing over the K classes.

$\square$

**The proper scoring rule decomposition holds for top-label calibration.**   Unlike classwise calibration, top-label calibration does not define a vector $C \in \mathbb{R}^K$ of calibrated probabilities. Instead, it defines a notion of calibration for a simpler binary problem in which labels indicate whether the classifier predicts the correct class for a given $X$. More precisely, the labels for this binary

problem are given by $Y' := 1_{Y = e_{\arg\max(S)}}$. Since $S$ is a function of $X$, the random variable $Y'$ is a function of $Y$ and $X$. Define now the scores associated to this binary problem as $S' := \max(S) \in \mathbb{R}$. Reformulated in terms of these notations, top-label calibration states that $S'$ is well calibrated if for all $s$, $P(Y' = 1 | S' = s) = s$. Thus, as for a classical binary problem, we can define $C' := \mathbb{E}[Y'|S']$ and $Q' = \mathbb{E}[Y'|X]$. $C'$ (resp $Q'$) gives the probability that the classifier predicts the correct class for a given score $S'$ (resp. a given input $X$). As the quantities $S'$, $C'$, $Q'$ and $Y'$ define a classical binary problem, the decomposition (6) into calibration, grouping, and irreducible loss holds for this problem. Compared to the classwise definition of calibration and grouping, here the calibration loss measures whether on average over all points scored $S$ *across all classes*, the proportion of correctly predicted points in actually $S$. In this setting, the grouping loss also measures to what extent there exist over-confident scores for certain classes that compensate under-confident scores for other classes.

### C.9.2   RESULTS HOLD FOR BRIER AND LOG-LOSS IN CLASSWISE SETTING

Appendix C.9.1 proves the scoring rule decomposition (6) in a classwise setting for Brier and log-loss scoring rules, which is necessary for the other results to hold. However, the proof of Lemma 4.1 does not readily apply to classwise calibration. Equation 22 uses a conditioning on the full vector of joint confiences $S$ to move $\phi(C, e_k)$ outside of the conditional expectation on $S$ and turn $Q_k$ into $C_k$ in expectation. In classwise calibration the conditioning is on each marginal $S_k$ instead of the joint $S$. As a result, in the general case, $\phi(C, e_k)$ cannot be moved outside of the conditional expectation given $S_k$ since $C$ depends on all marginals of $C$, not just $C_k$. However for some scoring rules, $\phi(p, e_k)$ depends only on $p_k$ and the proof can be adapted. This is the case of the log-loss for which $\phi^{LL}(p, e_k) = -\log(p_k)$.

**Lemma C.4** (Adaptation of Lemma 4.1 for classwise calibration)**.** *Suppose there exists $g : \mathbb{R}^K \to \mathbb{R}$ such that for all $k$ in $\{1, \dots, K\}$ and $x$ in $\mathbb{R}^K$, $\phi(x, e_k) = g(x_k)$. Define $h_k : p \mapsto -\phi(p, e_k)p_k$, the $k^{th}$ component of the negative entropy of the scoring rule $\phi$. The grouping loss $\mathrm{GL}$ of the classifier $S$ with calibrated scores $C_k = \mathbb{E}[Q_k \,|\, S_k]$ and scoring rule $\phi$ writes:*

$$\underbrace{\mathbb{E}[d_\phi(C, Q)]}_{\mathrm{GL}(S)} \;=\; \sum_{k=1}^K \mathbb{E}[\mathbb{V}_{h_k}[Q_k \,|\, S_k]] \tag{78}$$

*Proof of Lemma C.4.* Define the vector $C$ with $C_k = \mathbb{E}[Q_k \,|\, S_k]$ for all $k$ in $\{1, \dots, K\}$. Let $\phi$ be a scoring rule, $h : p \mapsto -s_\phi(p, p)$. Suppose for all $k$ in $\{1, \dots, K\}$, $\phi(x, e_k) = g(x_k)$ with $g : \mathbb{R}^K \to \mathbb{R}$. Then:

$$
\begin{aligned}
\mathbb{E}[d_\phi(C, Q)] &= \mathbb{E}[s_\phi(C, Q) - s_\phi(Q, Q)] && \text{Definition of } d_\phi && (79)\\
&= \mathbb{E}\Big[\textstyle\sum_{k=1}^K \phi(C, e_k)Q_k + \sum_{k=1}^K \phi(Q, e_k)Q_k\Big] && \text{Definition of } s_\phi && (80)\\
&= \textstyle\sum_{k=1}^K \mathbb{E}[\phi(C, e_k)Q_k - \phi(Q, e_k)Q_k] && \text{Linearity of expectation} && (81)\\
&= \textstyle\sum_{k=1}^K \mathbb{E}[g(C_k)Q_k - g(Q_k)Q_k] && \text{Hypothesis on } \phi && (82)\\
&= \textstyle\sum_{k=1}^K \mathbb{E}[\mathbb{E}[g(C_k)Q_k \,|\, S_k] - \mathbb{E}[g(Q_k)Q_k \,|\, S_k]] && \text{Law of total expectation} && (83)\\
&= \textstyle\sum_{k=1}^K \mathbb{E}[g(C_k)\mathbb{E}[Q_k \,|\, S_k] - \mathbb{E}[g(Q_k)Q_k \,|\, S_k]] && C_k \text{ is a function of } S_k && (84)\\
&= \textstyle\sum_{k=1}^K \mathbb{E}[g(C_k)C_k - \mathbb{E}[g(Q_k)Q_k \,|\, S_k]] && \text{Definition of } C_k && (85)\\
&= \textstyle\sum_{k=1}^K \mathbb{E}[\mathbb{V}_{h_k}[Q_k \,|\, S_k]] && \text{Definition of } \mathbb{V}_{h_k} && (86)
\end{aligned}
$$

$\square$

**Theorem C.2** (Results in classwise setting)**.** *Suppose Equation 78 is satisfied for the scoring rule $\phi$. For all $k \in \{1, \dots, K\}$, let $\mathcal{R}_k : \mathcal{X} \to \mathbb{N}$ be a partition of the feature space. It holds that:*

$$\mathrm{GL}(S) = \underbrace{\textstyle\sum_{k=1}^K \mathbb{E}[\mathbb{V}_{h_k}[\mathbb{E}[Q_k \,|\, S_k, \mathcal{R}_k] \,|\, S_k]]}_{\mathrm{GL}_{explained}(S)} + \underbrace{\textstyle\sum_{k=1}^K \mathbb{E}[\mathbb{V}_{h_k}[Q_k \,|\, S_k, \mathcal{R}_k]]}_{\mathrm{GL}_{residual}(S)} \tag{87}$$

$$\underbrace{\textstyle\sum_{k=1}^K \mathbb{E}[\mathbb{V}_{h_k}[Q_k \,|\, S_{B_k}]]}_{\mathrm{GL}(S_B)} = \underbrace{\textstyle\sum_{k=1}^K \mathbb{E}[\mathbb{V}_{h_k}[Q_k \,|\, S_k]]}_{\mathrm{GL}(S)} + \underbrace{\textstyle\sum_{k=1}^K \mathbb{E}[\mathbb{V}_{h_k}[C_k \,|\, S_{B_k}]]}_{\mathrm{GL}_{induced}(S, S_B)} \tag{88}$$

$$\mathrm{GL}(S) = \mathrm{GL}_{explained}(S_B) - \mathrm{GL}_{induced}(S, S_B) + \mathrm{GL}_{residual}(S_B) \tag{89}$$

*Moreover, if $h_k$ is convex, then:*

$$\mathrm{GL}(S) \geq \mathrm{GL}_{explained}(S) \geq 0 \tag{90}$$

$$\mathrm{GL}_{induced}(S, S_B) \geq 0 \tag{91}$$

$$\mathrm{GL}(S) \geq \underbrace{\mathrm{GL}_{explained}(S_B) - \mathrm{GL}_{induced}(S, S_B)}_{\mathrm{GL}_{\mathrm{LB}}(S, S_B)} \tag{92}$$

*Proof of Theorem C.2.* Applying the law of total variance (Lemma C.1) on each of the $\mathbb{V}_{h_k}[Q_k \,|\, S_k]$ with $\mathcal{R}_k$ as conditioning variable proves Equation 87. Similarly, applying the law of total variance on each of the $\mathbb{V}_{h_k}[Q_k \,|\, S_{B_k}]$ with $S_k$ as conditioning variable proves Equation 88. The proof for Equation 89 is the same as Proposition 4.2.

Using Jensen's inequality, if $h_k$ is convex, then $\mathbb{V}_{h_k} \geq 0$, which proves Equation 90, 91 and 92. □

For the log-loss scoring rule, we have $\phi^{LL}(p, e_k) = -\log(p_k)$ and $h_k(p) = \log(p_k)p_k$ wich is convex. Thus, Theorem C.2 holds for the log-loss. Unfortunately the Brier score does not satisfy the assumptions of Lemma C.4 since $\phi^{BS}(p, e_k)$ is not a function of $p_k$. But a forumlation similar to Equation 78 holds for the Brier score:

$$
\begin{align}
\mathbb{E}\big[d_{\phi^{BS}}(C, Q)\big] &= \mathbb{E}\big[s_{\phi^{BS}}(C, Q) - s_{\phi^{BS}}(Q, Q)\big] && \text{Definition of } d_{\phi^{BS}} \tag{93} \\
&= \mathbb{E}[(C - Q) \cdot (C - Q)] && \text{Definition of } s_{\phi^{BS}} \tag{94} \\
&= \mathbb{E}[(C \cdot C - 2C \cdot Q + Q \cdot Q)] && \tag{95} \\
&= \mathbb{E}[(Q \cdot Q - C \cdot C)] && \mathbb{E}[C \cdot Q] = C \cdot C \tag{96} \\
&= \textstyle\sum_{k=1}^{K} \mathbb{E}\big[(Q_k^2 - C_k^2)\big] && \text{Linearity of expectation} \tag{97} \\
&= \textstyle\sum_{k=1}^{K} \mathbb{E}\big[(\mathbb{E}\big[Q_k^2 \,|\, S_k\big] - C_k^2)\big] && \text{Law of total expectation} \tag{98} \\
&= \textstyle\sum_{k=1}^{K} \mathbb{E}[\mathbb{V}[Q_k \,|\, S_k]] && \text{Definition of the variance} \tag{99} \\
\text{with:} && \tag{100} \\
\mathbb{E}[C \cdot Q] &= \textstyle\sum_{k=1}^{K} \mathbb{E}[C_k Q_k] && \tag{101} \\
&= \textstyle\sum_{k=1}^{K} \mathbb{E}[\mathbb{E}[C_k Q_k \,|\, S_k]] && \text{Law of total expectation} \tag{102} \\
&= \textstyle\sum_{k=1}^{K} \mathbb{E}[C_k \mathbb{E}[Q_k \,|\, S_k]] && C_k \text{ is a function of } S_k \tag{103} \\
&= \textstyle\sum_{k=1}^{K} \mathbb{E}\big[C_k^2\big] && \text{Definition of } C_k \tag{104} \\
&= \mathbb{E}[C \cdot C] && \tag{105}
\end{align}
$$

Since $\mathbb{V} = \mathbb{V}_f$ with $f : x \mapsto x^2$ Equation 78 is satisfied for the Brier score. Since $f$ is convex, Theorem C.2 holds for the Brier score.

To conclude, Theorem C.2 holds for the Brier score and the log-loss in a classwise setting. It is likely that some other proper scoring rules satisfy Equation 78 and Theorem C.2.

## C.10 IMPACT OF RECALIBRATION ON THE GROUPING LOSS

**Lemma C.5.** *Let $\hat{c}$ be a recalibration mapping and $S' = \hat{c}(S)$ the classifier recalibrated with that mapping. The grouping loss of the recalibrated classifier $\mathrm{GL}(S')$ deviates from that of the original classifier $\mathrm{GL}(S)$ as follows:*

$$\mathrm{GL}(S') = \mathrm{GL}(S) + \mathbb{E}[\mathbb{V}_h[C \,|\, S']]$$

*If the mapping is perfect (i.e. $S' = C$) or invertible, then $\mathrm{GL}(S') = \mathrm{GL}(S)$.*

*Proof of Lemma C.5.*

$$\begin{aligned}
\mathrm{GL}(S') &= \mathbb{E}[\mathbb{V}_h[Q \mid S']] && \text{Definition of GL} \\
&= \mathbb{E}[\mathbb{V}_h[Q \mid S', S]] + \mathbb{E}[\mathbb{V}_h[\mathbb{E}[Q \mid S', S] \mid S']] && \text{Law of total } h\text{-variance on } S \text{ (Lemma C.1)} \\
&= \mathbb{E}[\mathbb{V}_h[Q \mid S]] + \mathbb{E}[\mathbb{V}_h[\mathbb{E}[Q \mid S] \mid S']] && S' \text{ is a function of } S \\
&= \mathrm{GL}(S) + \mathbb{E}[\mathbb{V}_h[C \mid S']] && \text{Definition of } \mathrm{GL}(S) \text{ and } C
\end{aligned}$$

If $S' = C$, then $\mathbb{V}_h[C \mid S'] = 0$, hence $\mathrm{GL}(S') = \mathrm{GL}(S)$. If the mapping $\hat{c}$ is invertible, then knowing $S'$ is knowing $S$. Hence $\mathbb{V}_h[C \mid S'] = \mathbb{V}_h[C \mid S] = 0$ since $C$ a function of $S$. Hence $\mathrm{GL}(S') = \mathrm{GL}(S)$. $\qquad\square$

# D  IMAGENET

ImageNet-1K (ILSVRC2012) (Deng et al., 2009) is a classification dataset for computer vision with 1 000 classes. Networks studied in this article are pre-trained on the training set of ImageNet-1K, comprising 1.2 million samples. Models' architectures and weights are available on PyTorch v0.12 (Paszke et al., 2019). We evaluated the networks on ImageNet variants' ImageNet-R and ImageNet-C (Appendix D.1, D.2) as well as the validation set of ImageNet-1K (Appendix D.3). We work in the high-level feature space of the networks, *i.e.* the output space of the penultimate layer (embedding space).

For each of ImageNet-R, ImageNet-C and the validation set of ImageNet, we plot the grouping diagrams of each network with and without post-hoc recalibration, obtained with a balanced decision stump. For each network, if several versions are available, we study both the smallest and the best performing one on the validation set of ImageNet-1K (usually the largest version). For ImageNet-R we also provide the grouping diagrams obtained with a 2-cluster $k$-means. Each experiment is detailed in Appendix D.1, D.2 and D.3.

**Detailed experimental method**  First, we forward each sample of the evaluation dataset (ImageNet-R, ImageNet-C or the validation set of ImageNet-1K) through the studied network. We build confidence scores by applying a softmax to the output logits. We extract a representation of the input images in the high-level feature space of the network (*i.e.* the input space of the last linear layer). Since there is not enough samples per class (50), we restrict our study to the top-label problem (Definition 3.3). For each sample, the class with the highest confidence is predicted. The label is 1 if the network predicted a correct class (0 otherwise) and the associated confidence score is the one of the predicted class. We divide the samples of the evaluation set in half making sure that the confidence score distribution is the same in both resulting subsets. On one set, we train the isotonic regression for calibration and calibrate the confidence scores of both sets. If no post-hoc recalibration is used, we skip this step. Then, we create groups of same-level confidences by binning the confidence scores with 15 equal-width bins in $[0, 1]$. We partition each of the 15 level sets independently. For each of them, we create the partition by training the partitioning method on the training samples of the isotonic regression. We then evaluate region scores on the remaining samples to avoid overfitting. For the grouping diagrams, we mainly use a balanced decision stump with 2 clusters (*e.g.* using scikit-learn's `DecisionTreeRegressor` with `min_samples_leaf` taken as half the samples in the bin), resulting in one split along one of the axis of the high-level feature space. For comparison, we also used $k$-means with 2 clusters. Constraining the partitioning methods to 2 regions is a choice to provide visually informative grouping diagrams rather than to maximize the lower bound $\widehat{\mathrm{GL}}_{explained}$. When optimizing the lower bound, (Figure 7 and Figure 14), we increase the number of allowed regions in the partition by setting a region ratio: the number of training samples in the bin over the number of allowed regions in the bin. Fixing a region ratio prevents from having regions with too few samples. In our experiments, we fix the region ratio to 30.

## D.1  IMAGENET-R

ImageNet-R (Hendrycks et al., 2021) is a variant of ImageNet containing renditions of the ImageNet classes. Example of renditions are: paintings, toys, tattoos and origami. There are 15 rendition types in total listed in Figure 13. The dataset contains 30 000 images and is limited to 200 of the 1 000 ImageNet classes. Figure 14a. compares estimated grouping loss lower bound and calibration errors of all networks (small and best versions) on ImageNet-R. Overall, we observe a strong grouping loss

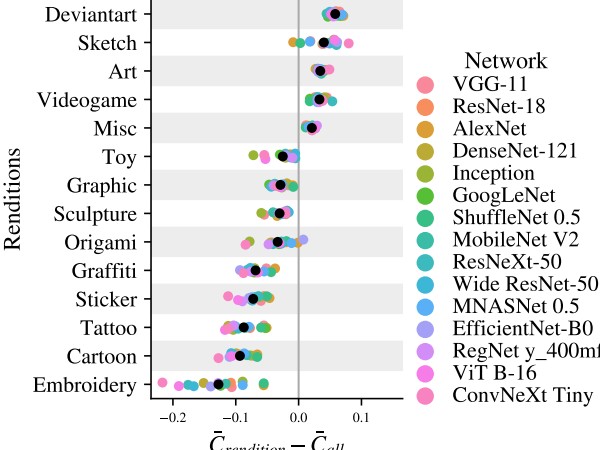

Figure 13: **Comparison of renditions on ImageNet-R.** Differences between the calibrated scores of samples of one rendition $\bar{C}_{rendition}$ and the calibrated scores of all samples $\bar{C}_{all}$, weighted by the number of samples of this rendition in the level set and summed over the 15 bins on confidence scores. Interpretation: should renditions define regions of the feature space, they would exhibit a high grouping loss lower bound.

in most of the networks, especially those with highest accuracy. The estimated debiased lower bound remains high after post-hoc recalibration. Grouping diagrams of all networks are available at:

- Section D.1.1: ImageNet-R – No post-hoc recalibration, small versions.
- Section D.1.2: ImageNet-R – No post-hoc recalibration, small versions, k-means.
- Section D.1.3: ImageNet-R – No post-hoc recalibration, best versions.
- Section D.1.4: ImageNet-R – Isotonic recalibration, small versions.
- Section D.1.5: ImageNet-R – Isotonic recalibration, best versions.

We also investigate whether there is heterogeneity among renditions. In Figure 13 we observe that some renditions are better predicted than average (*e.g.* deviant art, sketch or art) while some others are predicted worse than average (*e.g.* embroidery, cartoon, tattoo). These considerations would be useful in a fairness setting. Also, Figure 13 highlights that if we could build regions out of renditions (*i.e.* renditions are well separated in the feature space), this would result in a high grouping loss lower bound.

## D.2 IMAGENET-C

ImageNet-C is a variant of ImageNet containing corrupted versions of ImageNet images. Examples of corruptions are: blur, noise, saturate, contrast, brightness and compression. There are 19 corruption types in total. Each corruption has a severity ranging from 1 to 5. The dataset contains the 50 000 images of the validation set of ImageNet, each of them being applied 19 corruptions with 5 severity levels each. We built a merged version of ImageNet-C by randomly sampling one corruption for each image. We also study one corruption only (snow). For both the merged version and the snow version, we study the maximum severity of the corruption (5). Figure 14b. and c. compare estimated grouping loss lower bound and calibration errors of all networks (small and best versions) on ImageNet-C merged and snow. Overall, we observe similar effect than on ImageNet-R. However, when all samples have the same corruption (snow), we exhibit more grouping loss among the networks than when the 19 corruptions are randomly applied on the dataset (merged) (Figure 14c.). An intuition is that heterogeneity created by one corruption is canceled out by another one having heterogeneity in the opposite direction, leading to region scores closer to the average. Grouping diagrams of all networks are available at:

- Section D.2.1: ImageNet-C – No post-hoc recalibration, small versions.
- Section D.2.2: ImageNet-C – No post-hoc recalibration, best versions.
- Section D.2.3: ImageNet-C – Isotonic recalibration, small versions.
- Section D.2.4: ImageNet-C – Isotonic recalibration, best versions.

## D.3 IMAGENET-1K VALIDATION SET

The validation set of ImageNet-1K comprises 50 000 samples for 1 000 classes. Figure 14d. compares estimated grouping loss lower bound and calibration errors of all networks (small and best versions)

on the validation set of ImageNet-1K. Conversely to ImageNet-R and ImageNet-C, we cannot exhibit substantial grouping loss on any of the networks. The grouping diagrams (Figure 24) show however that ConvNeXt Tiny displays more heterogeneity than the other networks on this dataset. Grouping diagrams are available in:

- Section D.3.1: ImageNet-1K – No post-hoc recalibration, small versions.
- Section D.3.2: ImageNet-1K – No post-hoc recalibration, best versions.
- Section D.3.3: ImageNet-1K – Isotonic recalibration, small versions.
- Section D.3.4: ImageNet-1K – Isotonic recalibration, best versions.

## E  NLP

We use BART Large (Lewis et al., 2019) pre-trained on the Multi-Genre Natural Language Inference dataset (Williams et al., 2018) and fine-tuned on the Yahoo Answers Topics dataset for zero-shot topic classification. The fine-tuned model is available on HuggingFace at `https://huggingface.co/joeddav/bart-large-mnli-yahoo-answers`. Yahoo Answers Topics is composed of question titles and bodies and topic labels. There are $1\,400\,000$ training samples, $60\,000$ test samples and 10 topics. The dataset is available at `https://huggingface.co/datasets/yahoo_answers_topics`. The model is fine-tuned on 5 out of the 10 topics of the training set, totalizing $700\,000$ samples. Given a question title and a hypothesis (*e.g.* "This text is about Science & Mathematics"), the model outputs its confidence in the hypothesis to be true for the given question. The classification being zero-shot, the hypothesis can be about an unseen topic. We evaluate the model separately on the 5 unseen topics and the 5 seen topics of the test set (*i.e.* seen topics but unseen samples). This results in a binary classification task in which each sample is composed of a question title and a hypothesis and each label is 1 or 0 whether the hypothesis is correct or not. As for the clustering and calibration procedure, we used a balanced decision stump in the same way as described in Section D: "Detailed experimental method". We work in the high-level feature space of the network, *i.e.* the output space of the penultimate layer (embedding space).

**a. Vision**

| Network | $\widehat{\mathrm{CL}}$ | $\widehat{\mathrm{CL}}'$ | $\widehat{\mathrm{GL}}_{\mathrm{LB}}$ | $\widehat{\mathrm{GL}}'_{\mathrm{LB}}$ | Accuracy↑ (%) |
|---|---|---|---|---|---|
| ViT B-16 | 0.044 | 0.000 | 0.019 | 0.021 | 30.1 |
| ConvNeXt Tiny | 0.017 | 0.000 | 0.036 | 0.039 | 29.0 |
| Inception | 0.200 | 0.000 | 0.006 | 0.009 | 27.3 |
| Wide ResNet-50 | 0.082 | 0.000 | 0.020 | 0.019 | 25.9 |
| ResNeXt-50 | 0.091 | 0.000 | 0.013 | 0.013 | 25.1 |
| EfficientNet-B0 | 0.028 | 0.000 | 0.015 | 0.018 | 24.1 |
| DenseNet-121 | 0.059 | 0.000 | 0.013 | 0.010 | 23.9 |
| GoogLeNet | 0.009 | 0.001 | 0.011 | 0.011 | 23.3 |
| RegNet y_400mf | 0.084 | 0.000 | 0.011 | 0.010 | 21.0 |
| ResNet-18 | 0.057 | 0.000 | 0.010 | 0.009 | 20.4 |
| MobileNet V2 | 0.070 | 0.000 | 0.007 | 0.009 | 18.9 |
| MNASNet 0.5 | 0.017 | 0.000 | 0.005 | 0.007 | 15.9 |
| VGG-11 | 0.061 | 0.000 | 0.007 | 0.007 | 15.8 |
| ShuffleNet 0.5 | 0.072 | 0.000 | 0.004 | 0.003 | 14.6 |
| AlexNet | 0.063 | 0.000 | 0.003 | 0.003 | 12.6 |

**b. NLP**

| Setting | $\widehat{\mathrm{CL}}$ | $\widehat{\mathrm{CL}}'$ | $\widehat{\mathrm{GL}}_{\mathrm{LB}}$ | $\widehat{\mathrm{GL}}'_{\mathrm{LB}}$ | Accuracy ↑ (%) |
|---|---|---|---|---|---|
| in-distribution | 0.026 | 0.000 | 0.000 | 0.000 | 88.2 |
| out-of-distribution | 0.091 | 0.000 | 0.015 | 0.015 | 71.1 |

Table 1: Raw values of the estimators in the vision (Figure 7) and NLP experiments of Section 5.2, before ($\widehat{\mathrm{CL}}$ and $\widehat{\mathrm{GL}}_{\mathrm{LB}}$) and after ($\widehat{\mathrm{CL}}'$ and $\widehat{\mathrm{GL}}'_{\mathrm{LB}}$) isotonic recalibration.

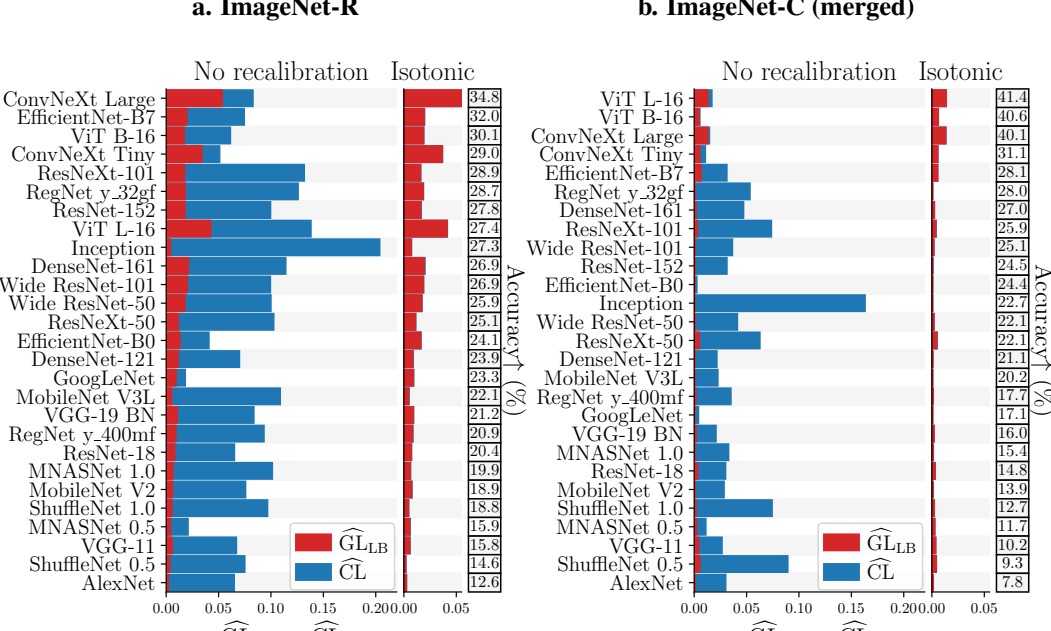

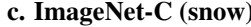

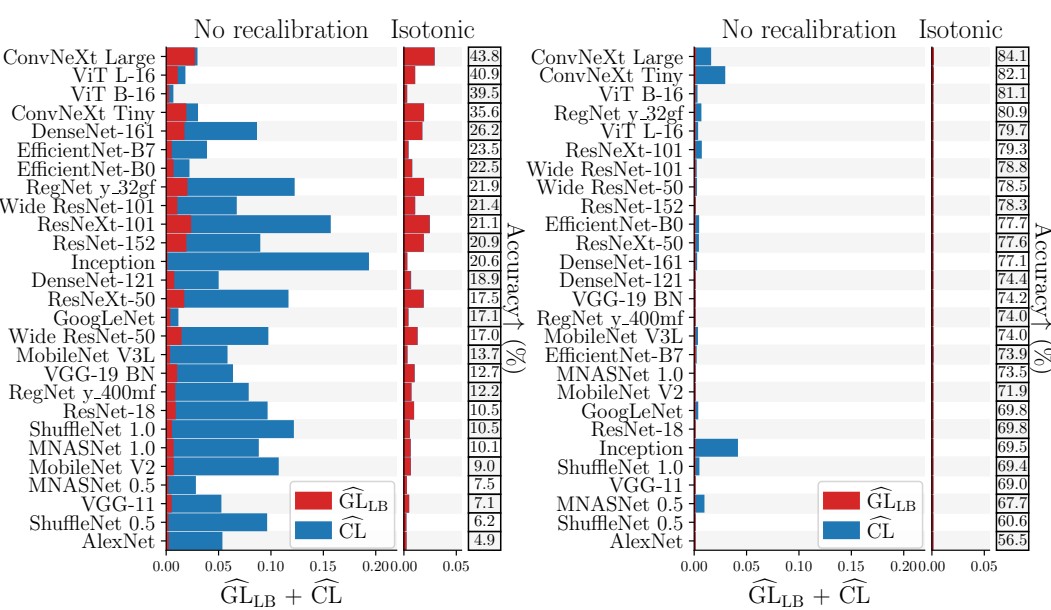

Figure 14: **Comparing vision models**: a debiased estimate of the grouping loss lower bound $\widehat{\mathrm{GL}}_{\mathrm{LB}}$ (Equation 13) and an estimate of the calibration loss $\widehat{\mathrm{CL}}$, both accounting for binning, evaluated on ImageNet-R, ImageNet-C and ImageNet-1K, sorted by model accuracy. Partitions $\mathcal{R}$ are obtained from a decision tree partitioning constrained to create at most $^{\#\,\mathrm{samples\,in\,bin}}/_{30}$ regions in each bin. Isotonic regression is used for post-hoc recalibration of the models (right).

**a. ImageNet-R**

| Network | $\widehat{\mathrm{CL}}$ | $\widehat{\mathrm{CL}}'$ | $\widehat{\mathrm{GL}}_{\mathrm{LB}}$ | $\widehat{\mathrm{GL}}'_{\mathrm{LB}}$ | Accuracy↑ (%) |
|---|---|---|---|---|---|
| ConvNeXt Large | 0.029 | 0.000 | 0.055 | 0.060 | 34.8 |
| EfficientNet-B7 | 0.055 | 0.000 | 0.022 | 0.021 | 32.0 |
| ViT B-16 | 0.044 | 0.000 | 0.019 | 0.021 | 30.1 |
| ConvNeXt Tiny | 0.017 | 0.000 | 0.036 | 0.039 | 29.0 |
| ResNeXt-101 | 0.114 | 0.000 | 0.019 | 0.018 | 28.9 |
| RegNet y_32gf | 0.108 | 0.000 | 0.020 | 0.020 | 28.7 |
| ResNet-152 | 0.082 | 0.000 | 0.020 | 0.018 | 27.8 |
| ViT L-16 | 0.096 | 0.000 | 0.044 | 0.043 | 27.4 |
| Inception | 0.200 | 0.000 | 0.006 | 0.009 | 27.3 |
| DenseNet-161 | 0.093 | 0.001 | 0.023 | 0.021 | 26.9 |
| Wide ResNet-101 | 0.079 | 0.000 | 0.022 | 0.020 | 26.9 |
| Wide ResNet-50 | 0.082 | 0.000 | 0.020 | 0.019 | 25.9 |
| ResNeXt-50 | 0.091 | 0.000 | 0.013 | 0.013 | 25.1 |
| EfficientNet-B0 | 0.028 | 0.000 | 0.015 | 0.018 | 24.1 |
| DenseNet-121 | 0.059 | 0.000 | 0.013 | 0.010 | 23.9 |
| GoogLeNet | 0.009 | 0.001 | 0.011 | 0.011 | 23.3 |
| MobileNet V3L | 0.104 | 0.000 | 0.007 | 0.006 | 22.2 |
| VGG-19 BN | 0.073 | 0.000 | 0.012 | 0.011 | 21.2 |
| RegNet y_400mf | 0.084 | 0.000 | 0.011 | 0.010 | 21.0 |
| ResNet-18 | 0.057 | 0.000 | 0.010 | 0.009 | 20.4 |
| MNASNet 1.0 | 0.095 | 0.000 | 0.008 | 0.008 | 19.9 |
| MobileNet V2 | 0.070 | 0.000 | 0.007 | 0.009 | 18.9 |
| ShuffleNet 1.0 | 0.091 | 0.000 | 0.007 | 0.006 | 18.8 |
| MNASNet 0.5 | 0.017 | 0.000 | 0.005 | 0.007 | 15.9 |
| VGG-11 | 0.061 | 0.000 | 0.007 | 0.007 | 15.8 |
| ShuffleNet 0.5 | 0.072 | 0.000 | 0.004 | 0.003 | 14.6 |
| AlexNet | 0.063 | 0.000 | 0.003 | 0.003 | 12.6 |

**b. ImageNet-C (merged)**

| Network | $\widehat{\mathrm{CL}}$ | $\widehat{\mathrm{CL}}'$ | $\widehat{\mathrm{GL}}_{\mathrm{LB}}$ | $\widehat{\mathrm{GL}}'_{\mathrm{LB}}$ | Accuracy↑ (%) |
|---|---|---|---|---|---|
| ViT L-16 | 0.004 | 0.000 | 0.016 | 0.016 | 41.4 |
| ViT B-16 | 0.001 | 0.000 | 0.007 | 0.009 | 40.6 |
| ConvNeXt Large | 0.002 | 0.000 | 0.016 | 0.016 | 40.1 |
| ConvNeXt Tiny | 0.005 | 0.000 | 0.008 | 0.008 | 31.1 |
| EfficientNet-B7 | 0.025 | 0.000 | 0.009 | 0.007 | 28.1 |
| RegNet y_32gf | 0.053 | 0.000 | 0.002 | 0.002 | 28.0 |
| DenseNet-161 | 0.045 | 0.000 | 0.004 | 0.004 | 27.0 |
| ResNeXt-101 | 0.071 | 0.000 | 0.005 | 0.006 | 25.9 |
| Wide ResNet-101 | 0.036 | 0.000 | 0.003 | 0.004 | 25.1 |
| ResNet-152 | 0.031 | 0.000 | 0.002 | 0.002 | 24.5 |
| EfficientNet-B0 | 0.003 | 0.000 | 0.002 | 0.002 | 24.4 |
| Inception | 0.171 | 0.000 | -0.006 | 0.001 | 22.7 |
| Wide ResNet-50 | 0.040 | 0.000 | 0.003 | 0.003 | 22.1 |
| ResNeXt-50 | 0.058 | 0.000 | 0.006 | 0.006 | 22.1 |
| DenseNet-121 | 0.021 | 0.000 | 0.002 | 0.002 | 21.1 |
| MobileNet V3L | 0.024 | 0.000 | 0.001 | 0.001 | 20.2 |
| RegNet y_400mf | 0.034 | 0.000 | 0.003 | 0.002 | 17.7 |
| GoogLeNet | 0.004 | 0.000 | 0.001 | 0.001 | 17.1 |
| VGG-19 BN | 0.019 | 0.000 | 0.003 | 0.003 | 16.0 |
| MNASNet 1.0 | 0.032 | 0.000 | 0.002 | 0.002 | 15.4 |
| ResNet-18 | 0.027 | 0.000 | 0.004 | 0.004 | 14.8 |
| MobileNet V2 | 0.028 | 0.000 | 0.002 | 0.002 | 13.9 |
| ShuffleNet 1.0 | 0.073 | 0.000 | 0.002 | 0.003 | 12.7 |
| MNASNet 0.5 | 0.009 | 0.000 | 0.003 | 0.003 | 11.7 |
| VGG-11 | 0.022 | 0.000 | 0.005 | 0.004 | 10.2 |
| ShuffleNet 0.5 | 0.084 | 0.000 | 0.006 | 0.005 | 9.3 |
| AlexNet | 0.029 | 0.000 | 0.001 | 0.001 | 7.8 |

**c. ImageNet-C (snow)**

| Network | $\widehat{\mathrm{CL}}$ | $\widehat{\mathrm{CL}}'$ | $\widehat{\mathrm{GL}}_{\mathrm{LB}}$ | $\widehat{\mathrm{GL}}'_{\mathrm{LB}}$ | Accuracy↑ (%) |
|---|---|---|---|---|---|
| ConvNeXt Large | 0.002 | 0.000 | 0.030 | 0.031 | 43.8 |
| ViT L-16 | 0.007 | 0.000 | 0.013 | 0.013 | 40.9 |
| ViT B-16 | 0.004 | 0.000 | 0.005 | 0.005 | 39.5 |
| ConvNeXt Tiny | 0.011 | 0.000 | 0.021 | 0.021 | 35.6 |
| DenseNet-161 | 0.069 | 0.000 | 0.019 | 0.019 | 26.2 |
| EfficientNet-B7 | 0.034 | 0.000 | 0.006 | 0.005 | 23.5 |
| EfficientNet-B0 | 0.015 | 0.000 | 0.008 | 0.009 | 22.5 |
| RegNet y_32gf | 0.102 | 0.000 | 0.021 | 0.020 | 21.9 |
| Wide ResNet-101 | 0.057 | 0.000 | 0.012 | 0.011 | 21.4 |
| ResNeXt-101 | 0.133 | 0.000 | 0.025 | 0.025 | 21.1 |
| ResNet-152 | 0.070 | 0.000 | 0.020 | 0.020 | 20.9 |
| Inception | 0.194 | 0.000 | 0.000 | 0.004 | 20.6 |
| DenseNet-121 | 0.042 | 0.000 | 0.009 | 0.007 | 18.9 |
| ResNeXt-50 | 0.100 | 0.000 | 0.018 | 0.019 | 17.5 |
| GoogLeNet | 0.008 | 0.000 | 0.005 | 0.005 | 17.1 |
| Wide ResNet-50 | 0.083 | 0.000 | 0.015 | 0.014 | 17.0 |
| MobileNet V3L | 0.055 | 0.000 | 0.004 | 0.003 | 13.7 |
| VGG-19 BN | 0.053 | 0.000 | 0.011 | 0.010 | 12.7 |
| RegNet y_400mf | 0.070 | 0.000 | 0.009 | 0.007 | 12.2 |
| ResNet-18 | 0.088 | 0.000 | 0.009 | 0.010 | 10.5 |
| ShuffleNet 1.0 | 0.116 | 0.000 | 0.006 | 0.005 | 10.5 |
| MNASNet 1.0 | 0.081 | 0.000 | 0.007 | 0.006 | 10.1 |
| MobileNet V2 | 0.100 | 0.000 | 0.007 | 0.006 | 9.0 |
| MNASNet 0.5 | 0.026 | 0.000 | 0.002 | 0.003 | 7.5 |
| VGG-11 | 0.047 | 0.000 | 0.005 | 0.004 | 7.1 |
| ShuffleNet 0.5 | 0.094 | 0.000 | 0.002 | 0.002 | 6.2 |
| AlexNet | 0.051 | 0.000 | 0.002 | 0.002 | 4.9 |

**d. ImageNet-1K (validation set)**

| Network | $\widehat{\mathrm{CL}}$ | $\widehat{\mathrm{CL}}'$ | $\widehat{\mathrm{GL}}_{\mathrm{LB}}$ | $\widehat{\mathrm{GL}}'_{\mathrm{LB}}$ | Accuracy↑ (%) |
|---|---|---|---|---|---|
| ConvNeXt Large | 0.015 | 0.000 | 0.002 | 0.002 | 84.1 |
| ConvNeXt Tiny | 0.028 | 0.000 | 0.002 | 0.002 | 82.1 |
| ViT B-16 | 0.004 | 0.000 | -0.000 | 0.000 | 81.1 |
| RegNet y_32gf | 0.009 | 0.000 | -0.001 | -0.000 | 80.9 |
| ViT L-16 | 0.003 | 0.000 | 0.001 | 0.001 | 79.7 |
| ResNeXt-101 | 0.010 | 0.000 | -0.002 | 0.000 | 79.3 |
| Wide ResNet-101 | 0.004 | 0.000 | -0.001 | -0.000 | 78.8 |
| Wide ResNet-50 | 0.004 | 0.000 | -0.001 | -0.000 | 78.5 |
| ResNet-152 | 0.004 | 0.000 | -0.001 | 0.000 | 78.3 |
| EfficientNet-B0 | 0.006 | 0.000 | 0.000 | 0.000 | 77.7 |
| ResNeXt-50 | 0.006 | 0.000 | -0.001 | 0.000 | 77.6 |
| DenseNet-161 | 0.004 | 0.000 | -0.001 | -0.000 | 77.1 |
| DenseNet-121 | 0.001 | 0.000 | -0.000 | -0.000 | 74.4 |
| VGG-19 BN | 0.002 | 0.000 | -0.001 | -0.000 | 74.2 |
| RegNet y_400mf | 0.001 | 0.000 | -0.001 | -0.000 | 74.0 |
| MobileNet V3L | 0.006 | 0.000 | -0.001 | -0.000 | 74.0 |
| EfficientNet-B7 | 0.001 | 0.000 | 0.003 | 0.002 | 73.9 |
| MNASNet 1.0 | 0.003 | 0.000 | -0.001 | -0.000 | 73.5 |
| MobileNet V2 | 0.001 | 0.000 | -0.001 | 0.000 | 71.9 |
| GoogLeNet | 0.006 | 0.000 | -0.000 | 0.000 | 69.8 |
| ResNet-18 | 0.001 | 0.000 | 0.000 | -0.000 | 69.8 |
| Inception | 0.050 | 0.000 | -0.007 | 0.001 | 69.5 |
| ShuffleNet 1.0 | 0.008 | 0.000 | -0.001 | 0.000 | 69.4 |
| VGG-11 | 0.000 | 0.000 | -0.000 | 0.000 | 69.0 |
| MNASNet 0.5 | 0.012 | 0.000 | -0.001 | -0.000 | 67.7 |
| ShuffleNet 0.5 | 0.004 | 0.000 | -0.001 | 0.000 | 60.6 |
| AlexNet | 0.001 | 0.000 | -0.001 | -0.001 | 56.5 |

Table 2: Raw values of the estimators of Figure 14, before ($\widehat{\mathrm{CL}}$ and $\widehat{\mathrm{GL}}_{\mathrm{LB}}$) and after ($\widehat{\mathrm{CL}}'$ and $\widehat{\mathrm{GL}}'_{\mathrm{LB}}$) isotonic recalibration.

### D.1.1    IMAGENET-R – NO POST-HOC RECALIBRATION, SMALL VERSIONS

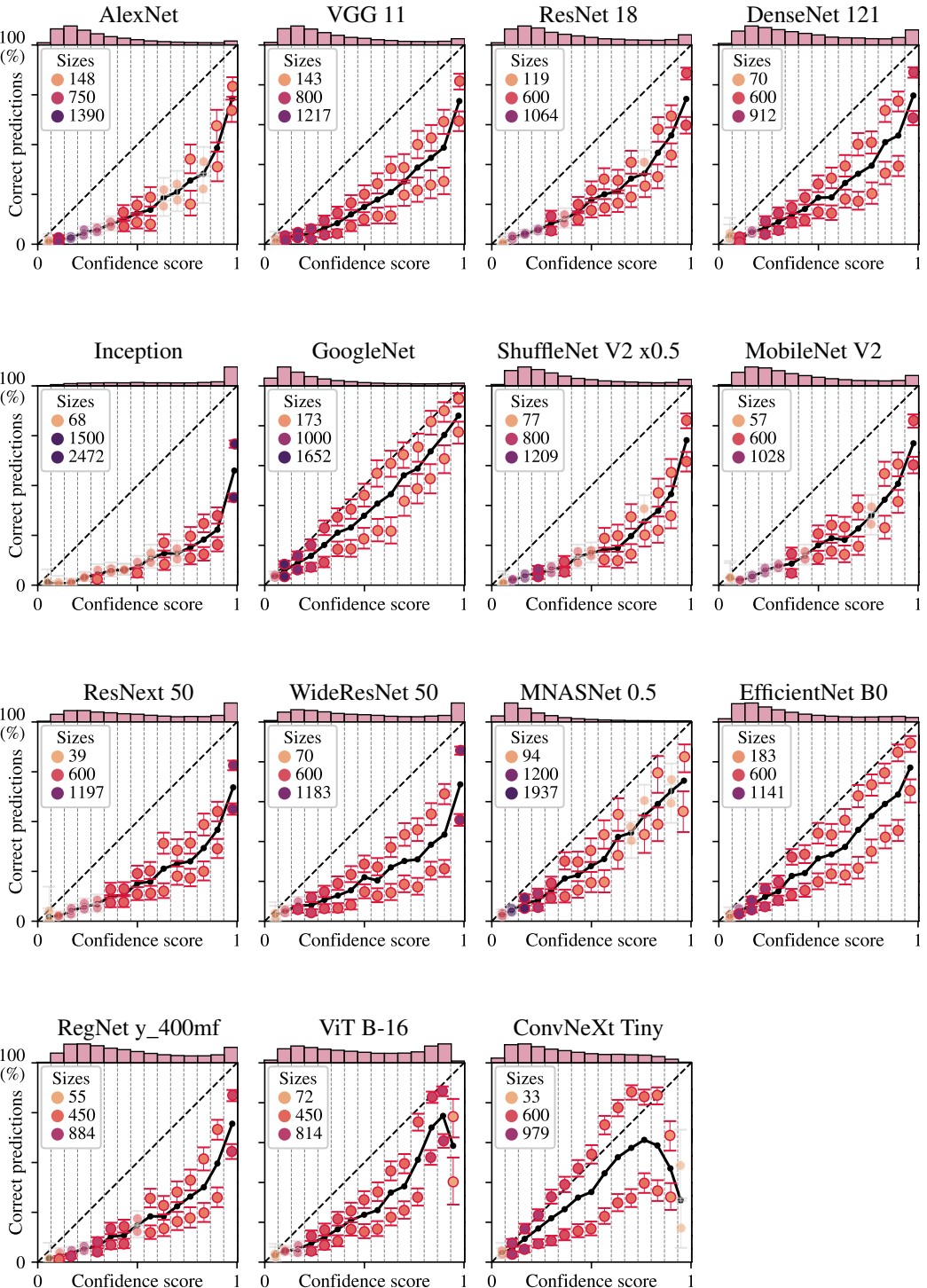

Figure 15: **Vision**: Fraction of correct predictions versus confidence score of predicted class ($\max_k S_k$) on ImageNet-R for small versions of pre-trained networks, without post-hoc recalibration. In each bin on confidence scores, the level set is partitioned into 2 regions with a decision stump constrained to one balanced split, with a 50-50 train-test split strategy.

### D.1.2 IMAGENET-R – NO POST-HOC RECALIBRATION, SMALL VERSIONS, K-MEANS

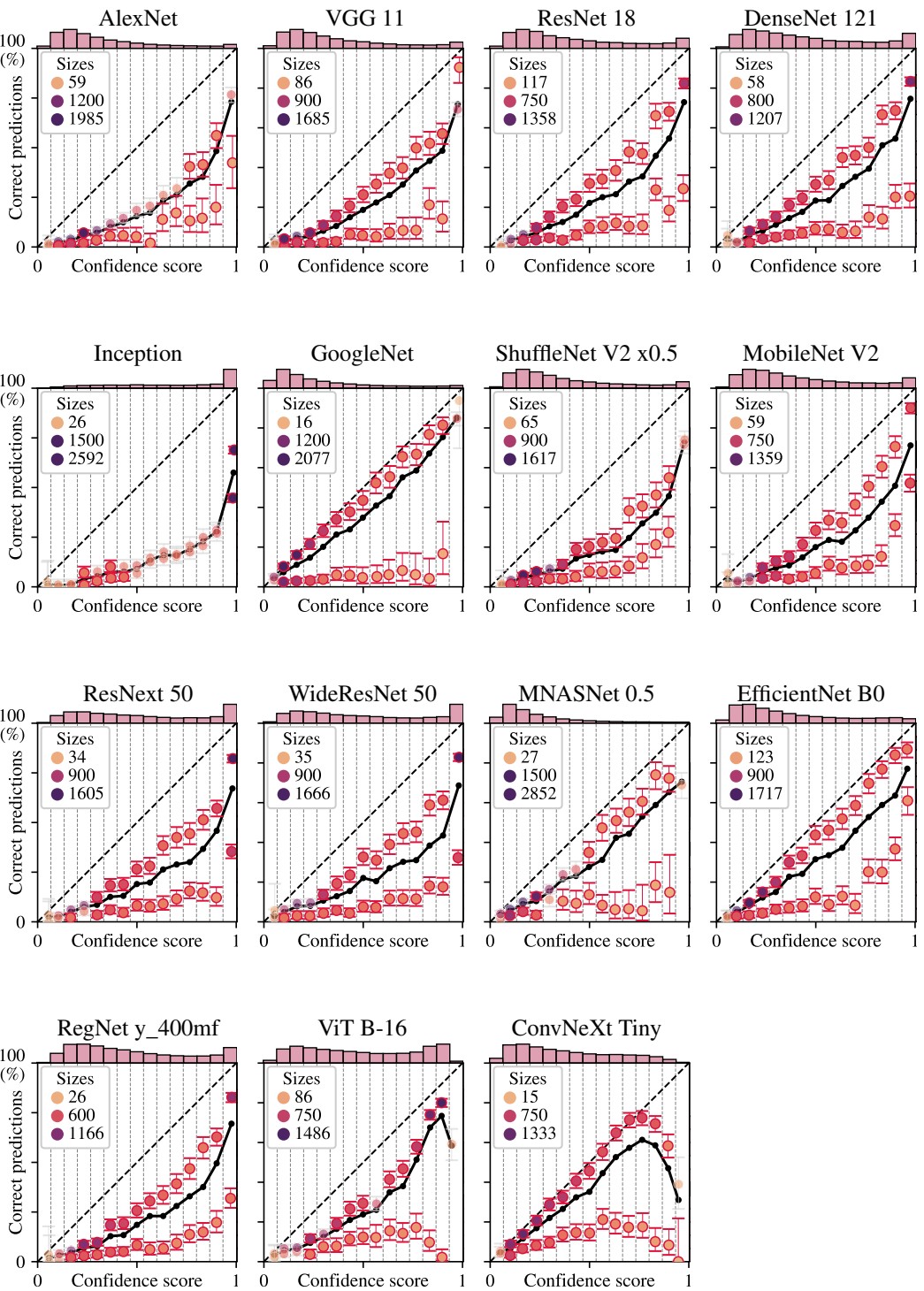

Figure 16: **Vision**: Fraction of correct predictions versus confidence score of predicted class ($\max_k S_k$) on ImageNet-R for small versions of pre-trained networks, without post-hoc recalibration. In each bin on confidence scores, the level set is partitioned into 2 regions with a $k$-means clustering, with a 50-50 train-test split strategy (for a fair comparison with decision stump clustering).

### D.1.3 IMAGENET-R – NO POST-HOC RECALIBRATION, BEST VERSIONS

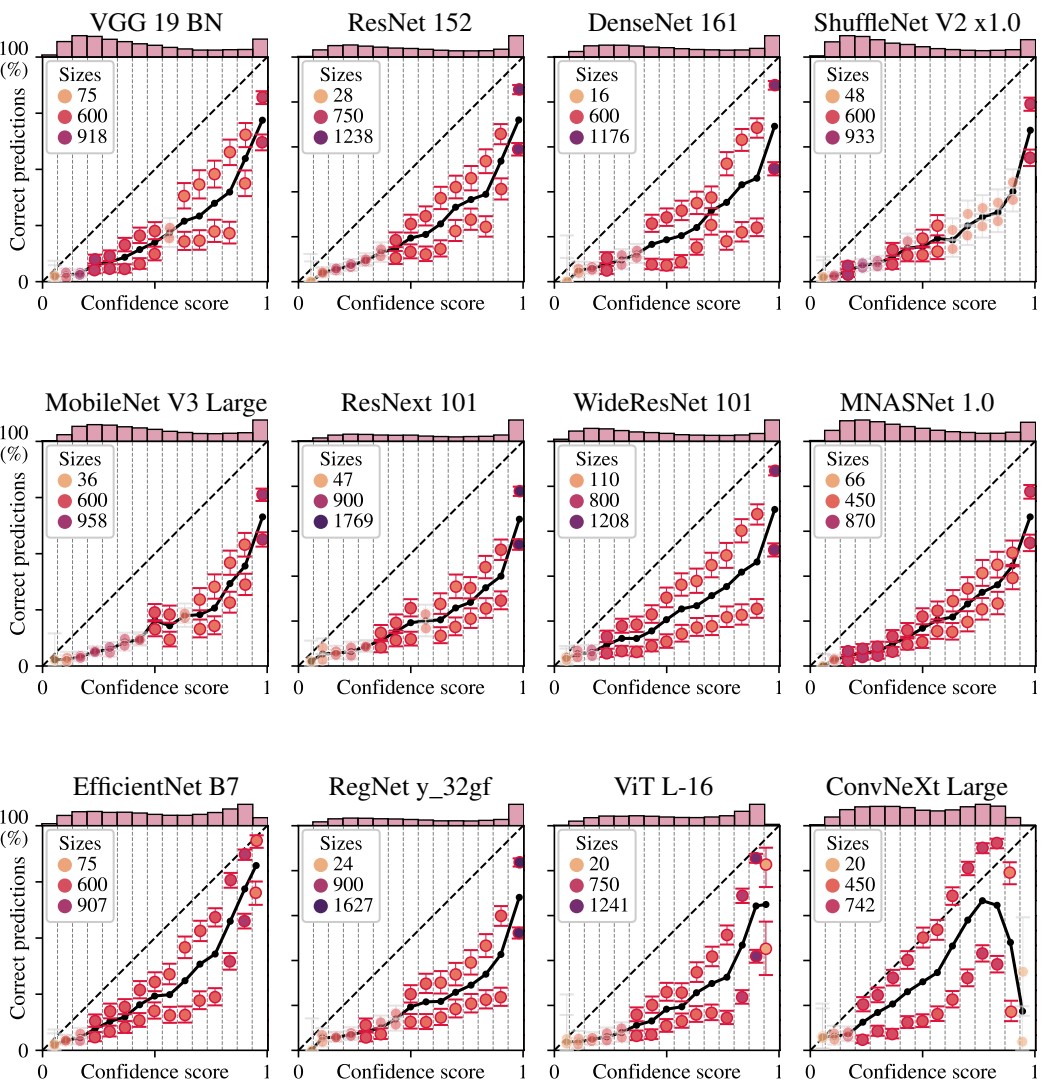

Figure 17: **Vision**: Fraction of correct predictions versus confidence score of predicted class ($\max_k S_k$) on ImageNet-R for best versions of pre-trained networks, without post-hoc recalibration. In each bin on confidence scores, the level set is partitioned into 2 regions with a decision stump constrained to one balanced split, with a 50-50 train-test split strategy.

### D.1.4 IMAGENET-R – ISOTONIC RECALIBRATION, SMALL VERSIONS

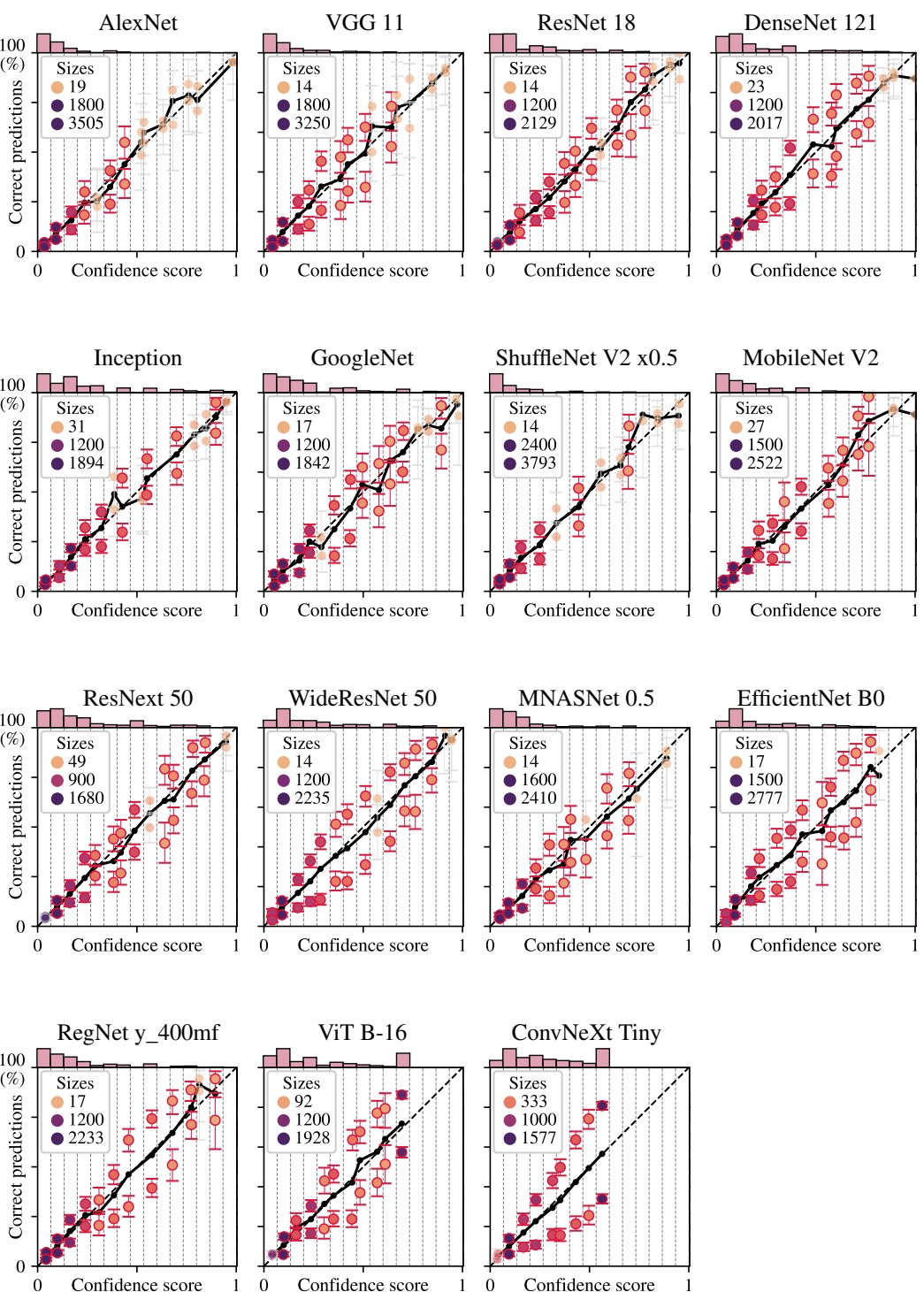

Figure 18: **Vision**: Fraction of correct predictions versus confidence score of predicted class ($\max_k S_k$) on ImageNet-R for small versions of pre-trained networks, with isotonic recalibration. In each bin on confidence scores, the level set is partitioned into 2 regions with a decision stump constrained to one balanced split, with a 50-50 train-test split strategy.

### D.1.5   IMAGENET-R – ISOTONIC RECALIBRATION, BEST VERSIONS

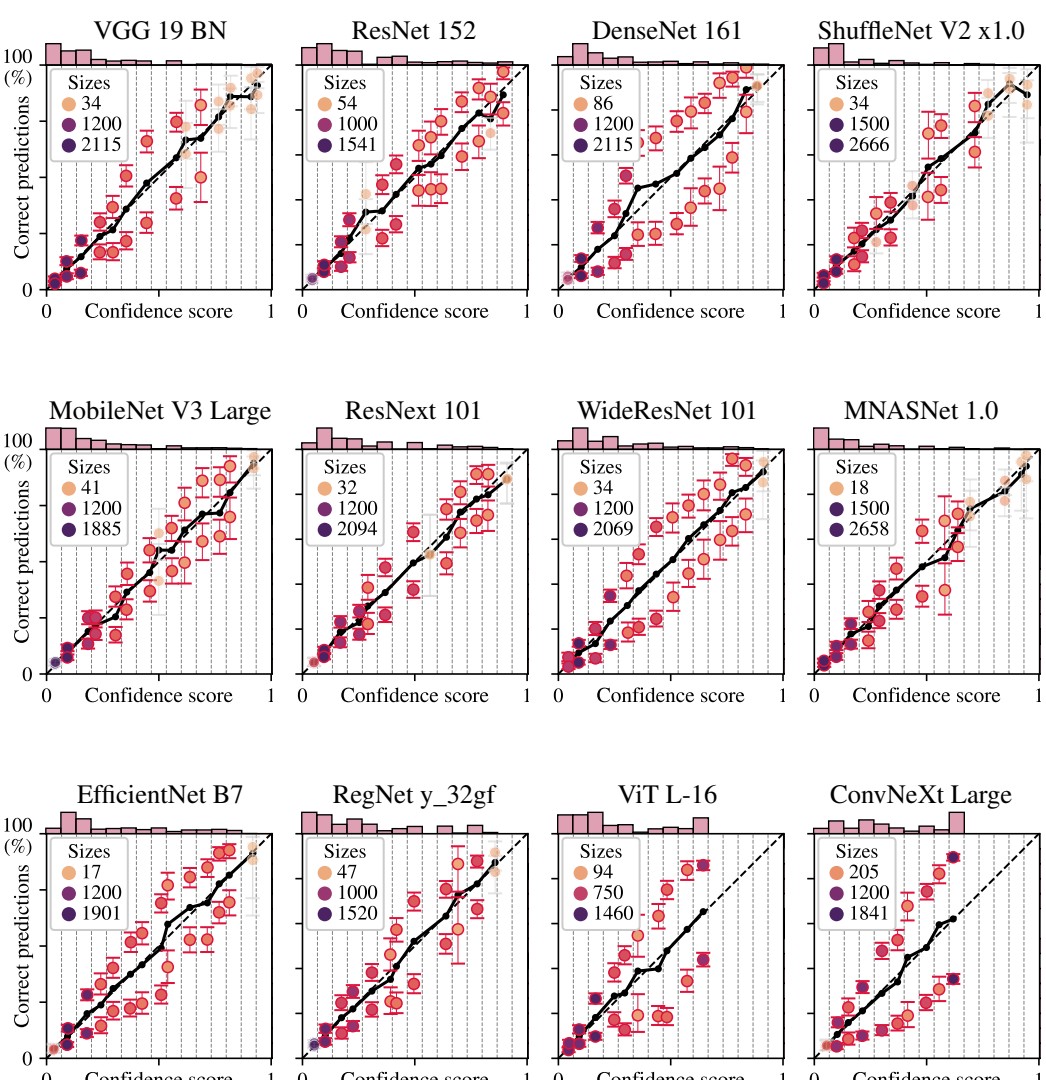

Figure 19: **Vision**: Fraction of correct predictions versus confidence score of predicted class ($\max_k S_k$) on ImageNet-R for best versions of pre-trained networks, with isotonic recalibration. In each bin on confidence scores, the level set is partitioned into 2 regions with a decision stump constrained to one balanced split, with a 50-50 train-test split strategy.

### D.2.1   IMAGENET-C – NO POST-HOC RECALIBRATION, SMALL VERSIONS

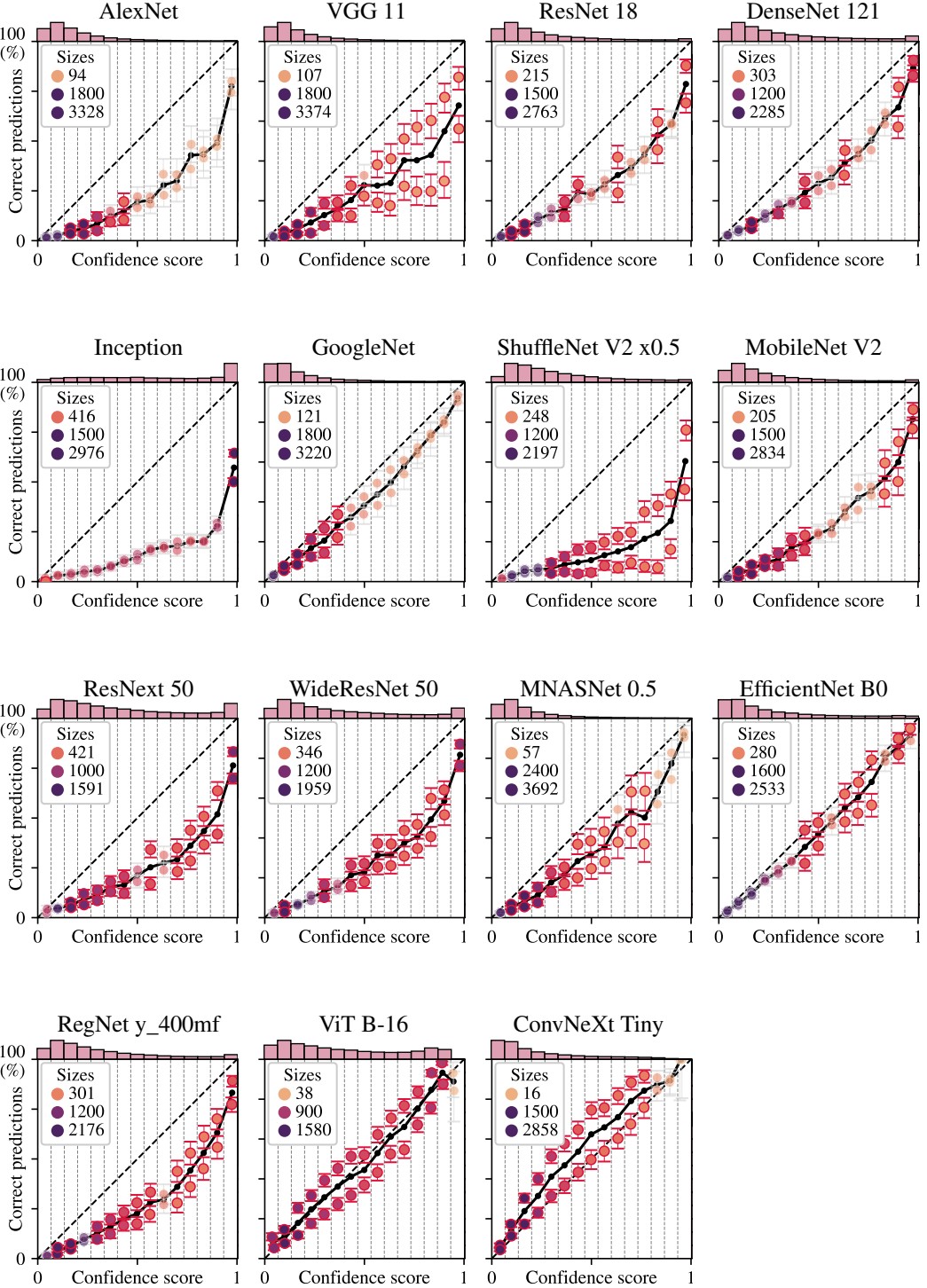

Figure 20: **Vision**: Fraction of correct predictions versus confidence score of predicted class ($\max_k S_k$) on ImageNet-C for small versions of pre-trained networks, without post-hoc recalibration. In each bin on confidence scores, the level set is partitioned into 2 regions with a decision stump constrained to one balanced split, with a 50-50 train-test split strategy.

### D.2.2  IMAGENET-C – NO POST-HOC RECALIBRATION, BEST VERSIONS

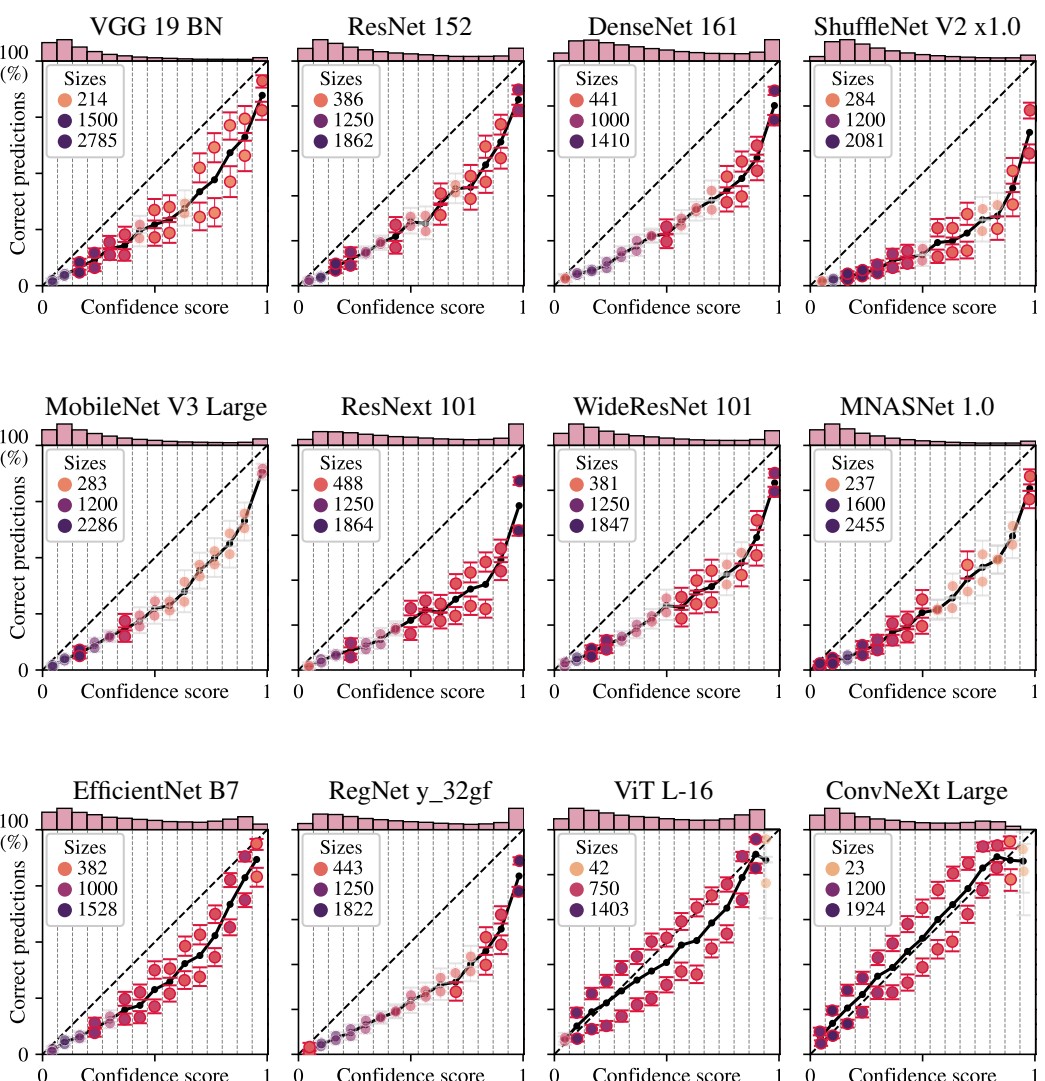

Figure 21: **Vision**: Fraction of correct predictions versus confidence score of predicted class ($\max_k S_k$) on ImageNet-C for best versions of pre-trained networks, without post-hoc recalibration. In each bin on confidence scores, the level set is partitioned into 2 regions with a decision stump constrained to one balanced split, with a 50-50 train-test split strategy.

### D.2.3  IMAGENET-C – ISOTONIC RECALIBRATION, SMALL VERSIONS

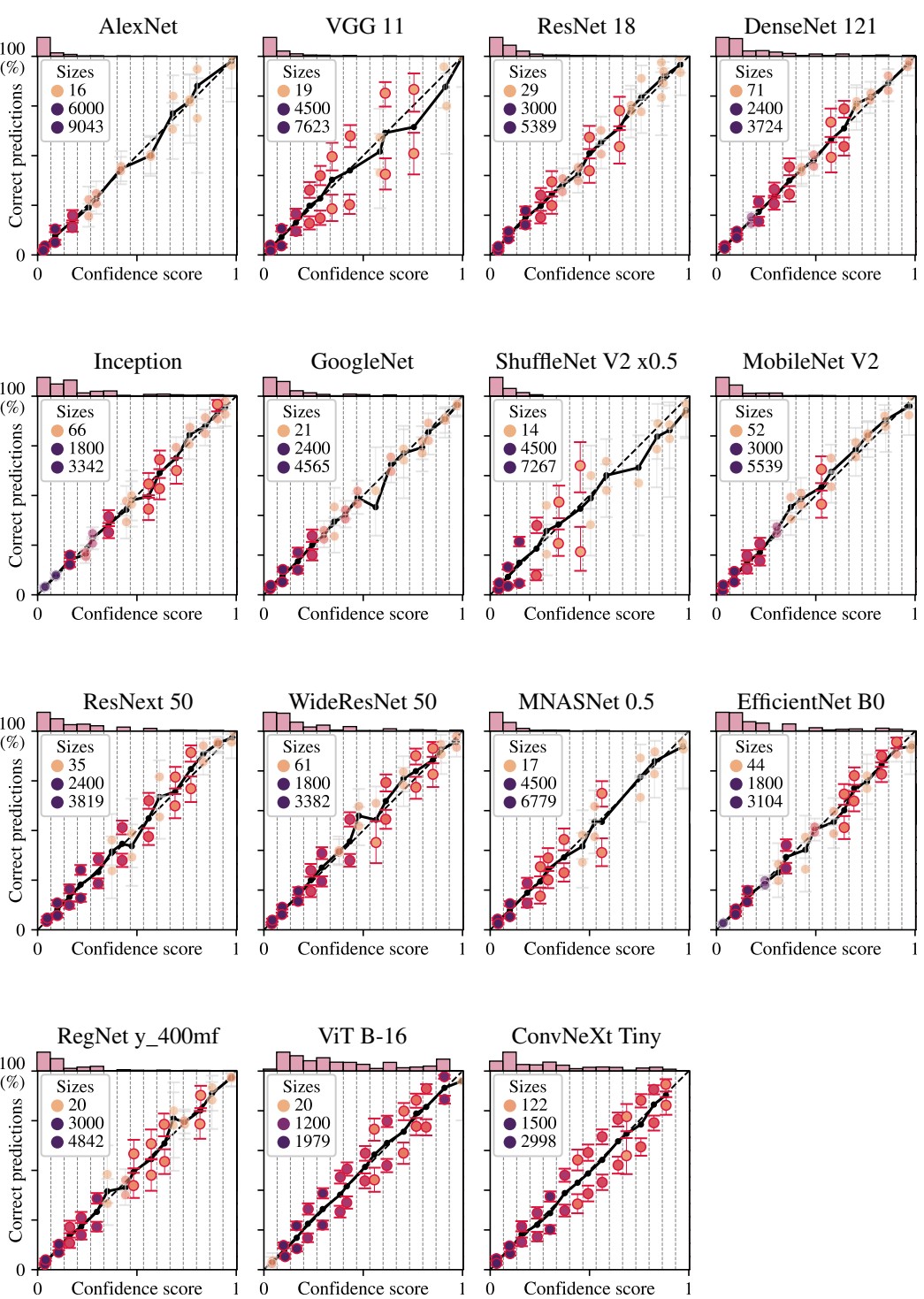

Figure 22: **Vision**: Fraction of correct predictions versus confidence score of predicted class ($\max_k S_k$) on ImageNet-C for small versions of pre-trained networks, with isotonic recalibration. In each bin on confidence scores, the level set is partitioned into 2 regions with a decision stump constrained to one balanced split, with a 50-50 train-test split strategy.

### D.2.4  IMAGENET-C – ISOTONIC RECALIBRATION, BEST VERSIONS

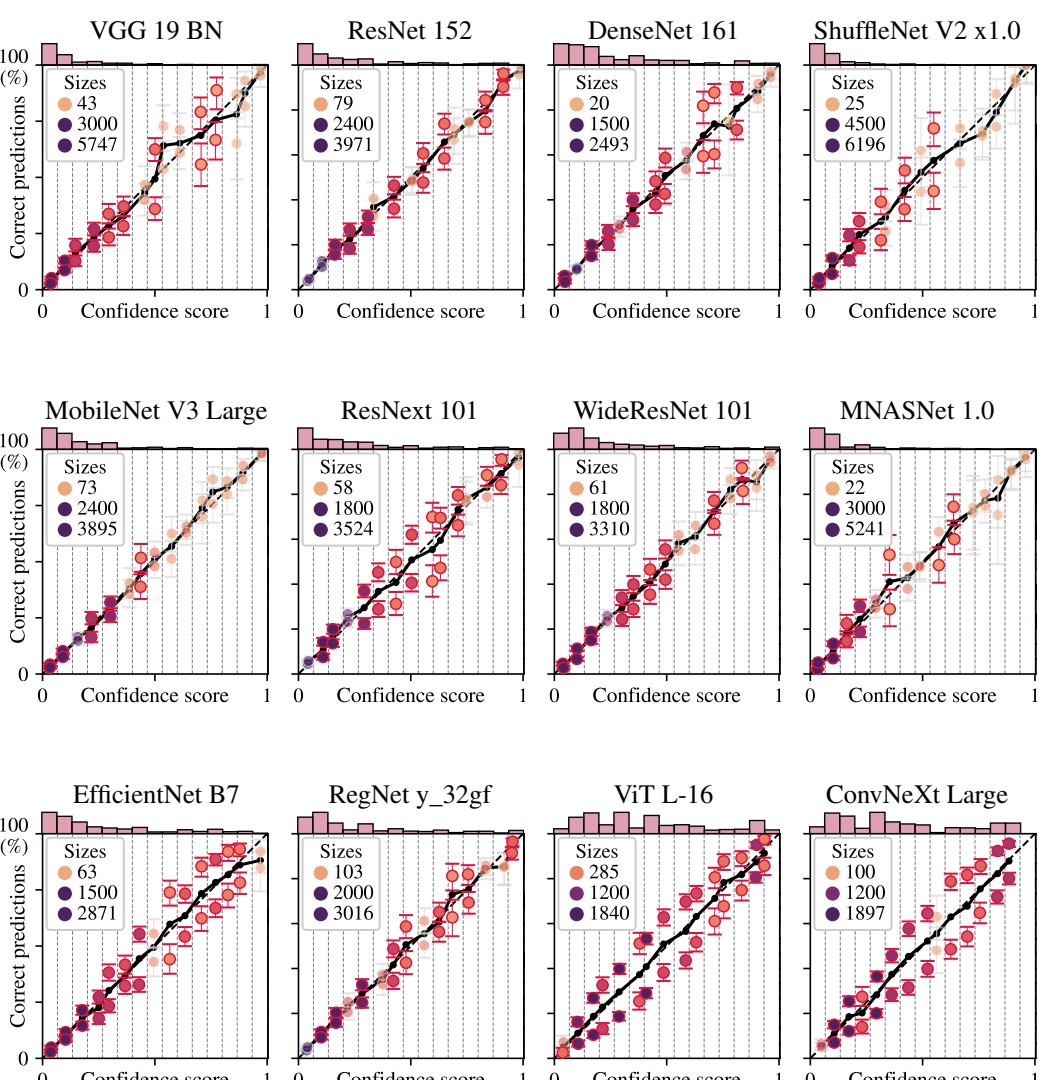

Figure 23: **Vision**: Fraction of correct predictions versus confidence score of predicted class ($\max_k S_k$) on ImageNet-C for best versions of pre-trained networks, with isotonic recalibration. In each bin on confidence scores, the level set is partitioned into 2 regions with a decision stump constrained to one balanced split, with a 50-50 train-test split strategy.

### D.3.1 IMAGENET-1K – NO POST-HOC RECALIBRATION, SMALL VERSIONS

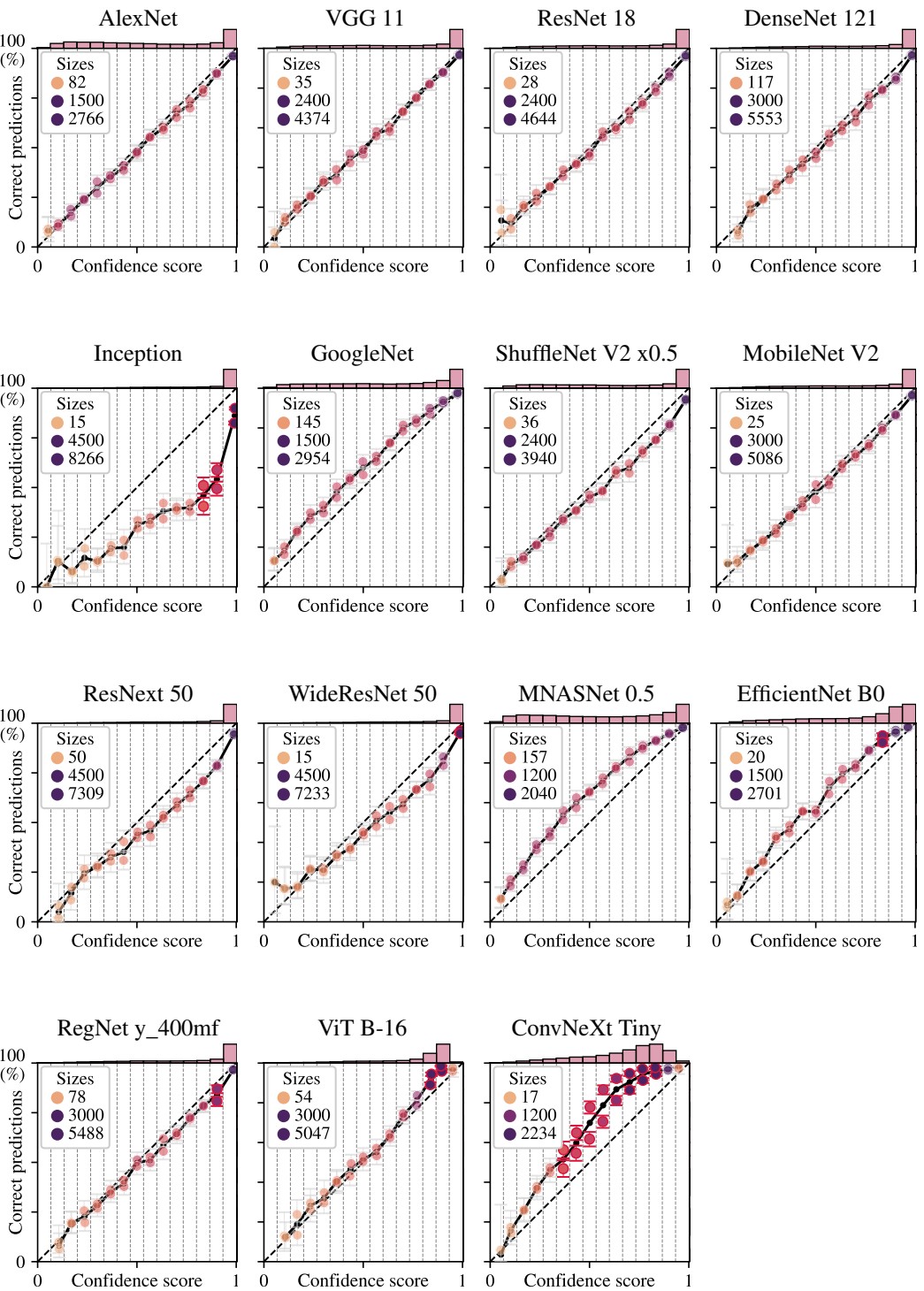

Figure 24: **Vision**: Fraction of correct predictions versus confidence score of predicted class ($\max_k S_k$) on ImageNet-1K (validation set) for small versions of pre-trained networks, without post-hoc recalibration. In each bin on confidence scores, the level set is partitioned into 2 regions with a decision stump constrained to one balanced split, with a 50-50 train-test split strategy.

### D.3.2 IMAGENET-1K – NO POST-HOC RECALIBRATION, BEST VERSIONS

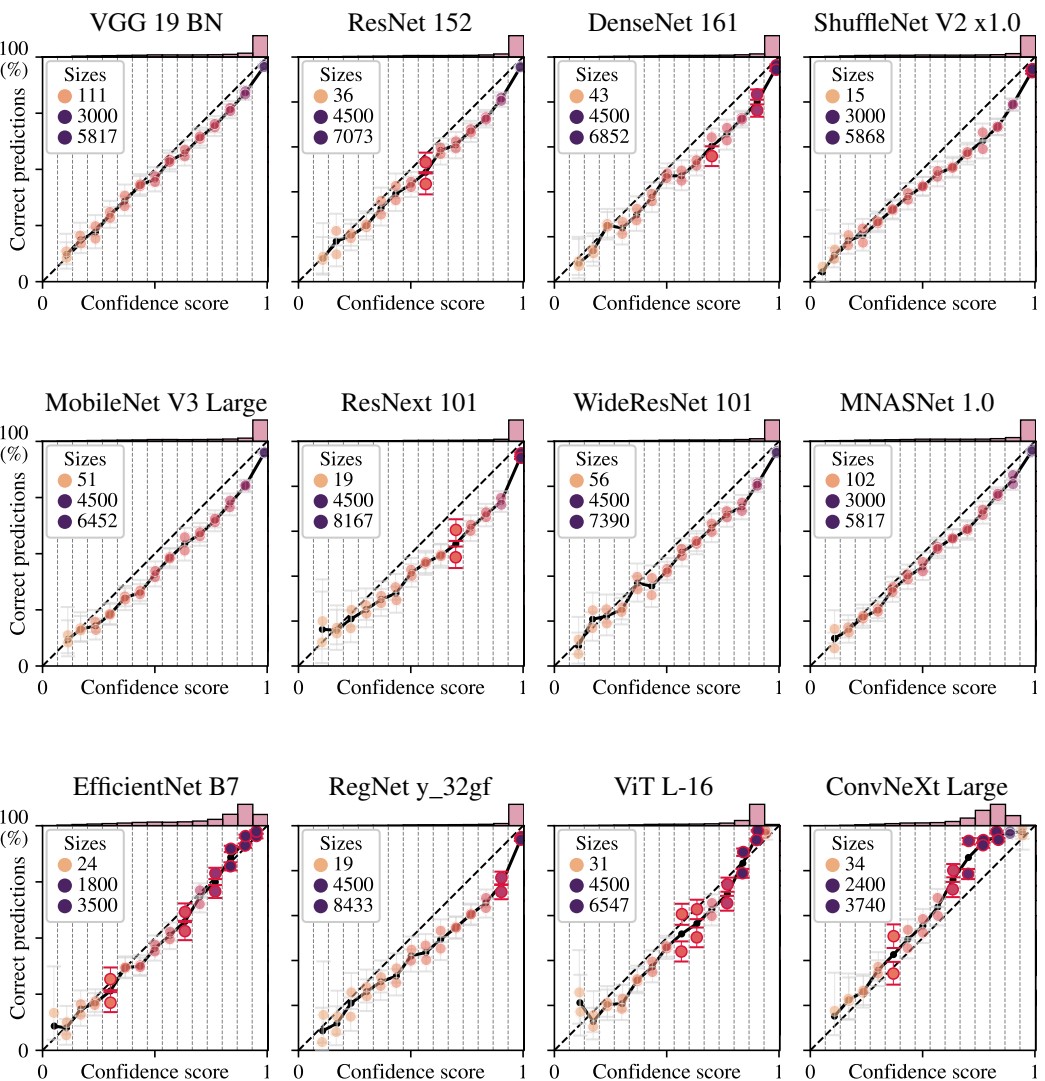

Figure 25: **Vision**: Fraction of correct predictions versus confidence score of predicted class ($\max_k S_k$) on ImageNet-1K (validation set) for best versions of pre-trained networks, without post-hoc recalibration. In each bin on confidence scores, the level set is partitioned into 2 regions with a decision stump constrained to one balanced split, with a 50-50 train-test split strategy.

### D.3.3  ImageNet-1K – Isotonic recalibration, small versions

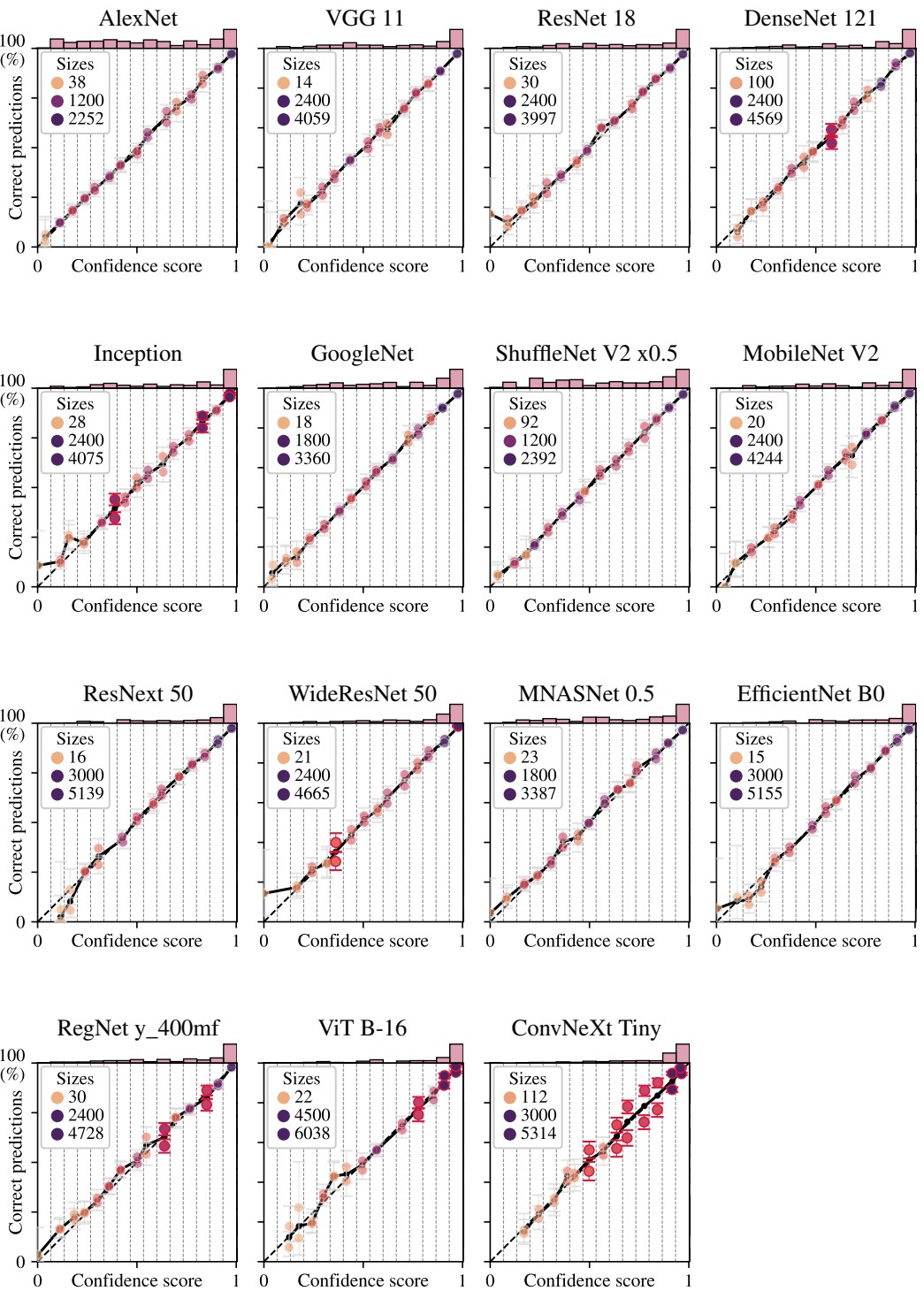

Figure 26: **Vision**: Fraction of correct predictions versus confidence score of predicted class ($\max_k S_k$) on ImageNet-1K (validation set) for small versions of pre-trained networks, with isotonic recalibration. In each bin on confidence scores, the level set is partitioned into 2 regions with a decision stump constrained to one balanced split, with a 50-50 train-test split strategy.

### D.3.4 IMAGENET-1K – ISOTONIC RECALIBRATION, BEST VERSIONS

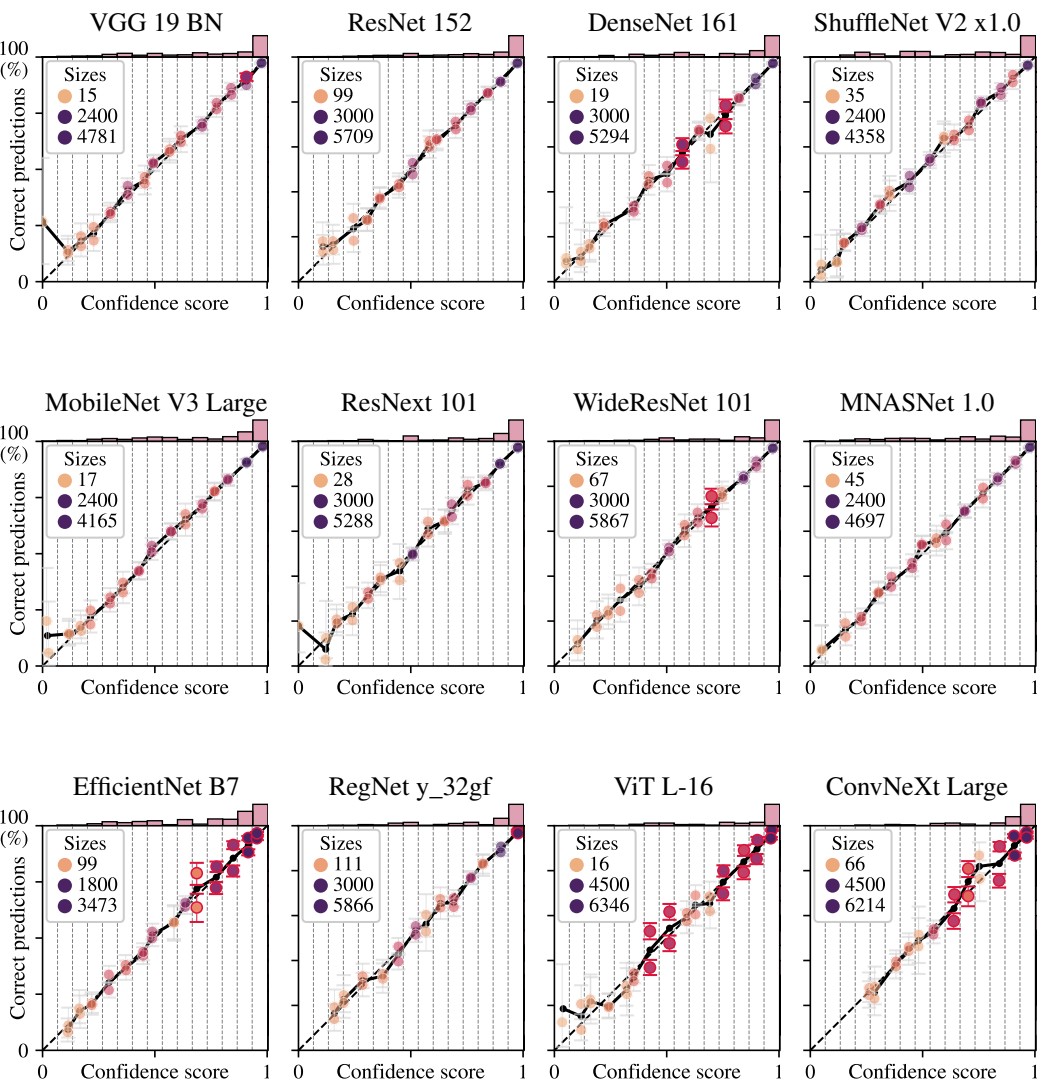

Figure 27: **Vision**: Fraction of correct predictions versus confidence score of predicted class ($\max_k S_k$) on ImageNet-1K (validation set) for best versions of pre-trained networks, with isotonic recalibration. In each bin on confidence scores, the level set is partitioned into 2 regions with a decision stump constrained to one balanced split, with a 50-50 train-test split strategy.

