# OpenReview forum: "Beyond calibration: estimating the grouping loss of modern neural networks"
_ICLR.cc/2023/Conference — ICLR 2023 poster_

### Official Review · Reviewer_sCnp · 2022-10-21

**Confidence:** 3
**Correctness:** 4
**Technical Novelty And Significance:** 4
**Empirical Novelty And Significance:** 4
**Recommendation:** 8

**Clarity, Quality, Novelty And Reproducibility:**

## Clarity

The text is very clear aside from a few small issues highlighted above.

## Quality

The quality is high.

## Novelty

To my knowledge, the work is sufficiently novel.

**Strength And Weaknesses:**

## Strengths

- The method is rigorously defined
- The results are interesting and highlight an exact problem and provide a possible solution to the shortcomings of ECE


## Weaknesses

- I am confused about the actual application of Proposition 4.3. $\text{GL}_{\text{explained}}$ will have an entry for every bin and every class, right? Then how are the figures in Figure 6 calculated on a multiclass problem like ImageNet-R? If only the top level prediction is considered, as is stated, then doesn't this add more uncertainty into the whole grouping since each class has a differing amount of irreducible uncertainty will all other classes? I understand the argument about small numbers of test instances in that particular dataset, but would it be possible to compare top-level grouping loss with per class grouping loss on a smaller dataset which has more samples, like CIFAR-10? I am curious how GL will behave when inter-class relationships are considered.

- To clarify the application of proposition 4.3 to a multiclass problem, would it be correct to characterize the sum of $\hat{\mu}_j$ in a multiclass problem as "count the number of times the actual class of region $j$ is the predicted class from the classifier?"

- Figure 7 does not quantify what happens to the grouping loss before and after isotonic recalibration. I would be very interested to see the information in figure 7 in the form of a table to see the actual differences in the numbers.

- Isotonic calibration performance is reported, but the most popular form of recalibration (temperature scaling) is not reported. Does temperature scaling perform any different than isotonic calibration?

- Do the authors have any intuition as to why the NLP tasks have a drastically lower grouping loss as compared to the vision tasks?

## Minor

- The end of the counterexample paragraph on page 2 refers to the grouping loss without defining what it is, which leads to confusion.

- NLP Section: "However, we found no evidence in the in-distribution one." This sentence is not clear. No evidence of what? What is the in-distribution 'one?'

- NLP Section: "This suggests again that our of distribution settings lead to stronger heterogeneity, and thus grouping loss." I find this sentence confusing. Why 'again?' was this brought up before in the text? Why would stronger heterogeneity cause larger grouping loss? It might seem intuitive that this is the case, but what is the reason that heterogeneity would cause higher grouping loss?

- Conclusion: "captures well the grouping loss." --> captures the grouping loss well.

**Summary Of The Paper:**

The authors offer a method of estimating the grouping loss, which fills in some shortcomings of common measures of network calibration. They then use their new metric to measure the grouping loss of a variety of different architectures and settings.

**Summary Of The Review:**

Overall, I think this work highlights and analyzes a known problem with ECE and recalibration methods in general. This is an important topic, and I think the knowledge here can be applied in a number of future works.

---

> ### Author Response · Authors · 2022-11-17
> **Author's response to Reviewer sCnp (Part 1/2)**
>
> We thank the reviewer for the enthusiastic comments. Please find our answers to the points raised below.
>
> ### Application of Proposition 4.3
> In the vision experiments, the top label prediction is considered. This results in a binary problem for which we can plot the grouping diagrams of Figure 6.
>
> We understand that the reviewer points to the following interesting fact: classwise miscalibration transforms into top-label grouping loss. In other words, having classes for which the model is either over or under-confident corresponds to miscalibration in the classwise setting, but grouping loss in the top-label setting. When it is possible, we believe that classwise definitions should be preferred to top-label as they give more information. However when it is not possible (not enough samples per class), top-label is considered. In this case, evaluating the grouping has the advantage of spotting potential over or under confidence per class, to which top-label calibration is blind.
>
> > Would it be possible to compare top-level grouping loss with per class grouping loss on a smaller dataset which has more samples, like CIFAR-10?
>
> It would indeed be interesting to test our method in the multiclass setting with the classwise definition of calibration. In our preliminary experiments, we tried CIFAR-10 both in the top-label and classwise settings. However, since it is an easy problem, all classes had either 0 or 1 confidence scores for the networks we tried. Hence there was no grouping loss (the grouping loss is upper bounded by $C(1-C)$. If all the mass is on $C=0$ or $C=1$, then $\\mathrm{GL}=0$) both in the classwise and top-label settings. ImageNet does not have enough samples per class to test the class-wise setting. The challenge is to find a task that has enough samples per class and is not too easy. It would be an interesting future work.
>
> The reviewer asks whether, in the application of Proposition 4.3 to a multiclass problem, the sum of $\\hat{\\mu}\_j$ can be characterized as "count the number of times the actual class of region $j$ is the predicted class from the classifier?". In the top-label setting, this is indeed the case. In the classwise setting however,  $\\hat{\\mu}\_j^{(s, k)}$ is the count of samples of true class $k$ in the region $j$ of the partition for the level set $s$. This difference comes from the definition of the labels in each case: ground truth class in classwise, and whether the class is well predicted in top-label.
>
> ### Estimator values
> > I would be very interested to see the information in figure 7 in the form of a table to see the actual differences in the numbers.
>
> Displaying Figure 7 as a table gives:
> | Network | $\\widehat{\\mathrm{CL}}$   | $\\widehat{\\mathrm{CL}}'$   | $\\widehat{\\mathrm{GL}}\_{\\mathrm{LB}}$   | $\\widehat{\\mathrm{GL}}\_{\\mathrm{LB}}'$   | Accuracy $\\uparrow$ (\\%)   |
> |:--|:--|:---|:----|:---|:----|
> | ViT B-16| 0.044| 0.000 | 0.019| 0.021 | 30.1 |
> | ConvNeXt Tiny   | 0.017| 0.000 | 0.036| 0.039 | 29.0 |
> | Inception   | 0.200| 0.000 | 0.006| 0.009 | 27.3 |
> | Wide ResNet-50  | 0.082| 0.000 | 0.020| 0.019 | 25.9 |
> | ResNeXt-50  | 0.091| 0.000 | 0.013| 0.013 | 25.1 |
> | EfficientNet-B0 | 0.028| 0.000 | 0.015| 0.018 | 24.1 |
> | DenseNet-121| 0.059| 0.000 | 0.013| 0.010 | 23.9 |
> | GoogLeNet   | 0.009| 0.001 | 0.011| 0.011 | 23.3 |
> | RegNet y\_400mf | 0.084| 0.000 | 0.011| 0.010 | 21.0 |
> | ResNet-18   | 0.057| 0.000 | 0.010| 0.009 | 20.4 |
> | MobileNet V2| 0.070| 0.000 | 0.007| 0.009 | 18.9 |
> | MNASNet 0.5 | 0.017| 0.000 | 0.005| 0.007 | 15.9 |
> | VGG-11  | 0.061| 0.000 | 0.007| 0.007 | 15.8 |
> | ShuffleNet 0.5  | 0.072| 0.000 | 0.004| 0.003 | 14.6 |
> | AlexNet | 0.063| 0.000 | 0.003| 0.003 | 12.6 |
>
> ("$'$" means after recalibration).
> We observe that isotonic recalibration performs well and barely changes the grouping loss. We added this table to Appendix D as Table 1. We also added the similar tables for Figure 14 as Table 2.
>
> ### On temperature scaling
> A perfect calibration ($\\hat{C} = C$) leaves the grouping loss unchanged (see newly-added Lemma C.5 of Appendix C.9.3). Good recalibration techniques will hence have negligible effect on the grouping loss. In our experiments, isotonic already recalibrates almost perfectly the confidence scores, (and as expected leaves the grouping loss unchanged).
>
> ### On the NLP experiment
> > Do the authors have any intuition as to why the NLP tasks have a drastically lower grouping loss as compared to the vision tasks?
>
> Actually the grouping loss found in NLP with the same settings as in vision (decision tree constrained to a number of regions per bin equal to n\_samples\_per\_bin/30) is similar or higher than the ones for the vision task. We found $\\widehat{\\mathrm{GL}}\_{\\mathrm{LB}} = 0.02$ for the in-distribution setting and $\\widehat{\\mathrm{GL}}\_{\\mathrm{LB}} = 0.08$ for the out-of-distribution setting (compared to $\\widehat{\\mathrm{GL}}\_{\\mathrm{LB}}$ ranging from $0.003$ to $0.036$ for the vision task).

---

> > ### Author Response · Authors · 2022-11-17
> > **Author's response to Reviewer sCnp (Part 2/2)**
> >
> > ### On minor comments
> > > The end of the counterexample paragraph on page 2 refers to the grouping loss without defining what it is, which leads to confusion.
> >
> > We changed the manuscript to avoid this confusion.
> >
> > > NLP Section: "However, we found no evidence in the in-distribution one." This sentence is not clear. No evidence of what? What is the in-distribution 'one?'
> >
> > In the NLP experiment, we compare in-distribution settings (same data distribution for training and testing) with out-of-distribution settings (data distribution shifts between training and testing). In the out-of-distribution setting, we found evidence of grouping loss with our estimator. However, we found no evidence of grouping loss in the in-distribution setting. We rephrased the sentence to remove ambiguity.
> >
> >
> > > NLP Section: "This suggests again that our of distribution settings lead to stronger heterogeneity, and thus grouping loss.”. I find this sentence confusing. Why would stronger heterogeneity cause larger grouping loss?
> >
> > Indeed, this sentence is redundant. By stronger heterogeneity we meant stronger heterogeneity of the classifier error and hence grouping loss. We rephrased the sentence.
> >
> > > Why 'again?' was this brought up before in the text?
> >
> > Yes, it was an observation of the vision experiment p8: “We observe the same effects on ImageNet-C, but little or none on ImageNet-1K. This suggests that stronger grouping loss arises in out-of-distribution settings.”.
> >
> >
> > > Conclusion: "captures well the grouping loss." --> captures the grouping loss well.
> >
> > Fixed, thank you.

---

> > > ### Comment · Reviewer_sCnp · 2022-11-24
> > > **One more question**
> > >
> > > Thank you for the responses. I just have one question regarding your response.
> > >
> > > > > Do the authors have any intuition as to why the NLP tasks have a drastically lower grouping loss as compared to the vision tasks?
> > >
> > > > Actually the grouping loss found in NLP with the same settings as in vision (decision tree constrained to a number of regions per bin equal to n_samples_per_bin/30) is similar or higher than the ones for the vision task. We found
> > >  for the in-distribution setting and
> > >  for the out-of-distribution setting (compared to
> > >  ranging from  to  for the vision task).
> > >
> > > If the OOD performance is lower for vision compared to NLP, then why does the grouping look so loose in Fig. 6(d) (Vision) but it looks really tight in the NLP OOD task Fig. 8(d)?

---

> > > > ### Author Response · Authors · 2022-11-29
> > > > **Answer to Reviewer sCnp's question**
> > > >
> > > > We thank the reviewer for raising this good point. Indeed, there is a mismatch between the values of $\\widehat{\\mathrm{GL}}\_{\\mathrm{LB}}$ given in our previous answer and the diagram in Figure 8a-d: the values (0.02 and 0.08) correspond to a top-label setting for NLP (def. 3.3), whereas Figures 8a-d are in the binary calibration setting (def. 3.1 and 3.2). The y-axis in Figures 8a-d gives the fraction of positives (so binary calibration setting), rather than the fraction of correct predictions (top-label setting) as in Figures 6a-d for vision. In the setting of Figure 8 (binary calibration setting), the estimated values are:
> > > >
> > > > | Setting             | $\\widehat{\\mathrm{CL}}$   | $\\widehat{\\mathrm{CL}}'$   | $\\widehat{\\mathrm{GL}}\_{\\mathrm{LB}}$   | $\\widehat{\\mathrm{GL}}\_{\\mathrm{LB}}'$   | Accuracy $\\uparrow$ (\\%)   |
> > > > |:--------------------|:--------------------------|:---------------------------|:----------------------------------------|:-----------------------------------------|:---------------------------|
> > > > | in-distribution     | 0.026                     | 0.000                      | 0.000                                   | 0.000                                    | 88.2                       |
> > > > | out-of-distribution | 0.091                     | 0.000                      | 0.015                                   | 0.015                                    | 71.1                       |
> > > >
> > > > We observe no grouping loss in the in-distribution setting, while the grouping loss in the out of distribution setting is $\\widehat{\\mathrm{GL}}\_{\\mathrm{LB}} = 0.015$ both before and after recalibration.
> > > > To compare with the vision experiment, a $\\widehat{\\mathrm{GL}}\_{\\mathrm{LB}}$ of 0.015 corresponds to the one estimated for EfficientNet-B0 (Table 1 p31). Corresponding grouping diagrams (Figures 15 and 18) are comparable to the ones obtained in Figures 8b and 8d.

---

### Official Review · Reviewer_LCym · 2022-10-23

**Confidence:** 2
**Correctness:** 4
**Technical Novelty And Significance:** 3
**Empirical Novelty And Significance:** 3
**Recommendation:** 8

**Clarity, Quality, Novelty And Reproducibility:**

The paper is well-written. I think it offers an interesting insight into calibration and the limitations of using ECE alone, even when also reporting accuracy (as shown in Figure 1). In terms of reproducibility, the paper should include more details about how decision trees are used to partition the space since this is a crucial part of the experiments.

**Strength And Weaknesses:**

The paper brings insight into calibration and supports its arguments theoretically. The authors conduct experiments on several architecture families in both language and vision in real-world datasets and also present experiments on synthetic data.

But, the primary weakness in my opinion lies in the motivation behind this work and I would appreciate it if the authors clarify the following point in their rebuttal. First, using a proper scoring rule, such as the log-loss and the Brier score, would automatically capture both calibration and group loss at the same time and they can be estimated easily. Is there a reason for practitioners to report a calibration error and a group loss, when they can report a proper score. My main takeaway from the paper is that one should not rely on calibration alone, such as using ECE, but rather use proper scoring rules because they capture group loss as well, which seems to be a simple and accurate message to take away. But, I don’t see the value of trying to calculate a lower bound on the group-loss term, especially given that the results on the simulated data in Figure 4 are not encouraging.

Second, I haven’t yet figured out how the regions or partitions are identified using decision trees in Section 4. What is the input to the decision tree? For example, in vision, are those raw images? I think it would be really useful to provide examples of images for each group.


Some minor comments:
In Lemma 4.1, “negative entropy” may not be the right term since the score rule can be something else.
In Figure 4, why is $GL_{LB}$ not a lower bound when the number of bins is small?


**Summary Of The Paper:**

The paper focuses on a decomposition of strictly proper scoring rules that reveals three terms: calibration loss, group loss, and an irreducible term. Intuitively, the group loss term (GL) deviates from the aggregate measures used in calibration; e.g. if a model is systematically over-confident on a subgroup and under-confident on another, the overall calibration can be small but GL is large. The authors argue that evaluating GL is important and they present an estimator that provides a lower bound to GL. They conduct experiments on vision and language tasks, which suggest that models continue to have a large group loss even after post-hoc calibration.


**Summary Of The Review:**

The paper is well-written and offers an interesting insight into calibration. All arguments are supported by experiments and proofs. I find the experimental results quite interesting. My only concerns are: (1) the practical implications are unclear yet since one can simply use proper scoring rules, and (2) more details should be provided about the experiments as I mention above.


======  Post rebuttal ======

Thank you for confirming my understanding and for the clarification.

I see now your argument of why quantifying the group loss can be useful. Since a proper scoring rule includes the irreducible term, it might be worthwhile instead to look into the sum of calibration and group loss. As you mention in the revised draft,  a classifier with a Brier score of 0.15 could have optimal probabilities (irreducible loss close to 0.15) or poor ones (irreducible loss close to 0).

I think this is a useful message and the proposed method using decision trees is simple and easy to apply. I have updated my scores.

---

> ### Author Response · Authors · 2022-11-17
> **Author's response to Reviewer LCym**
>
> We thank the reviewer for the encouraging comments. Please find our answers to the points raised below.
>
> ### On motivation and practical implications
>
> We addressed the point on motivation in a general comment “Clarifying motivation”. For the general comment, please refer to:
> https://openreview.net/forum?id=6w1k-IixnL8&noteId=0WVZk2IYVRb
>
> For the specific part on why we cannot use the proper scoring rule for this work, please refer to:
> https://openreview.net/forum?id=6w1k-IixnL8&noteId=Uagyji9cDVL
>
> ### On the experiments
>
> > I haven’t yet figured out how the regions or partitions are identified using decision trees in Section 4.
>
> With regards to the input of the decision tree, in the experiments (vision and NLP), we worked in the feature space of the penultimate layer of each architecture instead of the original feature space (images or text). The inputs are thus the data $X$ (features in the penultimate layer) and the binary outcome $Y$. Decision trees are trained to predict $Y$ from $X$. The regions are then taken to be the leaves of the fitted tree, as they form a partition of the space. $\\mathrm{GL}\_{\\mathrm{LB}}$ is then estimated on new samples with this partition fixed.
> The full experimental details for vision and NLP regarding partitioning are given in Appendix D, paragraph “Detailed experimental method”.
> We clarified in the main text (end of section 4) that for vision and NLP tasks, we worked in the penultimate layer
>
> > Some minor comments: In Lemma 4.1, “negative entropy” may not be the right term since the score rule can be something else.
>
> In the scoring rule theory, the entropy of any scoring rule $\\phi$ is defined as $p \\mapsto s\_{\\phi}(p, p)$, where $s\_{\\phi}$ is the expected score of rule $\\phi$. Hence we defined the negative entropy for any scoring rule as $p \\mapsto - s\_{\\phi}(p, p)$. For the Log-loss, it corresponds to the classical information entropy, but for example for the Brier score, it corresponds to the Gini index. To avoid any misunderstanding, we added a footnote in Lemma 4.1.
>
> > In Figure 4, why is $\\mathrm{GL}\_{\\mathrm{LB}}$ not a lower bound when the number of bins is small?
>
> The lower-bound slightly exceeds the true grouping loss when the number of bins is equal to 1 or 2. With only one or two bins, the binned scores $S\_B$ are very coarse approximations of the scores $S$, thus $\\mathrm{GL}\_{induced}$ is very large, and notably orders of magnitude larger than the true grouping loss. In this setting, $\\mathrm{GL}\_{induced}$ is too large to be well corrected, however no-one uses one or two bins to compute calibration.

---

> > ### Comment · Reviewer_LCym · 2022-11-17
> > **Quick question**
> >
> > Thanks for the response. So, if I understand this correctly, there is still some information in the pre-logit layer about the target $Y$ that is not captured by the classifier's head. Correct me please if I'm wrong: you build  a decision tree classifier on the pre-logit features to predict the target $Y$ directly. Then, you observe that the neural network either over-estimate or under-estimate its confidence in each leaf of the decision tree.
> >
> > Can you verify that the neural network is trained until convergence (i.e. achieves zero training error on the training split)? I couldn't find in the paper any details about the training setup (e.g. learning rate schedule, number of epochs, etc). It would be useful to include those anyway for reproducibility. Also, just to verify that your observations are not due to your training procedure, I would suggest doing the same analysis on ImageNet-pretrained models.

---

> > > ### Author Response · Authors · 2022-11-17
> > > **Answer to Reviewer LCym's question**
> > >
> > > Thank you for the quick feedback.
> > >
> > > First, the reviewer’s understanding about what is happening in the pre-logit layer is correct.
> > >
> > > Then, for all our experiments, we actually used pre-trained models. For vision architectures, we used the pre-trained weights available on PyTorch. According to the documentation, all the networks were pre-trained on the training set of ImageNet-1K. For NLP, we used the pre-trained weights of BART [available on HuggingFace](https://huggingface.co/joeddav/bart-large-mnli-yahoo-answers). According to the documentation, BART was pre-trained on the MNLI dataset. As the reviewer noticed, the goal was precisely to remove any implication from the training procedure into our results.
> > >
> > > This information is indicated at the beginning of the “Vision” and “NLP” paragraphs page 7 and 9 and the full experimental setting is given in Appendix D (Vision) and E (NLP). Since this is an important point of our analysis, we have better emphasized that the networks are pre-trained in the manuscript following the reviewer’s remark.
> > >
> > > > you build a decision tree classifier on the pre-logit features to predict the target $Y$ directly. Then, you observe that the neural network either over-estimate or under-estimate its confidence in each leaf of the decision tree.
> > >
> > > The reviewer’s summary of the use of the decision-tree classifier to observe over or under estimation of confidence in the leafs of the decision tree is accurate.
> > >
> > > Conceptually, we are in the model validation setting (see [the general comment on motivation](https://openreview.net/forum?id=6w1k-IixnL8&noteId=0WVZk2IYVRb) for more details). A model has been trained on some data, possibly different from the task at hand, and one wants to investigate whether the model is safe to use for the task (i.e. good estimated probabilities beyond good accuracy). Often, retraining the network is not an option: too costly, too large, black-box, etc (e.g. large models such as GPT-3). In this case, our estimator is able to assess if the model can be used or not for the new task.
> > >
> > > Note also that our estimator is not tied to this partitioning method. Any partition of the feature space is suitable. We used the decision tree method to find a smart partition likely to better capture the grouping loss.
> > >
> > > We would be glad to answer any further questions if needed.

---

### Official Review · Reviewer_rpCw · 2022-10-25

**Confidence:** 5
**Correctness:** 4
**Technical Novelty And Significance:** 2
**Empirical Novelty And Significance:** 2
**Recommendation:** 3

**Clarity, Quality, Novelty And Reproducibility:**

DETAILS ABOUT MAIN CONCERN:

1. I am not convinced from reading this paper that measuring the grouping loss is interesting.

1.a. It would be helpful to highlight specific examples of why that's the case. The experiments report the group loss of different models. Why is this information interesting? How is it actionable? Also, why do we even care about making the group loss small? If the model is initially over-confident, the group loss will increase after calibration, but that's a good thing, it's just correcting for overconfidence in the model. A discussion of this would be helpful. One way in which I can imagine why estimating the group loss may be useful is that we may potentially optimize this loss directly (not sure if this will lead to anything interesting, but it's an idea).

1.b. In particular, we know that a proper score decomposes as: proper_score = calibration + group_loss + irreducible_term. If we can measure the proper_score and calibration, then we can compute (group_loss + irreducible_term), where irreducible_term is constant across models. Why is this not sufficient? In the experiments section, the authors compare multiple models in terms of their group loss. It seems like we should be able to get a relative ranking by simply using (proper_score - calibration) = (group_loss + irreducible_term), which should be simpler to compute and wouldn't require complex math.

DETAILS ABOUT OTHER CONCERNS:

2. The experiments seem like they could be improved. There is only one relatively limited simulation that explores whether the bound is sufficiently tight. If I were to trust this, I would probably need more experiments. Also, the experiments on NLP simply report the group loss, but it's unclear to me how this connects to the main estimator. Also, there are not baselines, and I'm not sure what point that experiment is trying to make (except that the group loss is non-zero).

3. Some of the claims sound off. In particular, the claim that "a common confusion is to mistake confidence scores of a calibrated classifier with true posterior probabilities and think that a calibrated classifier outputs true posterior probabilities" seems off to me. I've been working in this area for a while, and I can't imagine researchers believing this is true. The authors provide some quotes, but these seem either misinterpreted (i.e., the author was imprecise and his claim can be interpreted in multiple ways, e.g., [Q1]) and/or come from outside the mainstream machine learning community (top ML conferences and stats journals). This is more of a minor point, but I wouldn't claim that debunking this misunderstanding is a major contribution of the paper.


[Q1]: "This is exactly the purpose of calibration techniques, which aim to map the predicted probabilities to the true ones in order to reduce the probability distribution error of the model"

**Strength And Weaknesses:**

STRENGTHS:
* The paper provides a novel estimator for the lower bound of the group loss. This term was previously hard to measure.
* The methods used are novel, technically sound, and non-trivial. The level of technical depth required to derive the estimator is significant.
* The experiments evaluate a lot of realistic state-of-the-art models.

WEAKNESSES:
* I am not convinced from reading this paper that measuring the grouping loss is something I would like to do in practice and why this is interesting.
* It is not clear to me how the experiments support the main claims of the paper.
* Some of the claims seem off off to me (see below).

**Summary Of The Paper:**

The authors of the paper derive an estimator of a term that lower bounds the grouping loss term of a proper scoring rule. They show that this estimator is tighter than baselines and use it to analyze existing models. The derivations seem novel and correct; however, it is unclear to me why having this estimator is interesting or useful in practice.

**Summary Of The Review:**

I am not convinced the problem studied by this paper is sufficiently significant.

---

> ### Author Response · Authors · 2022-11-17
> **Authors' response to Reviewer rpCw**
>
> We thank the reviewer for the insightful comments. Please find our answers to the points raised below.
>
> ### On motivation (Point 1.a and 1.b)
> We addressed these two points in a general comment “Clarifying motivation”. Please refer to:
> https://openreview.net/forum?id=6w1k-IixnL8&noteId=0WVZk2IYVRb
>
> For the specific part on why we cannot use the proper scoring rule for this work (Point 1.b), please refer to:
> https://openreview.net/forum?id=6w1k-IixnL8&noteId=Uagyji9cDVL
>
> With regards to the change in grouping loss during recalibration, we show in the newly-added Appendix C.9.3 that calibrating with a prefect calibration map brings the calibration error to zero but leaves the grouping loss unchanged. If the calibration map is not perfect (as is the case in realistic settings), then calibration can slightly change the grouping loss by $\\mathbb{E}[\\mathbb{V}\_{h}[C|S’]]$ where $S’$ is the recalibrated version of $S$ and $C$ the perfect calibration map (newly-added Lemma C.5). However, this term will be small for any good calibration map. Our experiments confirm that in realistic settings, recalibrating does not impact the grouping loss (Figure 7, Figure 14 and newly-added Tables 1 and 2 for raw values).
>
>
> ### On the experiments (Point 2)
> In the experiments, the simulation serves two purposes: 1) to investigate the effect of the number of bins and regions, 2) to validate that, with the correction terms, the estimated grouping loss is close to the actual one.
> Our simulation brings good evidence that the method provides a tight lower bound because the simulation is both a plausible scenario and one that is not directly favorable to the contributed method. The simulation is plausible because it is in a multivariate setting where the grouping loss is due to inhomogeneities in a different direction than calibration. It is challenging for our contributed method because the probabilities vary smoothly and in a non-axis-aligned way, while our approach is based on trees.
>
> The goal of the NLP experiment is to investigate whether grouping loss occurs in NLP tasks, including in a distribution-shift setting, as pretrained models are often reused in NLP. As the purpose here is model validation rather than model selection we focused on the grouping loss, illustrated with the grouping diagrams. The main result is however that distribution shifts substantially hinders the confidence scores by increasing the grouping loss ($\\widehat{\\mathrm{GL}}\_{\\mathrm{LB}} = 0.02$ for the in-distribution setting and $\\widehat{\\mathrm{GL}}\_{\\mathrm{LB}} = 0.08$ for the out-of-distribution setting), which confirms what we observed in the vision experiment.
>
> ### On the confusion about calibration (Point 3)
> We agree with the reviewer that the discussion on calibrated scores versus true posterior probabilities is not a major contribution of this paper. We also agree that imprecise wordings (as quoted in Appendix A) can be interpreted in multiple ways. We have no doubt that the definition of calibration is absolutely clear for people working on the subject, yet the ambiguous wordings leave room for misinterpretation for unfamiliar minds. The core of our contribution is to capture some of the remaining discrepancy between the calibrated scores and the true posterior probabilities, so we felt it was important to address the misconceptions on this point.

---

> > ### Comment · Reviewer_rpCw · 2022-12-08
> > **Response**
> >
> > I acknowledge having read the response.
> >
> > The essence of the authors' first argument boils down to the need to measure the grouping loss to determine whether the model is optimal or not. First, it's not clear to me when and why I would care about determining if the model is optimal. The authors' response in the other comment is fairly generic (e.g., importance for fairness, trustworthiness... but why exactly?). Second, even if that's the goal, why is the right approach to measure the group loss rather than the epistemic loss directly? It seems like estimating the group loss is tackling a different problem. The motivation for the paper is still not clear to me. Third of all, even if we are interested in estimating the group loss, I am not convinced that the estimator is the right one. In Figure 4, the estimator seems to significantly over or under-estimate the GL with different bin sizes, and it's unclear how to tune that bin size. Also, there is only one small experiment that evaluates its sensitivity to the bin size.
> >
> > As a result, I am not ready to accept the paper. I may not be a strong detractor, but I am still not convinced that this paper merits an accept.

---

> > > ### Author Response · Authors · 2022-12-09
> > > **Authors' response (Part 1/2)**
> > >
> > > > it's not clear to me when and why I would care about determining if the model is optimal.
> > >
> > > There is a consensus that calibration is an important quantity to investigate when validating machine learning models (it is strongly discussed not only in the machine learning literature, but also in the clinical one [1]). The grouping loss, as well as model optimality, matters for pretty much the same reasons (that we described in [the general comment on motivation](https://openreview.net/forum?id=6w1k-IixnL8&noteId=0WVZk2IYVRb)). A model with non zero grouping loss (so non optimal) is miscalibrated on subgroups of individuals. Many examples illustrating why it matters exactly can be given, we give two practical examples below.
> > >
> > > **Example 1:** Impact on fairness can be seen by considering a model predicting the probability of students graduating school. Suppose the school gives scholarships for students with a high probability of graduating. As shown by Rediet Abebe in her NeurIPS talk this year, such a model (the “dropout early warning system”) systematically underestimates the probability of graduating for non-white students and displays a different calibration plot for white and non-white students, i.e., it displays grouping loss. This leads to unfairness: for similar talents, non-white students get less scholarships.
> > >
> > > **Example 2:** Consider a fraud-detection setting where each data point is a transaction of a certain amount of money, and the probability of fraud is correlated to the amounts of the transactions. Suppose we would like to estimate the cost of frauds incurred to the company, using a probabilistic fraud classifier. We can mathematically show that a calibrated classifier with optimal accuracy will lead to a biased estimation of the costs because of its grouping loss:
> > >
> > > * $X \\sim \\mathcal{N}(0, 1)$: transaction data.
> > > * $Y \\in \\{0, 1\\}$: whether the transaction is a fraud.
> > > * $Q(X) = \\begin{cases}
> > >     	20\\% & \\text{if } X < 0\\\\
> > >     	40\\% & \\text{if } X \\geq 0
> > > 	\\end{cases}$, true probability of a fraud.
> > > * $S(X) = 30\\%$, probability of a fraud estimated by classifier $S$.
> > > * $V(X) = \\begin{cases}
> > >     	1\\,000 & \\text{if } X < 0\\\\
> > >     	10\\,000 & \\text{if } X \\geq 0
> > > 	\\end{cases}$, amount of the transaction.
> > > * Decision threshold 50\%.
> > >
> > > On this example, the cost of the transaction $X$ having real outcome $Y$ is:
> > > $\mathrm{Cost}(X, Y) = \mathbb{1}_{Y = 1}V$.
> > >
> > > Hence the expected cost of transaction $X$ is $\\mathbb{E}[\\mathrm{Cost}(X, Y)|X] = QV$.
> > >
> > > Using the probabilities estimated by the classifier $S$ instead of the true ones, leads to the following bias in the estimation of the total expected cost:
> > > \\begin{align*}
> > > 	\\text{Bias} & := \\mathbb{E}[QV] - \\mathbb{E}[SV]\\\\
> > > 	& = \\mathbb{E}[(Q-S)V]\\\\
> > > 	& = \\mathbb{E}[(Q-S)V|X \\geq 0]\\mathbb{P}(X \\geq 0)
> > > 	+ \\mathbb{E}[(Q-S)V|X < 0]\\mathbb{P}(X < 0)\\\\
> > > 	& = 450
> > > \end{align*}
> > > $S$ is **calibrated** and has **optimal accuracy**, yet is **unreliable** to estimate the total expected cost **due to the grouping loss.**
> > >
> > >
> > >
> > >
> > > > why is the right approach to measure the group loss rather than the epistemic loss directly?
> > >
> > > The epistemic loss is indeed a quantity of interest; it is equal to the sum of the calibration and grouping losses ($\\text{Epistemic Loss} = \\text{Calibration Loss} + \\text{Grouping Loss}$). Calibration is also of interest by itself and many works have characterized it and studied its estimation. Here, we complement these works with an estimator for the grouping loss. Combined with previous works on calibration, it gives us an estimation of the epistemic loss. This is what we show in Figures 7 and 14: the calibration loss, the grouping loss, and their sum. Note that when the classifier is calibrated, the epistemic loss boils down to the grouping loss.
> > >
> > > > It seems like estimating the group loss is tackling a different problem
> > >
> > > Since the grouping loss is part of the epistemic loss and the calibration loss is well estimated by prior work, estimating the grouping loss is precisely tackling the problem of epistemic loss estimation.

---

> > > > ### Author Response · Authors · 2022-12-09
> > > > **Authors' response (Part 2/2)**
> > > >
> > > > > the estimator seems to significantly over or under-estimate the GL with different bin sizes, and it's unclear how to tune that bin size
> > > >
> > > > We estimate a proxy ($\\mathrm{GL}\_{\\mathrm{LB}}$) that is a **lower bound** on the grouping loss $\\mathrm{GL}$, as shown in the paper (Proposition 4.2). The under-estimation is thus expected. As explained in the introduction, the grouping loss (and similarly the epistemic loss) cannot be estimated since we do not have access to the optimal model [2], so we need to resort to proxies. The calibration loss is a tractable but coarse proxy for the epistemic loss. We propose a refined proxy of the epistemic loss that includes the calibration loss and our estimator $\\mathrm{GL}\_{\\mathrm{LB}}$ of the grouping loss.
> > > >
> > > > Concerning the over-estimation, the lower-bound slightly exceeds the true grouping loss when the number of bins is equal to 1 or 2. With only one or two bins, the binned scores $S\_B$ are very coarse approximations of the scores $S$, thus  $\\mathrm{GL}\_{induced}$ is very large, and notably orders of magnitude larger than the true grouping loss. In this setting, $\\mathrm{GL}\_{induced}$ is too large to be well corrected; however no-one uses one or two bins in practice to compute calibration. We also have bounds on the error induced by the binning that guide the choice of the number of bins (Theorem C.1 and Corollary C.1).
> > > >
> > > > > there is only one small experiment that evaluates its sensitivity to the bin size.
> > > >
> > > > There are also theoretical results on bounds depending on the number of bins (Theorem C.1 and Corollary C.1). These show that with 15 bins, the binning-induced error is sufficiently small.
> > > >
> > > >
> > > > **References:**
> > > >
> > > > [1] Van Calster, B., McLernon, D.J., van Smeden, M. et al. Calibration: the Achilles heel of predictive analytics. BMC Med 17, 230 (2019). https://doi.org/10.1186/s12916-019-1466-7
> > > >
> > > > [2] Vovk, V., Gammerman, A., & Shafer, G. (2005). Algorithmic learning in a random world. In Algorithmic Learning in a Random World. Springer US. https://doi.org/10.1007/b106715

---

### Author Response · Authors · 2022-11-10
**Clarifying motivation (part 1/2)**

We would like to thank the reviewers for their constructive comments.
In light of the reviews, here we clarify the motivation behind our work.

### Model selection vs model validation
There is an important difference between model selection and evaluation, which we have not stressed enough in the paper. Model selection consists in comparing models and selecting the best one. Model validation relates to deciding whether a given model is sufficiently good (in terms of accuracy, reliability of the probabilities, etc) to be put in production. This is particularly important in high-stake areas such as medicine or autonomous driving. As detailed below, it may be preferable not to put in production the best-selected model.

### Proper scoring rules are enough for model selection, aside for top-label problems
We agree with the reviewers that often, the irreducible loss is constant across classifiers, and thus using proper scoring rules is enough to select the best probabilistic classifier. This is not the case though in top-label formulations, where the irreducible loss changes across classifiers. In top-label schemes, as the label reflects whether the prediction of the classifier is correct or not, labels are classifier-specific. Hence, the irreducible loss changes from one classifier to another. These may be arguments against using top-label schemes, but it is a widely adopted practice [Guo et al. 2017, Minderer et al. 2021].

### Validation is needed beyond selection
Before using a model, one must often check that it meets a set of specifications, in terms of performance but also trustworthiness, fairness, robustness to varying operating conditions… Validating the compliance of a model to these criteria often requires that confidence estimates are close to posterior probabilities, as we illustrate below:

* **Trustworthiness:** Stakeholders (decision-maker, policy-maker, physician, etc) need to trust probability estimates to reason on it. Calibration is a way of providing some guarantee but does not allow stakeholders to trust confidences at the individual or subgroup level [Pleiss et al. 2017, Barda et al. 2021].
* **Fairness:** While fairness is traditionally considered on predefined groups, recent works have stressed the importance of refining guarantees at the subgroups level, either with predefined subgroups [Kleinberg et al. 2016] as in intersectional fairness or more general subgroups [Hebert-Johnson et al. 2018]. Ensuring fairness requires that probabilities are closer to the posterior probabilities than what is ensured by calibration.
* **Robustness to operating conditions:** The threshold used to go from a prediction to a decision depends on the cost of the different types of misclassification. If the operating condition is known in advance and fixed, then the threshold is fixed and good probabilities are not required for optimal decisions (cost-sensitive learning suffices). However, if these costs vary over time (for example the cost of sending a patient to intensive care may depend on how full the intensive care unit is), or if they depend on the data (for example the cost of misclassifying a fraud increases with the amount of the fraud), then the threshold for an optimal decision varies and good probabilities are required.
* **Validity of pre-trained models:** In a context where model training and usage are more and more decoupled, models are increasingly used for multiple applications with varying utility functions. Having access to the epistemic loss (calibration loss + grouping loss) allows one to assess whether, on a new task or domain, the model is far from the optimal performances, and to decide when a model may not be good enough, and needs retraining.

Validating the compliance of a model to these criteria requires assessing the ‘closeness’ of estimated probabilities to the true posterior probabilities: the amount of epistemic loss (calibration loss + grouping loss). Estimating the true posterior probabilities is however impossible in general [Vovk et al. 2005], and recent works have attempted to refine guarantees on uncertainty estimates, starting from coarse average measures (calibration) and moving to finer group levels. In particular, Hebert-Jonhson et al. [2018] have introduced the notion of multicalibration, which generalizes the fairness notion of calibration within groups [Kleinberg et al. 2016] to *every* efficiently-identifiable subgroup. Barber et al. [2019, 2020] stress the impossibility to obtain individual guarantees and introduce conditional coverage guarantees for subgroups. Pushing this trend further, we refine trust in confidence scores by evaluating the grouping loss.

---

> ### Author Response · Authors · 2022-11-10
> **Clarifying motivation (part 2/2)**
>
> ### Proper scoring rules are not enough for model validation
> Proper scores such as the Brier score or the log-loss do not fulfill this need, as they include the irreducible loss: they cannot distinguish whether the score observed is due to aleatoric uncertainty, or to a high epistemic loss. Aleatoric uncertainty reflects the inherent randomness of the problem which limits the best possible performance of a model (irreducible loss) and is unknown, while the epistemic loss measures how far a model is from the best possible one (epistemic = calibration + grouping losses). For example, consider a classifier with a Brier score of 0.15. Its confidence scores could be optimal if there is a high aleatoric uncertainty (irreducible loss close to 0.15) or poor (irreducible loss close to 0). Prior to this work, we could not distinguish the two situations since the grouping loss and irreducible loss could not be disambiguated. Now, we can evaluate and rule out a classifier that provides untrustworthy probability estimates.
>
> Brier is enough to select the best model, but it cannot be interpreted as an error rate, so it is not enough to validate that this best model can safely be put in production.
>
> &nbsp;
>
> We uploaded a revised version of the manuscript with the introduction rewritten accordingly (added text written in blue).
>
> &nbsp;
>
> ### References
>
> Barber, R. F., Candès, E. J., Ramdas, A., & Tibshirani, R. J. (2019). The limits of distribution-free conditional predictive inference. Information and Inference, 10(2), 455–482.
>
> Barber, R. F. (2020). Is distribution-free inference possible for binary regression? Electronic Journal of Statistics, 14(2), 3487–3524.
>
> Barda, N., Yona, G., Rothblum, G. N., Greenland, P., Leibowitz, M., Balicer, R., Bachmat, E., & Dagan, N. (2021). Addressing bias in prediction models by improving subpopulation calibration. Journal of the American Medical Informatics Association, 28(3), 549–558.
>
> Guo, C., Pleiss, G., Sun, Y., & Weinberger, K. Q. (2017). On Calibration of Modern Neural Networks. 34th International Conference on Machine Learning, ICML 2017, 3, 2130–2143.
>
> Hébert-Johnson, U., Kim, M. P., Reingold, O., & Rothblum, G. N. (2018). Multicalibration: Calibration for the (Computationally-Identifiable) Masses (pp. 1939–1948). PMLR.
>
> Kleinberg, J., Mullainathan, S., & Raghavan, M. (2016). Inherent Trade-Offs in the Fair Determination of Risk Scores. Leibniz International Proceedings in Informatics, LIPIcs, 67.
>
> Minderer, M., Djolonga, J., Romijnders, R., Hubis, F., Zhai, X., Houlsby, N., Tran, D., & Lucic, M. (2021). Revisiting the Calibration of Modern Neural Networks.
>
> Pleiss, G., Raghavan, M., Wu, F., Kleinberg, J., & Weinberger, K. Q. (2017). On Fairness and Calibration. Advances in Neural Information Processing Systems, 2017-December, 5681–5690.
>
> Vovk, V., Gammerman, A., & Shafer, G. (2005). Algorithmic learning in a random world. In Algorithmic Learning in a Random World. Springer US.

---

### Author Response · Authors · 2022-11-19
**General reply and acknowledgments**

We would like to thank all the reviewers for the feedback. The discussions revealed that our submission was seen as solid, novel and clear, but its utility was not well laid out. Reviewers rpCw and LCym raised the point that the manuscript did not introduce well why one would want to measure the grouping loss in practice. We believe that we had originally not positioned our problematic of interest well, validating a fitted model for production, and the paper could be read with the angle of selecting the best model in a family, a task for which existing tools are sufficient.

We have uploaded a new version of the manuscript, having changed the abstract and thoroughly rewritten the introduction to explain the needs of validating a model before putting it in production, e.g. to control the associated operational risks. We have also linked to the fairness literature, detailing how grouping loss can be seen as an extension of measures used in fairness.
Having addressed the limitation raised by reviewer, the question of utility, we have also answered all the minor points, adjusting the manuscript to be clearer and adding an appendix with a theoretical result to answer a specific point of reviewer rpCw (change in grouping loss during model recalibration).

We think that the review process made our manuscript much more interesting to the reader. We hope settings where measuring grouping loss are now much clearer.

---

### Decision · Program_Chairs · 2023-01-20

**Decision:**

Accept: poster

**Justification For Why Not Higher Score:**

Paper needs to clarify on the motivation of group loss better. Reviewers argued formalism may be slightly too involved for broad audience adoption.

**Justification For Why Not Lower Score:**

Provides interesting insights about calibration through the lens of group loss. Interesting experiments on various SoTA models / workloads. Understanding ECE and re-calibration methods are important topics and we expect future works to build on top of this paper.

**Metareview: Summary, Strengths And Weaknesses:**

The paper focuses on a decomposition of strictly proper scoring rules. Under a decomposition there is calibration loss, group loss and an irreducible term. The authors derive an estimator of a term that lower bounds the grouping loss term.  Utility of group loss can capture systematic overconfidence over a sub group and underconfidence over another group which lead to low calibration error but captured in high group loss. Authors show that this estimator is tighter than baselines to utilize it to analyze existing modern neural networks in vision and language tasks. Authors suggest even after post-hoc calibration, models continue to have large group loss.

Strength
- Offers interesting insights into calibration pointing out limitation of using ECE by itself
- Strong and rigorous theoretical analysis and support
- Novel estimator for the lower bound of group loss which was hard to measure.
- Method are novel, technically sound and non-trivial
- Technical depth required to derive such estimator
- Experimental evaluation on lots of realistic language / vision models in real-world datasets as well as synthetic data
- Paper is well written with high quality


Weakness
- Main motivation why grouping loss would be interest in practice is unclear
   - During discussion phase partially address by author rebuttal for Reviewer `LCym`  but unsatisfactory for reviewer `rpCw`.
- Unclear whether experiments support the main claim of the paper


**Note From Pc:**

if the above contains the word "oral" or "spotlight" please see: "oral" presentation means -> notable-top-5% and "spotlight" means -> notable-top-25%. As stated in our emails, we are disassociating presentation type from AC recommendations

**Summary Of Ac-Reviewer Meeting:**

During the AC-reviewer meeting, reviewers had disagreement on significance of the study and motivation for using the group loss.

Reviewer `rpCw` was not convinced group loss is an interesting quantity to analyze and believed other existing metrics were sufficient to deal with issues the paper is mainly targeting.  Other reviewers were arguing there is interesting insight about how to think about calibration and is worth sharing with the ICLR community despite the method's complexity won't be easily adopted to use by practitioners but the argument was that the paper would serve as ground to think about issues in calibration.